# Mitigating Steady-State Bias in Off-Policy TD Learning via Distributional Correction

**Emani Naga Sai Venkata Sowmya**                                    *cs19b045@iittp.ac.in*
*Department of Computer Science and Engineering*
*Indian Institute of Technology Tirupati*

**Amit Kesari**                                                      *cs19b003@iittp.ac.in*
*Department of Computer Science and Engineering*
*Indian Institute of Technology Tirupati*

**Ajin George Joseph**                                              *ajin@iittp.ac.in*
*Department of Computer Science and Engineering*
*Indian Institute of Technology Tirupati*

**Reviewed on OpenReview:** *https://openreview.net/forum?id=QLZAHgiowr*

## Abstract

We explore the off-policy value prediction problem in the reinforcement learning setting, where one estimates the value function of the target policy using the sample trajectories obtained from a behaviour policy. Importance sampling is a standard tool for correcting action-level mismatch between behaviour and target policies. However, it only addresses single-step discrepancies. It cannot correct steady-state bias, which arises from long-horizon differences in how the behaviour policy visits states. In this paper, we propose an off-policy value-prediction algorithm under linear function approximation that explicitly corrects discrepancies in state visitation distributions. We provide rigorous theoretical guarantees for the resulting estimator. In particular, we prove asymptotic convergence under Markov noise and show that the corrected update matrix has favourable spectral properties that ensure stability. We also derive an error decomposition showing that the estimation error is bounded by a constant multiple of the best achievable approximation in the function class. This constant depends transparently on the quality of the distribution estimate and the choice of features. Empirical evaluation across multiple benchmark domains demonstrates that our method effectively mitigates steady-state bias and can be a robust alternative to existing methods in scenarios where distributional shift is critical.

## 1 Introduction

In the reinforcement learning (RL) setting (Sutton & Barto, 2018; Bertsekas, 2019; Meyn, 2022), an agent learns to interact with an environment to achieve a goal or maximize its cumulative reward by performing specific actions and receiving feedback from the environment in the form of rewards. The agent sequentially refines its behaviour using the data generated by its interactions, making RL a dynamic and adaptive learning framework. The central proposition in reinforcement learning is the ability to use observed data about earlier decisions and their rewards to conclude how alternative decision policies could perform and update their course of action. RL has been applied widely, including in game simulations (Silver et al., 2018), robotics, autonomous driving (Kiran et al., 2021), medicine (Yom-Tov et al., 2017; Tejedor et al., 2020) and communication systems (Huang et al., 2019). In these domains, agents learn strategies by exploring and refining their decisions over time. For example, RL agents in game environments achieve superhuman performance by analyzing large volumes of gameplay trajectories. Robotic systems use RL data to acquire complex motor skills such as grasping or navigation. Autonomous vehicles learn safe and efficient driving

patterns by analyzing logged driving data, while communication systems optimize bandwidth allocation by reasoning over previously observed traffic and channel conditions.

In this paper, we consider the policy evaluation problem in reinforcement learning, which refers to the task of estimating the value function, which represents the expected cumulative reward from a given state following a certain policy. The policy evaluation problem has two variants: on-policy and off-policy. In on-policy prediction, one tries to estimate the value function corresponding to a given target policy using the sample trajectories generated using that target policy itself. However, in the off-policy variant (Baird et al., 1995; Precup et al., 2000; Yu, 2012), one intends to learn the value function using a sample trajectory generated using a behaviour policy that may be different from the target policy. The behaviour policy is the policy followed during data collection, and the target policy is the policy for which the value function is being estimated. In many real-world settings, an agent cannot freely interact with the environment to collect trajectories under the policy of interest and must instead rely on logged trajectories generated by a behaviour policy. Although this mismatch may introduce statistical challenges, it also provides an important advantage: off-policy data reuse enables substantially greater flexibility. An agent can learn from trajectories produced by many different behaviour policies, allowing it to exploit existing datasets without additional interaction cost. This is particularly valuable when online exploration is expensive or unsafe—for example, autonomous driving systems that learn from human driving logs, clinical decision-support models trained on retrospective treatment pathways, or communication networks that leverage historical traffic traces. Off-policy learning also enables multiple value functions to be estimated in parallel from the same data stream, supporting scalable, model-free evaluation in large systems. Consequently, the distribution induced by the behaviour policy becomes the central object shaping the learning process. Off-policy estimation methods often rely on importance sampling (IS) (Rubinstein, 1981; Glynn & Iglehart, 1989) because it is an unbiased estimator. The fundamental concept behind IS (Tokdar & Kass, 2010) is to correct the samples obtained from a sample trajectory generated by a behaviour policy to align with the likelihood of that trajectory occurring under the targeted policy. The importance sampling approach integrated with many on-policy variants, such as gradient temporal difference (Sutton et al., 2009; Yu, 2017), temporal difference with correction (Sutton et al., 2009; Yu, 2017), and temporal difference with eligibility traces (Precup et al., 2001) to obtain the off-policy solution. However, an important drawback of this technique is its susceptibility to imprecision because of the high variance induced by the importance weights (Mandel et al., 2014) and the discrepancies associated with state appearance probabilities (Tsitsiklis & Van Roy, 1997).

In this paper, we analyze the deviation of the on-policy solution from the off-policy solution due to the steady-state bias which arises due to the discrepancy in the steady-state distribution induced by the target and behaviour policies. When one observes the marginal distributions from the target policy and behaviour policy after a sufficiently long time (mixing time), the marginal distributions settle down to the steady-state which is unique to the corresponding Markov chain. Steady-state bias arises whenever the state visitation distribution in the behaviour data differs from the target policy's steady-state distribution. This occurs when the trajectory provides only partial state coverage or, even when every state appears infinitely often, visits states in proportions that differ from those induced by the target policy. As a result, the average return may not accurately reflect the true expected return, which can lead to sub-optimal behaviour. Off-policy bias correction is a fundamental challenge in reinforcement learning, particularly in settings that utilize experience replay or batch data from previously executed policies (Precup et al., 2001; Sutton et al., 2016). The systematic bias introduced into the value function estimation due to the discrepancy between the behaviour policy's stationary distribution and the target policy's state visitation distribution persists even with unbiased importance sampling corrections, as it stems from long-horizon distributional mismatch rather than single-step policy differences (Chandak et al., 2021). This steady-state bias becomes particularly problematic in long-horizon tasks where distributional mismatch accumulates over time (Jiang & Li, 2016; Tang et al., 2020). Several recent approaches have addressed this by estimating stationary distribution corrections through marginalized importance sampling (Liu et al., 2018; 2019), dual function approximation (Zhang et al., 2020), or direct optimization of distribution matching objectives (Nachum et al., 2019; Yang et al., 2020). Frameworks like Universal Off-Policy Evaluation (Chandak et al., 2021) further improve estimation by enforcing consistency between learned value functions and off-policy estimators. These methods often formulate the correction as a minimax optimization problem over density ratios or leverage policy gradient with generalized advantage estimation (Schulman et al., 2015). The convergence and stability of such meth-

ods are closely tied to the "deadly triad" of function approximation, off-policy training, and bootstrapping, which can lead to divergence without careful regularization or correction mechanisms (Voloshin et al., 2019; Wang et al., 2017; Yu, 2017).

In this paper, we fundamentally analyze off-policy temporal difference learning by tackling the critical problem of steady-state distribution mismatch. We rigorously demonstrate that long-horizon bias stems not only from policy differences but also from the divergence in how states are visited under target versus behaviour policies. Our analysis shows how such a distributional shift amplifies approximation error through the deadly triad of bootstrapping, function approximation, and off-policy sampling. To address this issue, we introduce a dual correction mechanism. It combines standard per-step action reweighting with novel parametric estimation of stationary distribution discrepancies. We further show that the resulting update converges under ergodic Markov noise. We also establish that the corrected value estimates stabilize when the rebalanced Bellman operator exhibits spectral negativity. Most significantly, we derive an error decomposition showing that the total estimation error is bounded by a constant multiple of the best achievable approximation. This constant depends on the accuracy of the distribution estimate, the conditioning of the feature matrix, and the degree of policy misalignment.

Our work distinguishes itself from prior distribution correction methods by providing a targeted solution to the persistent problem of steady-state bias in off-policy TD learning. While frameworks like DualDICE (Nachum et al., 2019) and GradientDICE (Zhang et al., 2020) address general distribution ratio estimation through complex dual optimization, our approach offers a direct, computationally efficient correction that integrates seamlessly with standard TD updates. Unlike policy optimization methods such as CQL (Kumar et al., 2020) or OptiDICE (Lee et al., 2021) which focus on policy improvement, we specifically address value prediction accuracy under distributional shift. While (Liu et al., 2018) addressed the "curse of horizon" through marginalized importance sampling, our approach uniquely identifies and corrects for the persistent steady-state bias that remains even after one-step importance sampling is applied, offering a complementary perspective on the distributional mismatch problem in off-policy evaluation. Also Emphatic TD (Sutton et al., 2016; Yu, 2015), employs recursive emphasis weighting to implicitly approximate distribution correction, preventing the deadly triad (function approximation + bootstrapping + off-policy learning) from causing divergence using emphasis weights to prioritize updates for states that are important to the target policy. Relatedly, (Hallak & Mannor, 2017) tackles distribution-mismatch bias by learning the stationary density ratio between the behaviour and target policies, but requires solving an inherently unstable fixed-point ratio-estimation problem. Our focused approach—correcting steady-state bias through explicit distribution modeling rather than general optimization frameworks—provides both theoretical clarity and practical advantages for the fundamental problem of off-policy value prediction.

## 2 Background

The reinforcement learning setting is an optimal sequential decision-making paradigm under uncertainty characterized as Markov Decision Process (MDP) (Puterman, 2014; Bertsekas, 2019; Meyn, 2022), which is a controlled, time-homogeneous, stochastic process that is defined by the 4-tuple $(S, A, P, R)$, where $S$ is the state space and $A$ is the action space. In this paper, we consider a finite state and action spaces with $S = \{s^1, s^2, ..., s^n\}$. Here $P : S \times A \times S \to [0, 1]$ is the probability transition function, where $P(s, a, s') = \mathbb{P}(\mathbf{s}_{t+1} = s'|\mathbf{s}_t = s, \mathbf{a}_t = a, \mathbf{s}_{t-1} = \cdot, \mathbf{a}_{t-1} = \cdot, \dots) = \mathbb{P}(\mathbf{s}_{t+1} = s'|\mathbf{s}_t = s, \mathbf{a}_t = a)$ is the probability that the next state is $s'$ conditioned on the fact that the current state is $s$ and the current action is $a$. Additionally, the reward function $R : S \times A \times S \to \mathbb{R}$ assigns a numerical reward to each transition. $P$ and $R$ define the dynamics of the stochastic system. At each instant, an action is chosen according to a stationary stochastic policy $\pi : S \times A \to [0, 1]$, where $\pi(\cdot|s)$ is a probability mass function over the action space $A$ conditioned on the state $s \in S$.

In this paper, we consider the prediction problem in reinforcement learning, which is defined as follows: For a given target policy $\pi$ and discount factor $\gamma \in [0, 1)$ (that represents the agent's preference for immediate rewards versus future rewards), the goal is to evaluate the value function $V_\pi \in \mathbb{R}^n$ associated with the target

policy which is defined as the expected long-run $\gamma$-discounted cost:

$$V_\pi(s) = \mathop{\mathbb{E}}_{\tau \sim \pi}\Big[R(\tau)|\mathbf{s}_0 = s\Big], s \in S, \tag{1}$$

where $R(\tau) = \sum_{t=0}^\infty \gamma^t R(\mathbf{s}_t, \mathbf{a}_t, \mathbf{s}_{t+1})$, with $\mathbf{s}_t$ represents the state at instant $t$, $\mathbf{a}_t \sim \pi(\cdot|\mathbf{s}_t)$ represents the action chosen at time $t$ and $\mathbf{s}_{t+1} \sim P(\mathbf{s}_t, \mathbf{a}_t, \cdot)$ represents the next state. Note that the above definition is well-defined as $\gamma \in [0, 1)$ and by appealing to the bounded convergence theorem.

The value function in vector form is expressed as $V_\pi = [V_\pi(s^1), V_\pi(s^2), \ldots, V_\pi(s^n)]^\top \in \mathbb{R}^n$. The value function $V_\pi$ satisfies the Bellman equation: $V_\pi = T_\pi V_\pi$, where $T_\pi : \mathbb{R}^n \to \mathbb{R}^n$ is the Bellman operator with $T_\pi U = R_\pi + \gamma P_\pi U$. Here, $P_\pi \in \mathbb{R}^{n \times n}$ with $[P_\pi]_{ss'} = \sum_{a \in A} \pi(a|s)P(s, a, s')$ and $R_\pi(s) = \sum_{s' \in S} \sum_{a \in A} \pi(a|s)P(s, a, s')R(s, a, s')$ is the one-step average reward. From the Bellman equation, one can directly compute $V_\pi = (I - \gamma P_\pi)^{-1}R_\pi$ whose computational complexity is dominated by the matrix inversion ($O(n^c)$, $c \lesssim 2.374$). In the RL setting, the model parameters $P$ and $R$ are unknown, and one seeks to learn the value function $V_\pi$ under the generative model setting, where a realization of the stochastic process in the form of an infinitely long sample trajectory $\mathbf{s}_0, \mathbf{a}_0, \mathbf{r}_1, \mathbf{s}_1, \mathbf{a}_1, \mathbf{r}_2, \mathbf{s}_2, \ldots$ is available, with $\mathbf{s}_0 \sim P_0$ ($P_0$ initial distribution), $\mathbf{a}_t \sim \pi(\cdot|\mathbf{s}_t)$, $\mathbf{s}_{t+1} \sim P(\mathbf{s}_t, \mathbf{a}_t, \cdot)$ and $\mathbf{r}_{t+1} = R(\mathbf{s}_t, \mathbf{a}_t, \mathbf{s}_{t+1})$.

Temporal difference (TD) learning (Sutton & Barto, 2018) is the classical approach for the prediction problem, where the value function $V_t \in \mathbb{R}^n$ is iteratively updated in the direction of the temporal difference $\mathbf{r}_{t+1} + \gamma V_t(\mathbf{s}_{t+1}) - V_t(\mathbf{s}_t)$. However, when the state space is large, this method suffers from the curse of dimensionality (Tsitsiklis & Van Roy, 1997; Sutton & Barto, 2018). To overcome this, one effective strategy is to represent the value function in a lower-dimensional subspace, thus reducing computational and storage demands. Here one approximates $V_\pi$ using linear function approximation by projecting it into the subspace $\{\Phi x \mid x \in \mathbb{R}^k\} \subset \mathbb{R}^n$, where $k \ll n$ (Tsitsiklis & Van Roy, 1997). The feature matrix $\Phi$ contains basis functions that capture the critical characteristics of the state space. This projection not only renders the learning process more tractable but also preserves the essential dynamics of the original high-dimensional problem.

$$\Phi = \begin{bmatrix} -- \phi(s^1)^\top -- \\ \vdots \\ -- \phi(s^n)^\top -- \end{bmatrix}_{n \times k}, \tag{2}$$

where, $\phi(s) = [\phi_1(s), \phi_2(s), \ldots \phi_k(s)]^\top \in \mathbb{R}^k$ is called the feature vector associated with the state $s \in S$ and $\phi_i : S \to R$ are feature/basis functions. The most commonly used parameterized basis functions include radial basis functions (RBFs), polynomials, and Fourier basis functions. Radial basis functions are typically expressed in a Gaussian form: $\phi_i(s) = \exp(-(2\sigma_i^2)^{-1}||s - \mu_i||^2)$, depends solely on the distance between the state and the centre $\mu_i$, relative to the feature width, $\sigma_i$, with a parameter size of the order $\Theta(k)$.

In this paper, we consider the off-policy variant of the prediction problem (Precup et al., 2001; Sutton & Barto, 2018), where one seeks to estimate $V_\pi$, using a sample trajectory, where action at every instant is generated using a behaviour policy $\pi_b$ that may be different from the target policy $\pi$. This implies that for the given infinitely long sample trajectory $\tau_b = \mathbf{s}_0, \mathbf{a}_0, \mathbf{r}_1, \mathbf{s}_1, \mathbf{a}_1, \mathbf{r}_2, \mathbf{s}_2, \mathbf{a}_2, \ldots$, we have $\mathbf{s}_0 \sim P_0$, $\mathbf{a}_t \sim \pi_b(\cdot|\mathbf{s}_t)$, $\mathbf{s}_{t+1} \sim P(\mathbf{s}_t, \mathbf{a}_t, \cdot)$ and $\mathbf{r}_{t+1} = R(\mathbf{s}_t, \mathbf{a}_t, \mathbf{s}_{t+1})$.

**Assumption 1** (Ergodic Behaviour Policy)**.** *The Markov chain $\{\mathbf{s}_t\}_{t \geq 0}$ induced by the behaviour policy $\pi_b$ satisfies:*

   *(i) **Irreducibility:** $\forall s, s' \in S$, $\exists t \in \mathbb{N}$ such that $P_{\pi_b}^t(s, s') > 0$.*

   *(ii) **Aperiodicity:** The greatest common divisor of $\{t \geq 1 : P_{\pi_b}^t(s, s) > 0\}$ is 1 for every $s \in S$.*

*Consequently, the chain admits a unique stationary distribution $\nu_b$ with $\nu_b(s) > 0 \, \forall s \in S$ and $\nu_b^\top P_{\pi_b} = \nu_b^\top$.*

**Assumption 2** (Feature Independence)**.** *The feature matrix $\Phi \in \mathbb{R}^{n \times k}$ satisfies $\mathrm{rank}(\Phi) = k$, implying*

$$\sigma_{\min}(\Phi^\top \Phi) > 0, \qquad \ker(\Phi) = \{0\}, \qquad \Phi^\top \Phi \succ 0,$$

*where $\sigma_{\min}$ denotes the minimum singular value and $\succ 0$ denotes positive definiteness.*

**Assumption 3** (Coverage). *The behaviour policy $\pi_b$ dominates $\pi$ in the Radon–Nikodym sense:*

$$\forall\, (s,a) \in S \times A,\ \pi(a \mid s) > 0 \implies \pi_b(a \mid s) > 0.$$

*Equivalently, the importance ratio $\rho_t = \dfrac{\pi(\mathbf{a}_t \mid \mathbf{s}_t)}{\pi_b(\mathbf{a}_t \mid \mathbf{s}_t)}$ is almost surely bounded:* $\sup_t \rho_t < \infty$.

In off-policy linear function approximation, one projects the value function $V_\pi$ onto the column space of $\Phi$ (Tsitsiklis & Van Roy, 1997):

$$w^* = \arg\min_{w \in \mathbb{R}^k} \|V_\pi - \Phi w\|_{\nu_b}^2, \tag{3}$$

where the weighted norm is defined as $\|w\|_\nu^2 = \sum_{i=1}^{k} \nu_i w_i^2$. Here, $\nu_b$ is the unique steady-state distribution of the behaviour policy's Markov chain (i.e., $\nu_b(s) = \lim_{t\to\infty} \mathbb{P}(\mathbf{s}_t = s)$ and $\nu_b^\top P_{\pi_b} = \nu_b^\top$). Since $\{\Phi w \mid w \in \mathbb{R}^k\}$ is closed and convex, a unique $w^*$ exists ($\Phi$ has full column rank), yielding the approximation $V_\pi(s) \approx \phi(s)^\top w^*$ for all $s$. This optimization is solved by the off-policy TD update (Precup et al., 2001; 2000):

$$\mathbf{w}_{t+1} = \mathbf{w}_t + \alpha_t \rho_t \Big( \mathbf{r}_{t+1} + \gamma \phi(\mathbf{s}_{t+1})^\top \mathbf{w}_t - \phi(\mathbf{s}_t)^\top \mathbf{w}_t \Big) \phi(\mathbf{s}_t),$$

with the importance sampling ratio $\rho_t = \frac{\pi(\mathbf{a}_t \mid \mathbf{s}_t)}{\pi_b(\mathbf{a}_t \mid \mathbf{s}_t)}$, which corrects for the policy mismatch in the behaviour data. The limit point $w_{\text{off}}^{\text{TD}}$ of off-policy TD learning with linear function approximation is characterized by the fixed-point equation(Yu, 2012)

$$\Phi^\top \Xi_{\nu_b} (I - \gamma P_\pi) \Phi w_{\text{off}}^{\text{TD}} = \Phi^\top \Xi_{\nu_b} R_\pi, \tag{4}$$

which represents a projection of the Bellman equation onto the feature space weighted by the behaviour policy's stationary distribution. This solution constitutes the best approximation within the function class that satisfies the Bellman residual minimization under the behaviour policy's steady-state distributional mismatch, rather than the target policy's natural state visitation pattern.

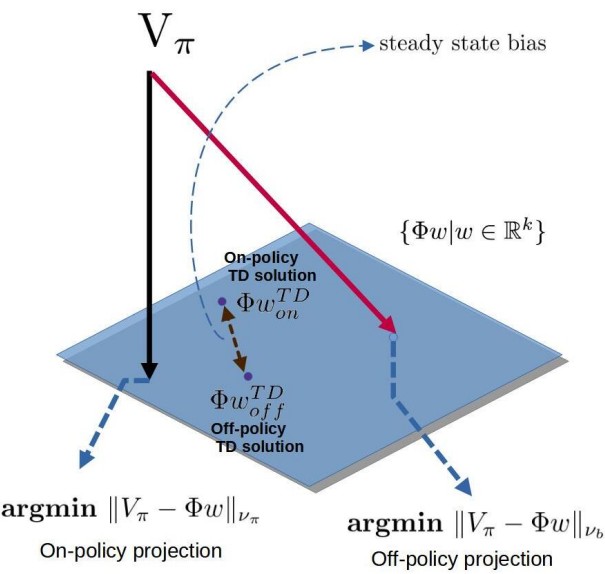

**Figure 1:** Illustration of steady-state bias in off-policy prediction: The mismatch between behaviour policy's steady-state distribution $\nu_b$ and that of target policy causes persistent prediction error, even after one-step importance sampling correction

To establish further theoretical guarantees for the off-policy TD method with linear function approximation, we first analyze key properties of the value function operator under the behaviour policy's stationary distribution. The following lemma quantifies fundamental operator norm bounds that govern the propagation of approximation errors through the Bellman operator.

**Lemma 1.** *Let $\nu_b$ be a strictly positive probability distribution over states, $P_\pi$ a Markov transition matrix induced by policy $\pi$, and $\gamma \in [0,1)$ a discount factor. Then the $\nu_b$-weighted operator norms satisfy:*

$$\|P_\pi\|_{\nu_b} \leq \sqrt{\kappa_b} \text{ and } \|I - \gamma P_\pi\|_{\nu_b} \leq 1 + \gamma\sqrt{\kappa_b} \tag{5}$$

*where $\|A\|_{\nu_b} = \sup_{\mathbf{x} \neq 0} \frac{\|A\mathbf{x}\|_{\nu_b}}{\|\mathbf{x}\|_{\nu_b}}$ and $\|\mathbf{x}\|_{\nu_b}^2 = \sum_s \nu_b(s)\mathbf{x}(s)^2$, with and the distribution mismatch coefficient $\kappa_b = \max_{s' \in S} \frac{\sum_s \nu_b(s) P_\pi(s' \mid s)}{\nu_b(s')}$.*

*Proof.* Let $\mu_b(s') = \sum_s \nu_b(s) P_\pi(s' \mid s)$. Then,

$$\|P_\pi\mathbf{x}\|_{\nu_b}^2 = \sum_s \nu_b(s) \left( \sum_{s'} P_\pi(s' \mid s)\mathbf{x}(s') \right)^2$$
$$\leq \sum_s \nu_b(s) \sum_{s'} P_\pi(s' \mid s)\mathbf{x}(s')^2 \quad \text{(Jensen's inequality)}$$
$$= \sum_{s'} \mathbf{x}(s')^2 \sum_s \nu_b(s) P_\pi(s' \mid s)$$
$$= \sum_{s'} \mathbf{x}(s')^2 \mu_b(s') = \sum_{s'} \nu_b(s')\mathbf{x}(s')^2 \frac{\mu_b(s')}{\nu_b(s')}$$
$$\leq \kappa_b \sum_{s'} \nu_b(s')\mathbf{x}(s')^2 = \kappa_b\|\mathbf{x}\|_{\nu_b}^2, \text{ where } \kappa_b = \max_{s'} \frac{\mu_b(s')}{\nu_b(s')}.$$

Thus we have the following operator norm bound:

$$\|P_\pi\|_{\nu_b} = \sup_{\mathbf{x} \neq 0} \frac{\|P_\pi\mathbf{x}\|_{\nu_b}}{\|\mathbf{x}\|_{\nu_b}} \leq \sqrt{\kappa_b}$$

Now for the composite operator, we get

$$\|(I - \gamma P_\pi)\mathbf{x}\|_{\nu_b} \leq \underbrace{\|I\mathbf{x}\|_{\nu_b}}_{\leq \|\mathbf{x}\|_{\nu_b}} + \gamma \underbrace{\|P_\pi\mathbf{x}\|_{\nu_b}}_{\leq \sqrt{\kappa_b}\|\mathbf{x}\|_{\nu_b}} \leq (1 + \gamma\sqrt{\kappa_b})\|\mathbf{x}\|_{\nu_b}$$
$$\Rightarrow \|I - \gamma P_\pi\|_{\nu_b} \leq 1 + \gamma\sqrt{\kappa_b}.$$

$\square$

Central to the above result is the *distribution mismatch coefficient* $\kappa_b$, which captures the maximum density ratio between the next-state distribution induced by the target policy and the stationary distribution of the behaviour policy. We now characterize the asymptotic approximation error of the off-policy TD solution under linear function approximation. The following theorem establishes a bound on the error $\|\Phi\mathbf{w}_{\text{off}}^{\text{TD}} - V_\pi\|_{\nu_b}$ of the TD fixed-point solution relative to the fundamental approximation limit $\|\Phi\mathbf{w}^* - V_\pi\|_{\nu_b}$.

**Theorem 1** (Error Bound for Off-policy TD). *Under Assumptions 1-3 and negative definiteness of $\Lambda_o = \Phi^\top \Xi_{\nu_b}(I - \gamma P_\pi)\Phi$, the solution $\mathbf{w}_{\text{off}}^{\text{TD}}$ satisfies:*

$$\left\|\Phi\mathbf{w}_{\text{off}}^{\text{TD}} - V_\pi\right\|_{\nu_b} \leq \left( \frac{\sigma_{\max}^2(\Phi)(\max_s \nu_b(s))^{3/2}(1 + \gamma\sqrt{\kappa_b})}{\lambda_{\min}(-\Lambda_o)\sqrt{\min_s \nu_b(s)}} + 1 \right) \left\|\Phi\mathbf{w}^* - V_\pi\right\|_{\nu_b}$$

*where $\sigma_{\max}^2(\Phi) = \lambda_{max}(\Phi^\top\Phi)$ and $\Xi_{\nu_b} = diag(\nu_b)$.*

*Proof.* From (4), we have

$$\Phi^\top \Xi_{\nu_b}(I - \gamma P_\pi)\Phi\,\mathbf{w}_{\mathrm{off}}^{\mathrm{TD}} = \Phi^\top \Xi_{\nu_b} R_\pi. \tag{6}$$

Also, the true value function $V_\pi$ satisfies the following Bellman equation:

$$V_\pi = R_\pi + \gamma P_\pi V_\pi \tag{7}$$

Now we bound $\left\|\Phi\mathbf{w}_{\mathrm{off}}^{\mathrm{TD}} - V_\pi\right\|_{\nu_b}$. Let $\mathbf{w}^*$ be the best linear approximator under $\nu_b$. Then,

$$\mathbf{w}^* = \arg\min_{\mathbf{w}} \|\Phi\mathbf{w} - V_\pi\|_{\nu_b}$$

so that $\Phi\mathbf{w}^* = \Pi_{\nu_b} V_\pi$, the projection of $V_\pi$ onto the column space of $\Phi$ under the $\nu_b$-weighted norm. The error decomposes as follows:

$$\Phi\mathbf{w}_{\mathrm{off}}^{\mathrm{TD}} - V_\pi = \left(\Phi\mathbf{w}_{\mathrm{off}}^{\mathrm{TD}} - \Phi\mathbf{w}^*\right) + \left(\Phi\mathbf{w}^* - V_\pi\right). \tag{8}$$

Hence,

$$\left\|\Phi\mathbf{w}_{\mathrm{off}}^{\mathrm{TD}} - V_\pi\right\|_{\nu_b} \leq \left\|\Phi\mathbf{w}_{\mathrm{off}}^{\mathrm{TD}} - \Phi\mathbf{w}^*\right\|_{\nu_b} + \left\|\Phi\mathbf{w}^* - V_\pi\right\|_{\nu_b}. \tag{9}$$

We define the approximation error $\varepsilon_{\mathrm{approx}} = \Phi\mathbf{w}^* - V_\pi$. To prove the claim of the theorem, we bound the term $\left\|\Phi\mathbf{w}_{\mathrm{off}}^{\mathrm{TD}} - \Phi\mathbf{w}^*\right\|_{\nu_b}$ in terms of $\left\|\varepsilon_{\mathrm{approx}}\right\|_{\nu_b}$. Note that both $\Phi\mathbf{w}_{\mathrm{off}}^{\mathrm{TD}}$ and $\Phi w^*$ lie in the column space of $\Phi$. The vector $\Phi w_{\mathrm{TD}}^{\mathrm{off}}$ satisfies

$$\Phi^\top \Xi_{\nu_b}(I - \gamma P_\pi)\Phi w_{\mathrm{TD}}^{\mathrm{off}} = \Phi^\top \Xi_{\nu_b} R_\pi, \tag{10}$$

whereas the projection $\Phi\mathbf{w}^*$ satisfies

$$\Phi^\top \Xi_{\nu_b}\left(\Phi\mathbf{w}^* - V_\pi\right) = 0.$$

Also, by multiplying the Bellman equation (7) by $\Phi^\top \Xi_{\nu_b}$, we obtain

$$\Phi^\top \Xi_{\nu_b} V_\pi = \Phi^\top \Xi_{\nu_b} R_\pi + \gamma\,\Phi^\top \Xi_{\nu_b} P_\pi V_\pi$$
$$\Rightarrow \Phi^\top \Xi_{\nu_b}(I - \gamma P_\pi)V_\pi = \Phi^\top \Xi_{\nu_b} R_\pi. \tag{11}$$

Combining (10) and (11), we get

$$\Phi^\top \Xi_{\nu_b}(I - \gamma P_\pi)\Phi\,\mathbf{w}_{\mathrm{off}}^{\mathrm{TD}} = \Phi^\top \Xi_{\nu_b}(I - \gamma P_\pi)V_\pi$$
$$\Rightarrow \Phi^\top \Xi_{\nu_b}(I - \gamma P_\pi)(\Phi\,\mathbf{w}_{\mathrm{off}}^{\mathrm{TD}} - V_\pi) = 0$$
$$\Rightarrow \Phi^\top \Xi_{\nu_b}(I - \gamma P_\pi)\,e = 0, \tag{12}$$

where $e = \Phi\,\mathbf{w}_{\mathrm{off}}^{\mathrm{TD}} - V_\pi$. Further, from (8), we have $e = \Phi\left(\mathbf{w}_{\mathrm{off}}^{\mathrm{TD}} - \mathbf{w}^*\right) + \varepsilon_{\mathrm{approx}}$. Substituting this above, we obtain

$$\Phi^\top \Xi_{\nu_b}(I - \gamma P_\pi)\left[\Phi\left(\mathbf{w}_{\mathrm{off}}^{\mathrm{TD}} - \mathbf{w}^*\right) + \varepsilon_{\mathrm{approx}}\right] = 0$$
$$\Rightarrow \Phi^\top \Xi_{\nu_b}(I - \gamma P_\pi)\Phi\left(\mathbf{w}_{\mathrm{off}}^{\mathrm{TD}} - \mathbf{w}^*\right) + \Phi^\top \Xi_{\nu_b}(I - \gamma P_\pi)\varepsilon_{\mathrm{approx}} = 0$$
$$\Rightarrow \Phi^\top \Xi_{\nu_b}(I - \gamma P_\pi)\Phi\left(\mathbf{w}_{\mathrm{off}}^{\mathrm{TD}} - \mathbf{w}^*\right) = -\Phi^\top \Xi_{\nu_b}(I - \gamma P_\pi)\varepsilon_{\mathrm{approx}}$$
$$\Rightarrow \Lambda_o\left(\mathbf{w}_{\mathrm{off}}^{\mathrm{TD}} - \mathbf{w}^*\right) = \Phi^\top \Xi_{\nu_b}(I - \gamma P_\pi)\varepsilon_{\mathrm{approx}}. \tag{13}$$

Now,

$$\left\|\Phi^\top \Xi_{\nu_b}(I - \gamma P_\pi)\varepsilon_{\mathrm{approx}}\right\| \leq \left\|\Phi^\top\right\| \left\|\Xi_{\nu_b}\right\| \left\|(I - \gamma P_\pi)\varepsilon_{\mathrm{approx}}\right\|, \tag{14}$$

where $\|\cdot\|$ is the spectral norm. But, $\|\Xi_{\nu_b}\| = \max_s \nu_b(s)$, and

$$
\begin{aligned}
\left\|(I - \gamma P_\pi)\varepsilon_{\text{approx}}\right\|^2 &= \sum_s \left[(I - \gamma P_\pi)\varepsilon_{\text{approx}}\right](s)^2 \\
&\leq \sum_s \frac{1}{\min_s \nu_b(s)} \nu_b(s)\left[(I - \gamma P_\pi)\varepsilon_{\text{approx}}\right](s)^2 \\
&= \frac{1}{\min_s \nu_b(s)} \left\|(I - \gamma P_\pi)\varepsilon_{\text{approx}}\right\|_{\nu_b}^2.
\end{aligned}
\tag{15}
$$

Hence,

$$
\left\|(I - \gamma P_\pi)\varepsilon_{\text{approx}}\right\| \leq \frac{1 + \gamma\sqrt{\kappa_b}}{\sqrt{\min_s \nu_b(s)}} \|\varepsilon_{\text{approx}}\|_{\nu_b},
$$

and therefore

$$
\left\|\Phi^\top \Xi_{\nu_b}(I - \gamma P_\pi)\varepsilon_{\text{approx}}\right\| \leq \|\Phi^\top\| \frac{\max_s \nu_b(s)}{\sqrt{\min_s \nu_b(s)}} \left(1 + \gamma\sqrt{\kappa_b}\right) \|\varepsilon_{\text{approx}}\|_{\nu_b}.
\tag{16}
$$

Note that $\|\Phi^\top\| = \|\Phi\|$, and the spectral norm of $\Phi$ is the large singular value. Let $\sigma_{\max}(\Phi) = \|\Phi\|$. Also note that, since $\Lambda_o$ is negative definite, we have

$$
\|\Lambda_o^{-1}\| \leq \frac{1}{\lambda_{\min}(-\Lambda_o)},
\tag{17}
$$

where $\lambda_{\min}(-\Lambda_o)$ is the smallest eigenvalue of $-\Lambda_o$. Combining (13), (14), (16) and (17), we get

$$
\left\|\mathbf{w}_{\text{off}}^{\text{TD}} - \mathbf{w}^*\right\| \leq \frac{\sigma_{\max}(\Phi)}{\lambda_{\min}(-\Lambda_o)} \frac{\max_s \nu_b(s)}{\sqrt{\min_s \nu_b(s)}} \left(1 + \gamma\sqrt{\kappa_b}\right) \|\varepsilon_{\text{approx}}\|_{\nu_b}.
\tag{18}
$$

For the projected error in the $\nu_b$-norm, we have

$$
\left\|\Phi\left(\mathbf{w}_{\text{off}}^{\text{TD}} - w^*\right)\right\|_{\nu_b} \leq \|\Phi\|_{\nu_b} \left\|w_{\text{TD}}^{\text{off}} - w^*\right\|,
\tag{19}
$$

where $\|\Phi\|_{\nu_b}$ is the operator norm of $\Phi$ from the Euclidean space to the $\nu_b$-normed space. Specifically,

$$
\begin{aligned}
\|\Phi\mathbf{w}\|_{\nu_b}^2 &= \mathbf{w}^\top \Phi^\top \Xi_{\nu_b} \Phi\mathbf{w} \leq \lambda_{\max}(\Phi^\top \Xi_{\nu_b}\Phi)\|\mathbf{w}\|^2 \\
&\Rightarrow \|\Phi\|_{\nu_b} \leq \sqrt{\lambda_{\max}(\Phi^\top \Xi_{\nu_b}\Phi)}.
\end{aligned}
\tag{20}
$$

Note that $\Phi^\top \Xi_{\nu_b} \Phi$ is a $k \times k$ matrix, and its largest eigenvalue is at most $\max_s \nu_b(s) \cdot \lambda_{\max}(\Phi^\top\Phi)$, because $\Xi_{\nu_b} \leq \max_s \nu_b(s)I$. And $\lambda_{\max}(\Phi^\top\Phi) = \sigma_{\max}^2(\Phi)$. Hence,

$$
\|\Phi(w_{\text{off}}^{\text{TD}} - \mathbf{w}^*)\|_{\nu_b} \leq \sigma_{\max}(\Phi)\sqrt{\max_s \nu_b(s)} \cdot \|\mathbf{w}_{\text{off}}^{\text{TD}} - \mathbf{w}^*\|
\tag{21}
$$

Hence from (9), (18) and (21), we get

$$
\|\Phi\mathbf{w}_{\text{off}}^{\text{TD}} - V_\pi\|_{\nu_b} \leq \|\Phi(\mathbf{w}_{\text{off}}^{\text{TD}} - \mathbf{w}^*)\|_{\nu_b} + \|\epsilon_{\text{approx}}\|_{\nu_b} \leq \left(\frac{\sigma_{\max}^2(\Phi)(\max_s \nu_b(s))^{\frac{3}{2}}\left(1 + \gamma\sqrt{\kappa_b}\right)}{\lambda_{\min}(-\Lambda_o)\sqrt{\min_s \nu_b(s)}} + 1\right)\|\varepsilon_{\text{approx}}\|_{\nu_b}.
$$

$\square$

The above theorem bound reveals three critical components that affect the off-policy TD convergence: First, the $\sigma_{\max}^2(\Phi)(\max_s \nu_b(s))^{3/2}$ term exposes the sensitivity to feature scaling and distribution skew, which shows that even optimal representations suffer when $\nu_b$ is non-uniform or features are poorly conditioned. Second, the $(1 + \gamma\sqrt{\kappa_b})$ factor quantifies how policy divergence ($\kappa_b \gg 1$) amplifies approximation error through

temporal credit assignment - a manifestation of the deadly triad where bootstrapping, function approximation, and off-policy sampling interact destructively. Third, the dependence on $\lambda_{\min}(-\Lambda_o)^{-1}$ formalizes the hardness of Bellman inversion under distribution shift, as $\Lambda_o$ becomes ill-conditioned when the behaviour policy's transitions poorly align with the target dynamics. This provides a closed-form characterization of deadly triad interactions in off-policy TD convergence. The bound exclusively characterizes the fundamental approximation error of the asymptotic off-policy TD solution, isolating it from transient algorithmic effects. When $\pi = \pi_b$ ($\kappa_b = 1$), the bound simplifies to the on-policy case, but the exponential scaling $\gamma\sqrt{\kappa_b}$ explains the severe degradation under policy mismatch.

**Theorem 2.** *Under Assumptions 1, 2, and 3, if $\kappa_b \gamma^2 < 1$ then $\Lambda_o$ is negative definite.*

*Proof.* For any $\mathbf{w} \neq 0$, let $\mathbf{u} = \Phi\mathbf{w}$. By Assumption 2, $\mathbf{u} \neq 0$. Consider the quadratic form:

$$
\begin{aligned}
\mathbf{w}^\top \Lambda_o \mathbf{w} &= \mathbf{w}^\top \Phi^\top \Xi_{\nu_b}(\gamma P_\pi - I)\Phi\mathbf{w} \\
&= \mathbf{u}^\top \Xi_{\nu_b}(\gamma P_\pi - I)\mathbf{u} \\
&= \gamma \underbrace{\mathbf{u}^\top \Xi_{\nu_b} P_\pi \mathbf{u}}_{Q_1} - \underbrace{\mathbf{u}^\top \Xi_{\nu_b}\mathbf{u}}_{Q_2}
\end{aligned}
\tag{22}
$$

Now $Q_2 = \mathbf{u}^\top \Xi_{\nu_b}\mathbf{u} = \sum_s \nu_b(s)\mathbf{u}(s)^2 = \|\mathbf{u}\|_{\nu_b}^2 > 0$. Since $\nu_b > 0$ (ergodicity) and $\mathbf{u} \neq 0$, we have

$$
\begin{aligned}
Q_1 = \mathbf{u}^\top \Xi_{\nu_b} P_\pi \mathbf{u} &= \sum_s \nu_b(s)\mathbf{u}(s)(P_\pi \mathbf{u})(s) \\
&= \sum_s \nu_b(s)\mathbf{u}(s)\Big(\sum_{s'} P_\pi(s'|s)\mathbf{u}(s')\Big) \\
&\leq \|\mathbf{u}\|_{\nu_b} \cdot \|P_\pi \mathbf{u}\|_{\nu_b} \quad \text{(by Cauchy-Schwarz inequality)}
\end{aligned}
\tag{23}
$$

Further, by Lemma 1, we have

$$
\|P_\pi \mathbf{u}\|_{\nu_b}^2 = \kappa_b \|\mathbf{u}\|_{\nu_b}^2
\tag{24}
$$

Therefore, from (23) and (24), we have

$$
Q_1 \leq |Q_1| \leq \|\mathbf{u}\|_{\nu_b} \cdot \|P_\pi \mathbf{u}\|_{\nu_b} \leq \sqrt{\kappa}\|u\|_{\nu_b}^2
$$

Substitute into (22) to obtain

$$
\mathbf{w}^\top \Lambda_o \mathbf{w} \leq \gamma\sqrt{\kappa_b}\|\mathbf{u}\|_{\nu_b}^2 - \|\mathbf{u}\|_{\nu_b}^2 = (\gamma\sqrt{\kappa_b} - 1)\|\mathbf{u}\|_{\nu_b}^2
\tag{25}
$$

Since $\|\mathbf{u}\|_{\nu_b}^2 > 0$ and $\gamma\sqrt{\kappa_b} - 1 < 0$ *iff* $\kappa_b < \frac{1}{\gamma^2}$ we have:

$$
\mathbf{w}^\top \Lambda_o \mathbf{w} \leq (\gamma\sqrt{\kappa_b} - 1)\|\mathbf{u}\|_{\nu_b}^2 < 0 \quad \text{when} \quad \kappa_b \gamma^2 < 1
$$

Equality holds only if $\mathbf{w} = 0$, proving $\Lambda_o$ is negative definite. $\qquad\square$

**Corollary 1.** *When $\pi = \pi_b$, $\Lambda_o$ is negative definite for any $\gamma < 1$.*

*Proof.* When $\pi = \pi_b$, we have $\kappa_b = 1$. Then:

$$
\mathbf{w}^\top \mathbf{A}\mathbf{w} \leq (\gamma - 1)\|\mathbf{v}\|_{\nu_b}^2 < 0 \quad \forall \mathbf{w} \neq 0
$$

$\qquad\square$

Theorem 2 establishes a fundamental condition for convergence in off-policy temporal difference learning: when the product of the policy alignment constant $\kappa_b$ and the squared discount factor $\gamma^2$ is less than one, the critical matrix governing the TD update dynamics becomes negative definite. This condition, $\kappa_b \gamma^2 < 1$, provides profound theoretical insight into the feasibility of off-policy learning. The policy alignment constant

$\kappa_b$ quantifies the maximum discrepancy between the next-state distribution under the target policy and the behaviour policy's stationary distribution ($\nu_b$). When $\kappa_b$ is large, it indicates significant distributional mismatch; certain states are visited much more frequently under the target policy than would be expected from the behaviour policy's steady-state distribution. The theorem reveals that such mismatches become increasingly problematic as the discount factor $\gamma$ approaches 1, explaining why long-horizon tasks with high $\gamma$ values are particularly challenging for off-policy methods. Notably, when policies are identical ($\pi = \pi_b$), we have $\kappa_b = 1$, and the condition simplifies to $\gamma < 1$, which always holds for standard MDPs. However, as policy dissimilarity increases ($\kappa_b > 1$), the allowable discount factor must decrease to maintain convergence guarantees. This theoretical boundary precisely characterizes the "deadly triad" interaction between function approximation, bootstrapping, and off-policy learning, and directly motivates the steady-state bias correction.

## 3  Our Algorithm

Here, we propose a double correction approach to address the discrepancy introduced by the steady-state distribution of the behaviour policy in the solution of the off-policy TD algorithm by effectively reducing the policy alignment constant through distributional reweighting, while simultaneously incorporating per-step policy mismatch correction $\rho_t$. To achieve this, we employ the importance sampling method to the existing off-policy TD method with one-step lookahead, where a separate state-distribution reweighting factor $q(s)/h(s; \theta^*, \lambda^*)$ is tied to the existing off-policy TD recursion. Here $q(\cdot)$ is the design probability distribution to which the solutions are guided, and $h(\cdot; \theta^*, \lambda^*) = \lambda_1^* g_{\theta_1^*}(\cdot) + \cdots + \lambda_\ell^* g_{\theta_\ell^*}(\cdot)$ is a surrogate probability mixture distribution chosen from a parametrized family of distributions $\{g_\theta : \mathbb{R}^p \to \mathbb{R} | \theta \in \Theta, \int g(x)dx = 1, g \geq 0\}$ which best approximates the steady-state distribution of the Markov chain induced by the behaviour policy with respect to the Kullback-Leibler divergence (moment projection).

$$\begin{bmatrix} \theta^* \\ \lambda^* \end{bmatrix} = \underset{\substack{\theta_i \in \Theta, \\ \lambda_i \in [0,1]}}{\arg\min} \mathcal{D}_{\mathrm{KL}}(\nu_b \| \lambda_1 g_{\theta_1} + \cdots + \lambda_\ell g_{\theta_\ell}), \text{ subject to } \sum_{i=1}^{\ell} \lambda_i = 1, \tag{26}$$

where $\mathcal{D}_{\mathrm{KL}}(f \| g) = \mathbb{E}_f \left[ \log \frac{f(\mathbf{x})}{g(\mathbf{x})} \right]$.

To model the component distributions $g_\theta$ efficiently, we employ a flexible and analytically tractable parametric family. One theoretically well-founded choice is the Natural Exponential Family (NEF) (Brown, 1986). The NEF is a class of probability distributions which provides a unified framework for probability distributions through its canonical form that encompasses many commonly used distributions such as Gaussian, Poisson, and Bernoulli distributions, among others. The NEF has several desirable properties, including a closed-form expression with a convex log-partition function, which simplifies the computation of the importance sampling ratio and allows for efficient parameter updates during the learning process. A parameterized family $\{g_\theta | \theta \subseteq \mathbb{R}^b\}$ is called a natural exponential family if $g_\theta(x) = \exp(\theta^T \Gamma(x) - K(\theta))$, where $\Gamma : \mathbb{R}^p \to \mathbb{R}^b$ and $K : \mathbb{R}^b \to \mathbb{R}$ are continuous functions with $\Theta = \{\theta \in \mathbb{R}^b | \ |K(\theta)| < \infty\}$. Note that $K(\theta)$ is strictly convex in the interior of $\Theta$ and $\nabla K(\theta) = \mathbb{E}_{g_\theta}[\Gamma(\mathbf{x})]$. Also, $\nabla_\theta^2 K(\theta) = \mathtt{Cov}_{g_\theta}[\Gamma(\mathbf{x})] \succ 0$. These ensure the Fisher information matrix $I(\theta) = \nabla_\theta^2 K(\theta)$ is non-degenerate, guaranteeing well-posed maximum likelihood estimation. While all the NEF member distributions provide analytical tractability through their exponential structure, we employ Gaussian mixture models in our experiments for their superior approximation capabilities. We formalize this approximation and aim to guarantee that, with a sufficient number of components, the KL-divergence between the true steady-state distribution and its Gaussian mixture approximation can be made arbitrarily small. A subtlety arises because the behaviour policy's stationary distribution $\nu_b$ is discrete and therefore lacks a density, making the KL divergence ill-defined. To address this, we mollify $\nu_b$ by convolving it with a Gaussian kernel, yielding a smooth surrogate. We first define the following:

For the stationary distribution of the behaviour policy $\nu_b$, which is supported on finitely many states $\{s^1, \ldots, s^n\} \subset \mathbb{R}^p$ and the smoothing parameter $\sigma > 0$, we define the mollified density as follows:

$$p_\sigma(x) = (\nu_b * \mathcal{N}(0, \sigma^2 I_p))(x) = \sum_{i=1}^{n} \nu_b(s^i) \mathcal{N}(x; s^i, \sigma^2 I_p),$$

For each $\sigma > 0$, $p_\sigma$ is a valid probability density, being a finite convex combination of Gaussian kernels and is absolutely continuous on $\mathbb{R}^p$. Further, for a compact region $X$ and clipping level $\eta > 0$, we define the clipped proxy

$$f_\eta(x) = p_\sigma(x)\mathbf{1}_{X^c}(x) + \max\{p_\sigma(x), \eta\}\,\mathbf{1}_X(x), \tag{27}$$

where clipping is applied only on $X$ and the density remains unchanged on $X^c$. The normalizing constant is

$$Z_\eta = \int_{\mathbb{R}^p} f_\eta(x)\,dx, \qquad \tilde{p}_{\sigma, X, \eta}(x) = \frac{f_\eta(x)}{Z_\eta}.$$

Consider a Gaussian-mixture model $h_{\theta, \lambda}(x) = \sum_{j=1}^\ell \lambda_j g_{\theta_j}(x)$ with $\lambda \in \Delta^\ell$ the probability simplex of mixture weights, and each component $g_{\theta_j}$ has a non-degenerate covariance matrix.

**Theorem 3** (Gaussian-mixture approximation of the steady-state distribution)**.** *For any $\varepsilon > 0$, there exist $\sigma > 0$, $0 < \eta < 1$, and $\ell \in \mathbb{N}$ such that for some parameters $(\theta^*, \lambda^*)$*

$$\mathcal{D}_{\mathrm{KL}}\big(p_\sigma \,\|\, h_{\theta^*, \lambda^*}\big) \leq (O(\eta) + O(\sigma))\log(1/\eta) + O\Big(\tfrac{\varepsilon}{\eta^2}\Big) + O\Big(\tfrac{1}{\eta^2 \ell}\Big),$$

*and the functions $\tilde{p}_{\sigma, X, \eta}$ and $h_{\theta^*, \lambda^*}$ satisfy $\eta \leq \tilde{p}_{\sigma, X, \eta}(x), h_{\theta^*, \lambda^*}(x) \leq M$ on $X$ for some finite $M$.*

The proof proceeds through three stages—mollification, restriction and clipping, and mixture approximation—followed by a change-of-measure decomposition. We develop the proof through the individual results below.

**Remark 1.** *For any bounded continuous test function $f$, we have*

$$\int f(x)p_\sigma(x)\,dx = \sum_i \nu_b(s^i)\,\mathbb{E}[f(s^i + \mathbf{z}_\sigma)], \qquad \mathbf{z}_\sigma \sim \mathcal{N}(0, \sigma^2 I_p).$$

*Since $\mathbf{z}_\sigma \to 0$ in probability, $\mathbb{E}[f(s^i + \mathbf{z}_\sigma)] \to f(s^i)$ by the dominated convergence theorem, implying that $p_\sigma$ converges weakly to $\nu_b$ as $\sigma \downarrow 0$. This step establishes a continuous surrogate distribution for $\nu_b$, ensuring the KL divergence is well-defined.*

For the state space $S$ and given $R > 0$, we define the compact region

$$X_R = \bigcup_{i=1}^n \{\, x : \|x - s^i\| \leq R \,\}, \text{ and the tail mass as } \tau(R, \sigma) = \int_{X_R^c} p_\sigma(x)\,dx. \tag{28}$$

**Lemma 2.** *For every $\delta \in (0, 1)$ and fixed $\sigma > 0$, there exists $R \equiv R(\sigma, \delta)$ such that $\tau(R, \sigma) \leq \delta$. Moreover, for mixtures of Gaussians as considered here, one can choose $R$ so that*

$$\tau(R, \sigma) \leq C\,n\,\exp\Big(-\tfrac{R^2}{2\sigma^2}\Big),$$

*for a constant $C = C(p)$ depending only on the dimension.*

*Proof.* Using the union bound and the Gaussian tail inequality, for each $i$ we have

$$\int_{\|x-s^i\|>R} \mathcal{N}(x; s^i, \sigma^2 I_p)\,dx \leq C(p)\,\exp\Big(-\tfrac{R^2}{2\sigma^2}\Big),$$

where $C(p)$ is a dimension-dependent constant. By summing over $i = 1, \ldots, n$ and weighting by $\nu_b(s^i) \leq 1$, we get

$$\tau(R, \sigma) \leq C(p)\,n\,\exp\Big(-\tfrac{R^2}{2\sigma^2}\Big).$$

Hence, for any $\delta > 0$, it suffices to choose

$$R \geq \sigma\sqrt{2\log\Big(\tfrac{Cn}{\delta}\Big)}$$

to ensure $\tau(R, \sigma) \leq \delta$. $\qquad\square$

Henceforth, we fix a compact set $X = X_R$ with tail mass $\tau = \tau(R, \sigma)$ as small as required. We do not define a zero-outside renormalized density. Such a truncation would make the KL divergence infinite. Instead, we clip the density from below everywhere and then renormalize, which maintains strict positivity on $\mathbb{R}^p$ while preserving the total mass. Note that by construction, $\tilde{p}_{\sigma, X, \eta}$ is a strictly positive density. Further, on $X$ we have $f_\eta(x) = \max\{p_\sigma(x), \eta\} \geq p_\sigma(x)\mathbf{1}_X(x)$, and on $X^c$ we leave the density unchanged, i.e., $f_\eta(x) = p_\sigma(x)$. Thus

$$Z_\eta \;=\; \int_X f_\eta(x)\,dx + \int_{X^c} f_\eta(x)\,dx \;\in\; \left[\; \int_X p_\sigma(x)\,dx, \;\int_X (p_\sigma(x) + \eta)\,dx \right] \;=\; [\,1,\; 1 + \eta|X|\,]. \tag{29}$$

The lower bound uses $f_\eta \geq p_\sigma$, and the upper bound uses $f_\eta \leq p_\sigma + \eta$ on $X$ together with $f_\eta = p_\sigma$ on $X^c$.

**Lemma 3** ($L^1$ perturbation)**.** *There exist universal constants $c_1, c_2 > 0$ such that*

$$\| p_\sigma - \tilde{p}_{\sigma, X, \eta} \|_1 \;\leq\; c_1\tau + c_2\eta|X|.$$

*Proof.* By definition,

$$\|p_\sigma - \tilde{p}_{\sigma, X, \eta}\|_1 = \int_{\mathbb{R}^p} |p_\sigma - \tilde{p}_{\sigma, X, \eta}| = \int_X |p_\sigma - \tilde{p}_{\sigma, X, \eta}| + \int_{X^c} |p_\sigma - \tilde{p}_{\sigma, X, \eta}|.$$

We now bound the integrals explicitly for each region. For all $x \in X^c$ we have $f_\eta(x) = p_\sigma(x)$ and therefore

$$\tilde{p}_{\sigma, X, \eta}(x) = \frac{p_\sigma(x)}{Z_\eta}.$$

Hence

$$\int_{X^c} |p_\sigma - \tilde{p}_{\sigma, X, \eta}| = \int_{X^c} p_\sigma(x)\left|1 - \frac{1}{Z_\eta}\right|dx = \left|1 - \frac{1}{Z_\eta}\right| \int_{X^c} p_\sigma(x)\,dx = \left|1 - \frac{1}{Z_\eta}\right|\tau. \tag{30}$$

From (29), we have $Z_\eta \in [1, 1 + \eta|X|]$, so

$$\left|1 - \frac{1}{Z_\eta}\right| = \frac{|Z_\eta - 1|}{Z_\eta} \leq |Z_\eta - 1| \leq \eta|X|.$$

Thus

$$\int_{X^c} |p_\sigma - \tilde{p}_{\sigma, X, \eta}| \leq \tau\,\eta|X|. \tag{31}$$

Now for $X$, we split $X$ into $A$ and $B$, where $A = X \cap \{p_\sigma \geq \eta\}$, and $B = X \cap \{p_\sigma < \eta\}$.

For $x \in A$, $f_\eta(x) = p_\sigma(x)$, so $\tilde{p}_{\sigma, X, \eta}(x) = p_\sigma(x)/Z_\eta$. Hence

$$|p_\sigma - \tilde{p}_{\sigma, X, \eta}| = p_\sigma(x)\left|1 - \frac{1}{Z_\eta}\right| \leq p_\sigma(x)\,\eta|X|.$$

Integrating over $A$ gives

$$\int_A |p_\sigma - \tilde{p}_{\sigma, X, \eta}| \leq \eta|X| \int_A p_\sigma(x)\,dx \leq \eta|X|.$$

For $x \in B$, $p_\sigma(x) < \eta$ and $f_\eta(x) = \eta$, so

$$\tilde{p}_{\sigma, X, \eta}(x) = \frac{\eta}{Z_\eta}.$$

Thus

$$|p_\sigma(x) - \tilde{p}_{\sigma, X, \eta}(x)| \leq p_\sigma(x) + \frac{\eta}{Z_\eta} \leq \eta + \eta = 2\eta,$$

using $Z_\eta \geq 1$. Hence

$$\int_B |p_\sigma - \tilde{p}_{\sigma, X, \eta}| \leq 2\eta|X|.$$

Finally, by combining the integral bounds on $A$ and $B$, we get

$$\int_X |p_\sigma - \tilde{p}_{\sigma,X,\eta}| \leq \eta|X| + 2\eta|X| = 3\eta|X|. \tag{32}$$

Combining the bounds from $X^c$ ((31)) and $X$ ((32)), we obtain

$$\|p_\sigma - \tilde{p}_{\sigma,X,\eta}\|_1 \leq \tau\,\eta|X| + 3\eta|X| \leq c_1\tau + c_2\eta|X|,$$

for suitable absolute constants $c_1, c_2$. □

**Lemma 4** (KL perturbation)**.** *There exist constants $C_1, C_2 > 0$ (depending on the dimension and the mixture envelope on $X$) such that*

$$\mathcal{D}_{\mathrm{KL}}\big(p_\sigma \,\|\, \tilde{p}_{\sigma,X,\eta}\big) \;\leq\; C_1\eta + C_2\tau\eta.$$

*Proof.* By definition, we have

$$\mathcal{D}_{\mathrm{KL}}\big(p_\sigma \,\|\, \tilde{p}_{\sigma,X,\eta}\big) \;=\; \int p_\sigma \log p_\sigma \;-\; \int p_\sigma \log \tilde{p}_{\sigma,X,\eta}.$$

We split the domain into three parts: (i) $X \cap \{p_\sigma \geq \eta\}$; (ii) $X \cap \{p_\sigma < \eta\}$; (iii) $X^c$ and explicitly develop bounds for $\int p_\sigma \log \tilde{p}_{\sigma,X,\eta}$ for each region as follows:

*(i) Region $X \cap \{p_\sigma \geq \eta\}$.* Here $f_\eta = p_\sigma$, so $\tilde{p}_{\sigma,X,\eta} = p_\sigma/Z_\eta$, and

$$\int_{X \cap \{p_\sigma \geq \eta\}} p_\sigma \log\Big(\frac{p_\sigma}{p_\sigma/Z_\eta}\Big) \;=\; \int_{X \cap \{p_\sigma \geq \eta\}} p_\sigma \log Z_\eta \;\leq\; \log\Big(1 + \eta|X|\Big) \;\leq\; c\eta,$$

for some constant $c > 0$.

*(ii) Region $X \cap \{p_\sigma < \eta\}$.* Here $\tilde{p}_{\sigma,X,\eta} = \eta/Z_\eta$, so,

$$\int_{p_\sigma < \eta} p_\sigma \log\Big(\frac{p_\sigma}{\eta/Z_\eta}\Big) \;=\; \int_{p_\sigma < \eta} p_\sigma \log\Big(\frac{p_\sigma}{\eta}\Big) \;+\; \log Z_\eta \int_{p_\sigma < \eta} p_\sigma \leq c'\eta.$$

for some constant $c' > 0$. This follows since on $\{p_\sigma < \eta\}$, $p_\sigma \log(p_\sigma/\eta) \leq 0$.

*(iii) Region $X^c$.* Here $\tilde{p}_{\sigma,X,\eta} = p_\sigma/Z_\eta$, and,

$$\int_{X^c} p_\sigma \log\Big(\frac{p_\sigma}{p_\sigma/Z_\eta}\Big) \;=\; \int_{X^c} p_\sigma \log Z_\eta \leq c''\tau\eta, \text{ for some constant } c'' > 0. \tag{33}$$

Now collecting the bounds from (i)–(iii) and absorbing constants yields

$$\mathcal{D}_{\mathrm{KL}}\big(p_\sigma \,\|\, \tilde{p}_{\sigma,X,\eta}\big) \;\leq\; C_1\eta + C_2\tau\eta.$$

□

**Proof of Theorem 3:** By Lemma 4.1 of (Zeevi & Meir, 1997), for any $\varepsilon > 0$ there exists an $\ell$-component Gaussian mixture $h_{\theta^*,\lambda^*}$ such that

$$\mathcal{D}_{\mathrm{KL}}\big(\tilde{p}_{\sigma,X,\eta} \,\|\, h_{\theta^*,\lambda^*}\big) \;\leq\; O\Big(\frac{\varepsilon}{\eta^2}\Big) \;+\; O\Big(\frac{1}{\eta^2\ell}\Big), \tag{34}$$

and, on the compact set $X$, the mixture satisfies uniform bounds $\eta \leq \tilde{p}_{\sigma,X,\eta}(x),\; h_{\theta^*,\lambda^*}(x) \leq M,\quad x \in X$, for some finite $M > 0$. Now

$$\mathcal{D}_{\mathrm{KL}}(p_\sigma\|h) = \mathcal{D}_{\mathrm{KL}}(p_\sigma\|f) + \mathcal{D}_{\mathrm{KL}}(f\|h) + (\mathbb{E}_{p_\sigma} - \mathbb{E}_f)\Big[\log\frac{f}{h}\Big], \qquad \text{with } f = \tilde{p}_{\sigma,X,\eta}, \text{ and } h = h_{\theta^*,\lambda^*}. \tag{35}$$

By Lemma 4, we have, for constants $C_1, C_2 > 0$, $\tau = \tau(R, \sigma) < 1$, and $\eta < 1$,

$$\mathcal{D}_{\mathrm{KL}}(p_\sigma \| \tilde{p}_{\sigma,X,\eta}) \leq C_1\eta + C_2\eta\tau \leq C_1\eta + C_2(\eta + \tau)\log(1/\eta) \text{ (Using AM-GM inequality)}$$
$$= (O(\eta) + O(\sigma))\log(1/\eta). \tag{36}$$

The last equality follows since $\tau = O(\sigma)$ by Lemma 2. Now for the last term in (35), we have,

$$(\mathbb{E}_{p_\sigma} - \mathbb{E}_f)\left[\log \frac{f}{h}\right] = \int (p_\sigma(x) - f(x)) \log \frac{f(x)}{h(x)} \, dx,$$

$$\Rightarrow \left| (\mathbb{E}_{p_\sigma} - \mathbb{E}_f)\left[\log \frac{f}{h}\right] \right| \leq \|p_\sigma - f\|_1 \sup_x \left| \log \frac{f(x)}{h(x)} \right|.$$

On $X$, we have $\eta/Z_\eta \leq f(x), h(x) \leq M$ for $Z_\eta \in [1, 1 + \eta|X|]$, so

$$\sup_{x \in X} \left| \log \frac{f(x)}{h(x)} \right| \leq \log\left(\frac{MZ_\eta}{\eta}\right) = O\big(\log(1/\eta)\big),$$

and outside $X$ the contribution can be absorbed into the constants since the tail mass $\tau$ is $O(\sigma)$. Lemma 3 implies

$$\|p_\sigma - f\|_1 \leq c_1\tau + c_2\eta|X| = O(\sigma) + O(\eta),$$

so,

$$\left| (\mathbb{E}_{p_\sigma} - \mathbb{E}_f)\left[\log \tfrac{f}{h}\right] \right| \leq \big(O(\eta) + O(\sigma)\big) \log\big(1/\eta\big), \tag{37}$$

Finally, combining the three bounds (34), (36) and (37) in (35), we obtain

$$\mathcal{D}_{\mathrm{KL}}(p_\sigma \| h_{\theta^*,\lambda^*}) \leq (O(\eta) + O(\sigma))\log(1/\eta) + O\left(\frac{\varepsilon}{\eta^2}\right) + O\left(\frac{1}{\eta^2\ell}\right), \tag{38}$$

which is the bound claimed in Theorem 3. $\qquad\square$

Theorem 3 shows that the mollified stationary distribution $\nu_b^{(\sigma)} = \nu_b * \mathcal{N}(0, \sigma^2 I)$ admits a finite–mixture Gaussian approximation with controlled KL error given by,

$$\mathcal{D}_{\mathrm{KL}}\big(\nu_b^{(\sigma)} \,\|\, h(\cdot; \theta^*, \lambda^*)\big) \leq O(\eta + \sigma)\log(1/\eta) + O(\varepsilon/\eta^2) + O(1/(\eta^2\ell)),$$

where $\eta$ is the clipping level and $\ell$ is the number of mixture components. The first term captures the error introduced by mollification and clipping. Since $(\eta + \sigma)\log(1/\eta) \to 0$ as $\eta, \sigma \to 0$, this contribution can be made arbitrarily small by choosing $\eta$ and $\sigma$ sufficiently small. For any fixed $\eta > 0$, the remaining terms decrease monotonically as $\ell$ increases and the approximation error $\varepsilon$ is reduced (e.g., by refining the mixture class), and the empirical curves in Figures. 2a and 2b illustrate this decay. This guarantees that $h(\cdot; \theta^*, \lambda^*)$ converges to $\nu_b^{(\sigma)}$ in KL, and hence in total variation by Pinsker's inequality. The resulting mixture $h(\cdot; \theta^*, \lambda^*)$ therefore provides a numerically stable surrogate for $\nu_b$ and is required for the distribution-correction step of our algorithm.

We therefore compute the optimal surrogate by directly minimizing the KL divergence to the mollified target $\nu_b^{(\sigma)}$. Specifically, the mixture parameters are obtained by solving the optimization problem:

$$(\theta^*, \lambda^*) = \arg\min_{\bar{\theta} \in \Theta^\ell, \, \bar{\lambda} \in \Delta^\ell} \mathcal{D}_{\mathrm{KL}}\big(\nu_b^{(\sigma)} \,\|\, h(\cdot; \bar{\theta}, \bar{\lambda})\big) = \arg\max_{\bar{\theta}, \bar{\lambda}} \mathbb{E}_{\mathbf{x} \sim \nu_b^{(\sigma)}}[\log h(\mathbf{x}; \bar{\theta}, \bar{\lambda})].$$

The equivalence to maximizing the expected log-likelihood of the mixture under the mollified law follows from the definition of KL divergence. The objective is differentiable and also well-posed because $\nu_b^{(\sigma)}$ is absolutely continuous with respect to the Gaussian mixture model, ensuring the KL divergence is finite and the optimization landscape is regular. The convolution structure further allows us to rewrite the expectation

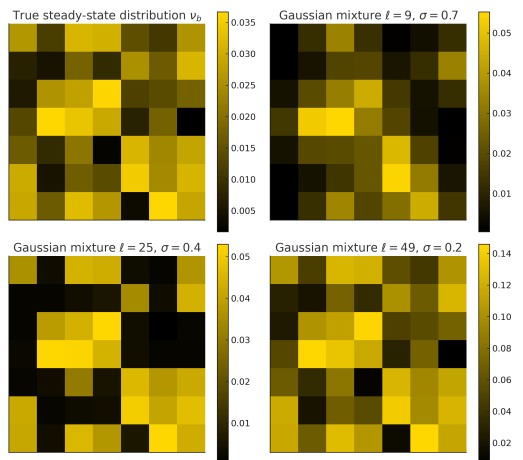
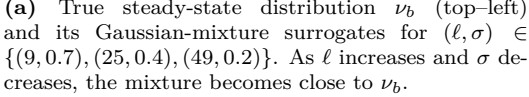

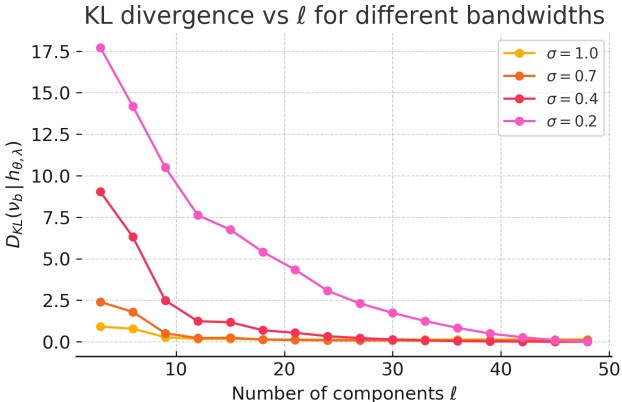

**(a)** True steady-state distribution $\nu_b$ (top–left) and its Gaussian-mixture surrogates for $(\ell, \sigma) \in \{(9, 0.7), (25, 0.4), (49, 0.2)\}$. As $\ell$ increases and $\sigma$ decreases, the mixture becomes close to $\nu_b$.

**(b)** KL–divergence $\mathcal{D}_{\mathrm{KL}}\big(\nu_b \,\|\, h_{\bar{\theta}, \bar{\lambda}}\big)$ versus component count $\ell$ for $\sigma \in \{1.0, 0.7, 0.4, 0.2\}$. Curves decay monotonically, corroborating the $\ell$ and $\sigma$ dependence in Theorem 3.

**Figure 2:** (a) Gaussian-mixture surrogates approaching the mollified stationary distribution; (b) KL decay with increasing component count.

in terms of samples from the original Markov chain. If we draw $\mathbf{s}_t \sim \nu_b$ and $\varepsilon_t \sim \mathcal{N}(0, \sigma^2 I)$ independently and let $\mathbf{x}_t = \mathbf{s}_t + \varepsilon_t$, then $\mathbf{x}_t \sim \nu_b^{(\sigma)}$ and $\mathbb{E}_{\mathbf{x} \sim \nu_b^{(\sigma)}}[\log h(\mathbf{x}; \bar{\theta}, \lambda)] = \mathbb{E}_{\mathbf{s}_t \sim \nu_b, \, \varepsilon_t}\big[\log h(\mathbf{s}_t + \varepsilon_t; \bar{\theta}, \lambda)\big]$, producing unbiased stochastic gradients. Now let

$$F(\bar{\theta}, \bar{\lambda}) = \mathbb{E}_{\nu_b^{(\sigma)}}[\log h(\mathbf{x}; \bar{\theta}, \bar{\lambda})]. \tag{39}$$

Then, the dominated convergence theorem yields $\nabla F(\bar{\theta}, \bar{\lambda}) = \mathbb{E}_{\nu_b^{(\sigma)}}[\nabla \log h(x; \bar{\theta}, \bar{\lambda})]$, so the gradient coincides with the expected score of the mixture.

We now leverage the above results to maintain a running estimate of the approximation of $\nu_b$ through online maximum likelihood estimation, where the parameters $(\bar{\theta}, \bar{\lambda})$ are updated via incremental projected stochastic-gradient ascent with Polyak–Ruppert averaging (Polyak, 1990; Ruppert, 1988). At iteration $t$, the noisy gradient $\nabla \log h(\cdot; \bar{\theta}_t, \bar{\lambda}_t)$ replaces $\nabla F(\bar{\theta}_t, \bar{\lambda}_t)$. The mixture parameters evolve on a possibly different step-size sequence $(\alpha_t)$ than the TD parameters $(\beta_t)$, allowing flexible calibration of the density model relative to the corrected value estimator. This two-timescale structure couples the density estimation with the off-policy correction step and supports convergence of the overall algorithm.

$$\begin{aligned} \begin{bmatrix} \bar{\theta}_{t+1} \\ \bar{\lambda}_{t+1} \end{bmatrix} &= \Pi_{\Theta^\ell \times \Delta^\ell} \left( \begin{bmatrix} \bar{\theta}_t \\ \bar{\lambda}_t \end{bmatrix} + \alpha_t \, \nabla_{\bar{\theta}, \lambda} \log h\big(\tilde{\mathbf{s}}_{t+1}; \bar{\theta}_t, \bar{\lambda}_t\big) \right) \\ \begin{bmatrix} \widehat{\theta}_{t+1} \\ \widehat{\lambda}_{t+1} \end{bmatrix} &= \begin{bmatrix} \widehat{\theta}_t \\ \widehat{\lambda}_t \end{bmatrix} + \frac{1}{t+1} \left( \begin{bmatrix} \bar{\theta}_{t+1} \\ \bar{\lambda}_{t+1} \end{bmatrix} - \begin{bmatrix} \widehat{\theta}_t \\ \widehat{\lambda}_t \end{bmatrix} \right), \end{aligned} \tag{40}$$

where $\tilde{\mathbf{s}}_{t+1} = \mathbf{s}_{t+1} + \varepsilon_{t+1}$, the state sequence $(\mathbf{s}_t)_{t \geq 0}$ is generated by the behaviour policy as a Markov chain on $S$, and $\varepsilon_{t+1} \sim \mathcal{N}(0, \sigma^2 I)$ is an i.i.d. Gaussian perturbation independent of $\mathbf{s}_t$. Under standard ergodicity assumptions, the chain admits a unique stationary distribution $\nu_b$, and the empirical law of $\mathbf{s}_t$ converges to $\nu_b$; in the mean-field and ODE analysis we therefore interpret the expectation of the update in (40) with respect to the mollified stationary law $\nu_b^{(\sigma)}$. Further, $\Pi_{\Theta^\ell \times \Delta^\ell}$ is the projection operator, which projects $\bar{\theta}_t$ onto the constraint set $\Theta^\ell$ and $\bar{\lambda}_t$ onto the probability simplex $\Delta^\ell$. This ensures iterates $[\bar{\theta}_t, \bar{\lambda}_t]^\top$ stay feasible. In the interior of $\Theta^\ell \times \Delta^\ell$, it acts as the identity, and near the boundary, it projects orthogonally onto the boundary. Here $\alpha_t \in (0, 1)$ is the step-size parameter, fixed apriori. Polyak-Ruppert averaging is employed to enhance

stability by maintaining running averages $\widehat{\theta}_t, \widehat{\lambda}_t$ of the stochastic iterates. This averaging scheme reduces the effects of noise in gradient estimates and provides more robust parameter estimates for the distribution correction step.

**Remark 2.** *By leveraging the properties of the NEF, one can obtain a closed form expression for $\nabla \log h(\cdot; \bar{\theta}, \bar{\lambda})$ as follows:*

$$\frac{\partial}{\partial_{\theta_j}} \nabla \log h(s; \bar{\theta}, \bar{\lambda}) = \frac{(\Gamma(x) - \nabla K(\theta_j)) g_{\theta_j}(s)}{h(s; \bar{\theta}, \bar{\lambda})}$$

$$\frac{\partial}{\partial_{\lambda_j}} \log h(s; \bar{\theta}, \bar{\lambda}) = \frac{g_{\theta_j}(s)}{h(s; \bar{\theta}, \bar{\lambda})}$$

In our algorithm, we use a multi-timescale stochastic approximation framework. The stochastic gradient ascent for tracking the steady-state distribution and the TD recursion for the off-policy solution are updated on a faster timescale, while the PR averaging step is updated on a slower one. Specifically, the step-sizes for the gradient ascent ($\alpha_t$) and TD recursion ($\beta_t$) are orders of magnitude larger than the PR-averaging step. This means that while the faster updates capture rapid changes, the slower, smaller step-size of the averaging step smooths out the fluctuations, stabilizing the learning process and reducing noise. This timescale relationship is formally defined as follows:

$$\alpha_t, \beta_t \in (0,1), \quad \sum_{t \geq 0} \alpha_t = \sum_{t \geq 0} \beta_t = \infty, \quad \sum_{t \geq 0} (\alpha_t^2 + \beta_t^2) < \infty, \quad \alpha_t, \beta_t = \Omega(\frac{1}{t+1}). \tag{41}$$

Further, we modify the TD recursion to correct the steady-state bias by incorporating the steady-state distribution correction factor ($\zeta_t$) as follows:

$$\mathbf{x}_{t+1} = \mathbf{x}_t + \beta_t \rho_t \zeta_t \underbrace{\left( \mathbf{r}_{t+1} + \gamma \phi_{t+1}^\top \mathbf{x}_t - \phi_t^\top \mathbf{x}_t \right)}_{\delta_t: \text{ TD error}} \phi_t, \text{ where } \phi_t = \phi(\mathbf{s}_t), \ \rho_t = \frac{\pi(\mathbf{a}_t|\mathbf{s}_t)}{\pi_b(\mathbf{a}_t|\mathbf{s}_t)} \text{ and } \zeta_t = \frac{q(\mathbf{s}_t)}{h(\mathbf{s}_t; \widehat{\theta}_t, \widehat{\lambda}_t)}. \tag{42}$$

Intuitively, $\rho_t$ reweights the TD error $\delta_t$ by how likely the chosen action is under the target *vs.* behaviour policy, while $\zeta_t$ reweights by how likely the state $\mathbf{s}_t$ is under the behaviour's steady-state distribution, *w.r.t.* the desired design distribution $q$. By introducing the correction factor $\zeta_t$, we re-weight updates to emphasize states in accordance with a predefined design distribution $q$. In the ideal case, we set $q(s) = \nu_\pi(s)$ (the target policy's true stationary distribution), but even if $\nu_\pi$ is unknown, we can choose $q(s)$ to be a reasonable proxy. It is a predefined, domain-specific distribution or heuristic approximation, carefully handcrafted to suit the problem context. For instance, in risk-aware or safety-constrained applications, $q$ may emphasize certain critical regions of the state space, while in healthcare, it could overweight underrepresented patient conditions to ensure equitable learning. This correction adjusts state visitation frequencies, ensuring that states infrequently visited by the behaviour policy receive appropriate weight during learning. The pseudocode of our approach is given in Algorithm 1.

**Assumption 4. *Geometric mixing (spectral gap).*** *There exist constants $M > 0$ and $\rho \in (0,1)$ such that $\|P^t(s, \cdot) - \nu_b\|_{\mathrm{TV}} \leq M \rho^t, \forall s \in S, \ t \geq 0$.*

**Assumption 5. *Parameter Space Regularity.*** *The parameter space $\Theta$ is compact with smooth boundary.*

**Assumption 6. *Uniformly bounded score function.*** *There is a constant $G < \infty$ such that $\|\nabla \log h(s; \bar{\theta}, \bar{\lambda})\| \leq G, \forall \bar{\theta} \in \Theta^\ell, \forall \bar{\lambda} \in \Delta^\ell, \ \forall s \in S$.*

**Remark 3.** *For the mixture NEF model $h(s; \bar{\theta}, \bar{\lambda}) = \sum_{j=1}^\ell \lambda_j g_{\theta_j}(s)$, we note that the parameters $v = (\bar{\theta}, \bar{\lambda})$ lie in a compact set (because $\Theta$ is compact and $\bar{\lambda}$ lies in the simplex, which is also compact). Then, for each $s$, $h(s; v)$ is a continuous function of $v$ (as a finite sum of products of continuous functions) and hence attains a minimum and maximum over the compact parameter space. Again, because $g_{\theta_j}(s) > 0$ and $\lambda_j \geq 0$ with $\sum \lambda_j = 1$, we have $h_v(s) \geq \min_j g_{\theta_j}(s) \geq \eta > 0$ and $h_v(s) \leq \max_j g_{\theta_j}(s) \leq M$. Moreover, the same bounds hold uniformly in $s$ because there are finitely many states. Thus, for the mixture model, we also have:*

$$0 < \eta \leq h(s; v) \leq M < \infty, \quad \forall v \in \Theta^\ell \times \Delta^\ell, \ \forall s \in S.$$

---

**Algorithm 1:** Off-policy TD with linear function approximation and distributional correction

---

**Function** `Off-TD-SSBC`$(\pi, \pi_b)$

    **for** *each transition* $(\mathbf{s}_t, \mathbf{a}_t, \mathbf{r}_{t+1}, \mathbf{s}_{t+1})$ **do**

        Calibrate parameters as follows:

$$\begin{bmatrix} \bar{\theta}_{t+1} \\ \bar{\lambda}_{t+1} \end{bmatrix} = \Pi_{\Theta^\ell \times \Delta^\ell} \left( \begin{bmatrix} \bar{\theta}_t \\ \bar{\lambda}_t \end{bmatrix} + \alpha_t \nabla \log h(\tilde{\mathbf{s}}_{t+1}; \bar{\theta}_t, \bar{\lambda}_t) \right), \text{where } \tilde{\mathbf{s}}_{t+1} = \mathbf{s}_{t+1} + \varepsilon_{t+1}, \text{ and } \varepsilon_{t+1} \sim \mathcal{N}(0, \sigma^2 I)$$

$$\begin{bmatrix} \widehat{\theta}_{t+1} \\ \widehat{\lambda}_{t+1} \end{bmatrix} = \begin{bmatrix} \widehat{\theta}_t \\ \widehat{\lambda}_t \end{bmatrix} + \frac{1}{t+1} \left( \begin{bmatrix} \bar{\theta}_{t+1} \\ \bar{\lambda}_{t+1} \end{bmatrix} - \begin{bmatrix} \widehat{\theta}_t \\ \widehat{\lambda}_t \end{bmatrix} \right)$$

$$\mathbf{x}_{t+1} = \mathbf{x}_t + \beta_t \rho_t \zeta_t \left( \mathbf{r}_{t+1} + \gamma \phi_{t+1}^\top \mathbf{x}_t - \phi_t^\top \mathbf{x}_t \right) \phi_t,$$

        where $\rho_t = \dfrac{\pi(\mathbf{a}_t | \mathbf{s}_t)}{\pi_b(\mathbf{a}_t | \mathbf{s}_t)}$ and $\zeta_t = \dfrac{q(\mathbf{s}_t)}{h(\mathbf{s}_t; \widehat{\theta}_t, \widehat{\lambda}_t)}$

---

*This then leads to the boundedness of the score function as previously explained.*

To establish the convergence properties of Algorithm 1, we analyze the stochastic updates of the distribution parameters $\bar{\theta}_t$ and $\bar{\lambda}_t$. Let $\upsilon_t = [\bar{\theta}_t, \lambda_t]^\top$, $h(\cdot; \upsilon_t) = h(\cdot; \bar{\theta}_t, \bar{\lambda}_t)$, $U = \Theta^\ell \times \Delta^\ell$ and $\mathcal{F}_t = \sigma(\theta_k, \lambda_k, \widehat{\theta}_k, \widehat{\lambda}_k, \mathbf{s}_k, \mathbf{a}_k, \mathbf{x}_k, \varepsilon_k, 0 \le k \le t)$ be the natural filtration generated by all variables up to time $t$. Then the update recursion of $[\bar{\theta}_t, \bar{\lambda}_t]^\top$ can be decomposed into a deterministic drift, a martingale noise, and a bias as follows:

$$\begin{aligned} \upsilon_{t+1} &= \Pi_U \left( \upsilon_t + \alpha_t \nabla \log h(\tilde{\mathbf{s}}_{t+1}; \upsilon_t) \right) \\ &= \Pi_U \left( \upsilon_t + \alpha_t \left( \nabla F(\upsilon_t) + \mathbb{M}_{t+1}^\upsilon + b_t^\upsilon \right) \right), \end{aligned}$$

$$\text{where } \mathbb{M}_{t+1}^\upsilon = \nabla \log h(\tilde{\mathbf{s}}_{t+1}; \upsilon_t) - \mathbb{E}\left[ \nabla \log h(\tilde{\mathbf{s}}_{t+1}; \upsilon_t) | \mathcal{F}_t \right], \text{ and } b_t^\upsilon = \mathbb{E}\left[ \nabla \log h(\tilde{\mathbf{s}}_{t+1}; \upsilon_t) | \mathcal{F}_t \right] - \nabla F(\upsilon_t).$$
$$(43)$$

First we establish a fundamental result on the bias term which shows that the bias term is geometrically decaying and therefore summable.

**Lemma 5.** *Let Assumptions 4 and 6 hold. Then the bias term $b_t^\upsilon$ satisfies $\|b_t^\upsilon\| \le GM \rho^t, \forall t \ge 0$ and $\sum_{t=0}^{\infty} \alpha_t \|b_t^\upsilon\| < \infty.$*

*Proof.* By conditioning first on $\mathbf{s}_{t+1}$ and then on the Gaussian perturbation, we get

$$\mathbb{E}[\nabla \log h(\tilde{\mathbf{s}}_{t+1}; \upsilon_t) \mid \mathcal{F}_t] = \sum_{s' \in S} \mathbb{P}_{\pi_b}(\mathbf{s}_{t+1} = s') g_{\upsilon_t}(s'), \tag{44}$$

where we define the mollified score $g_\upsilon(s') = \mathbb{E}_{\varepsilon \sim \mathcal{N}(0, \sigma^2 I)}[\nabla_\upsilon \log h(s' + \varepsilon; \upsilon)]$. Under Assumption 6, the (unperturbed) score is uniformly bounded, $\|\nabla_\upsilon \log h(x; \upsilon)\| \le G$ for all $x$, and hence $\|g_\upsilon(s')\| \le G$ for all $s' \in S$ and all $\upsilon$. The gradient of the mollified objective $F(\upsilon_t)$ is

$$\nabla F(\upsilon_t) = \mathbb{E}_{\mathbf{s} \sim \nu_b, \ \varepsilon \sim \mathcal{N}(0, \sigma^2 I)}\left[ \log h(\mathbf{s} + \varepsilon; \upsilon) \right], = \sum_{s' \in S} \nu_b(s') g_{\upsilon_t}(s'). \tag{45}$$

Subtracting (45) from (44) and let $\mu_t(s) = \mathbb{P}_{\pi_b}(\mathbf{s}_t = s)$, we get,

$$b_t^\upsilon = \sum_{s' \in S} \left( \mu_t(s') - \nu_b(s') \right) g_{\upsilon_t}(s'). \tag{46}$$

Using the earlier uniform bound $\|g_{\upsilon_t}(s')\| \le G$ and the identity $\|\mu_t - \nu_b\|_1 = 2\|\mu_t - \nu_b\|_{\text{TV}}$, we obtain

$$\|b_t^\upsilon\| \le G\|\mu_t - \nu_b\|_1 = 2G \|\mu_t - \nu_b\|_{\text{TV}}.$$

From Assumption 4 (geometric mixing of the behaviour chain), we obtain

$$\|b_t^v\| \leq 2GM\rho^t, \quad \text{where } M > 0 \text{ and } 0 < \rho < 1.$$

Further,

$$\sum_{t=0}^{\infty} \alpha_t \|b_t^v\| \leq 2GM \sum_{t=0}^{\infty} \alpha_t \rho^t.$$

Since the step-sizes satisfy $\alpha_t \to 0$, $\sum_t \alpha_t = \infty$, and $\sum_t \alpha_t^2 < \infty$ (from (41)), the weighted geometric series $\sum_t \alpha_t \rho^t$ converges. Thus $\sum_t \alpha_t \|b_t^v\| < \infty$. □

The following theorem establishes that the sequence $\{[\bar{\theta}_t, \bar{\lambda}_t]^\top\}$ converges to Karush-Kuhn-Tucker (KKT) points—first-order optimality conditions where the gradient aligns with the normal cone of $\Theta^\ell \times \Delta^\ell$. This guarantees the learned mixture distribution $h(\cdot; \bar{\theta}, \bar{\lambda})$ converges to a stationary point of the KL-divergence minimization problem.

**Theorem 4** (Convergence of Distribution Approximation). *Let the step-size $\{\alpha_t\}$ satisfy (41). Let Assumptions 4-6 hold. Then the sequence $\{[\bar{\theta}, \bar{\lambda}]^\top\}$ converges almost surely to the set of KKT points:*

$$\{v = [\bar{\theta}, \bar{\lambda}]^\top \in \Theta^\ell \times \Delta^\ell : -\nabla F(v) \in N_U(v)\},$$

*where $N_U(v)$ denotes the normal cone to $Theta^\ell \times \Delta^\ell$ at $v$, defined as:*

$$N_U(v) = \left\{ d \in \mathbb{R}^{dim(\Theta^\ell)+\ell} : \langle d, u - v \rangle \leq 0, \ \forall u \in U \right\}.$$

*Proof.* Let $g_t = \nabla F(v_t) + \mathbb{M}_{t+1}^v + b_t^v$. Then,

$$
\begin{aligned}
v_{t+1} &= \Pi_U(v_t + \alpha_t g_t), \\
&= v_t + \alpha_t \Gamma_U(g_t) + \Pi_U(v_t + \alpha_t g_t) - v_t - \alpha_t \Gamma_U(g_t) \\
&= v_t + \alpha_t \left( \Gamma_U(g_t) + \frac{\Pi_U(v_t + \alpha_t g_t) - v_t}{\alpha_t} - \Gamma_U(g_t) \right) \\
&= v_t + \alpha_t \left( \Gamma_U(g_t) + o(\alpha_t) \right).
\end{aligned}
\tag{47}
$$

The last equality follows since

$$\lim_{\varepsilon \to 0} \frac{\Pi_U(v_t + \varepsilon g_t) - v_t}{\varepsilon} = \underbrace{\Pi_{T_U(v_t)}(g_t)}_{\Gamma_U(g_t)}, \tag{48}$$

where

$$T_U(v) = \overline{\left\{ u \in \mathbb{R}^{\dim(\Theta^\ell)+\ell} : v + \tau u \in U \text{ for some } \tau > 0 \right\}}.$$

In the interior of $U$, $T_U(v_t) = \mathbb{R}^{\dim(\Theta^\ell)+\ell}$ (unconstrained) and near the boundary of $U$, $T_U(v_t) = \{u \in \mathbb{R}^{\dim(\Theta^\ell)+\ell} \mid u$ points into $U\}$. Thus, $\Gamma_U(g_t)$ is the directional derivative of the projection operator $\Pi_U$ at point $v_t$ in the direction $g_t$, which is equivalent to the projection of $g_t$ onto the tangent cone of $U$ at $v_t$ (Rockafellar, 2015). Intuitively, it captures the "feasible component" of $g_t$ that aligns with the constraints of $U$.

For the noise $\mathbb{M}_{t+1}^v$, $\mathbb{E}[\mathbb{M}_{t+1}^v \mid \mathcal{F}_t] = 0$ (by definition). Further, using the triangle inequality,

$$
\begin{aligned}
\|\mathbb{M}_{t+1}^v\| &= \left\| \nabla \log h(\tilde{\mathbf{s}}_{t+1}; v_t) - \mathbb{E}[\nabla \log h(\tilde{\mathbf{s}}_{t+1}; v_t) \mid \mathcal{F}_t] \right\| \\
&\leq G + G = 2G \quad a.s.
\end{aligned}
$$

By squaring and taking conditional expectation, we get $\mathbb{E}[\|\mathbb{M}_{t+1}^v\|^2 \mid \mathcal{F}_t] \leq (2G)^2 = 4G^2$. Thus $\{\mathbb{M}_t^v\}$ is a square-integrable martingale-difference sequence. Now, consider $S_t = \sum_{k=0}^{t-1} \alpha_k \mathbb{M}_{k+1}^v$. Note that

$$\sum_{t \geq 0} \mathbb{E}\left[\|S_{t+1} - S_t\|^2 | \mathcal{F}_t\right] = \sum_{t \geq 0} \alpha_t^2 \mathbb{E}\left[\|\mathbb{M}_{t+1}^v\|^2 | \mathcal{F}_t\right] < 4G^2 \sum_{t \geq 0} \alpha_t^2 < \infty. \tag{49}$$

By martingale convergence theorem, it follows that $S_t$ converges, *i.e.*, $\sum_{t=0}^{\infty} \alpha_t \mathbb{M}_{t+1}^v < \infty \quad a.s.$

Now rearranging (47), we get

$$v_{t+1} = v_t + \alpha_t\big(\Gamma_U(\nabla F(v_t)) + \underbrace{\Gamma_U(g_t) - \Gamma_U(\nabla F(v_t))}_{\xi_t} + o(\alpha_t)\big) \tag{50}$$

Using the non-expansive property of $\Gamma_U$, we have

$$\begin{aligned}
\|\xi_t\| &= \|\Gamma_U(g_t) - \Gamma_U(\nabla F(v_t))\| \\
&\leq \|g_t - \nabla F(v_t)\| = \|\mathbb{M}_{t+1}^v + b_t^v\| \\
&\leq \|\mathbb{M}_{t+1}^v\| + \|b_t^v\|.
\end{aligned} \tag{51}$$

Hence,

$$\sum_t \alpha_t \|\xi_t\| \leq \sum_t \alpha_t \|\mathbb{M}_{t+1}^v\| + \sum_t \alpha_t \|b_t^v\| < \infty \quad a.s. \tag{52}$$

Therefore by (Borkar, 2008), it follows that $\{v_t\}$ asymptotically tracks the ODE

$$\dot{v} = \Gamma_U(\nabla F(v)). \tag{53}$$

However, because $F$ is smooth and the constraint set $U$ is convex, the above differential equation is well-defined and corresponds to the projected gradient ascent. By the theory of stochastic approximation (see (Borkar, 2008)), the sequence $\{v_t\}$ converges to a (possibly sample path dependent) internally chain transitive invariant set of the above ODE. Since $F$ is $C^1$,

$$\frac{d}{dt} F\big(v(t)\big) = \big\langle \nabla F\big(v(t)\big), \dot{v}(t) \big\rangle = \big\langle \nabla F(v), \Gamma_U\big[\nabla F(v)\big] \big\rangle.$$

Apply Moreau's decomposition to obtain $\nabla F(v) = \Pi_{T_U(v)}[\nabla F(v)] + \Pi_{N_U(v)}[\nabla F(v)]$ and $\langle \Pi_{T_U(v)}[\nabla F(v)], \Pi_{N_U(v)}[\nabla F(v)] \rangle = 0$. Then,

$$\begin{aligned}
\langle \nabla F(v), \Gamma_U(\nabla F(v)) \rangle &= \langle \Pi_{T_U(v)}[\nabla F(v)] + \Pi_{N_U(v)}[\nabla F(v)], \Gamma_U(\nabla F(v)) \rangle \\
&= \langle \Gamma_U(\nabla F(v)), \Gamma_U(\nabla F(v)) \rangle \\
&= \big\|\Gamma_U(\nabla F(v))\big\|^2 \geq 0.
\end{aligned} \tag{54}$$

Hence

$$\frac{d}{dt} F\big(v(t)\big) = \big\|\Gamma_U\big[\nabla F\big(v(t)\big)\big]\big\|^2 \geq 0 \tag{55}$$

with equality *iff* $\Gamma_U[\nabla F(v(t))] = 0$. Therefore, the invariant set of the above ODE is the stationary (equilibrium) set:

$$\big\{v \in U : \Gamma_U\big[\nabla F(v)\big] = 0\big\} = \big\{v \in U : -\nabla F(v) \in N_U(v)\big\},$$

which are the Karush-Kuhn-Tucker (KKT) points. The last equality follows again by Moreau's decomposition of $\nabla F(v)$. □

Having established the almost sure convergence of the distribution parameters $\{v_t\}$ to $v^*$ in Theorem 4, we now analyze the temporal difference learning dynamics given by (42). Prior to this, observe that the

unilateral timescale separation between the faster distribution estimation updates ($\upsilon_t = [\bar\theta_t, \bar\lambda_t]^\top$) and slower Polyak-Ruppert averaging ($\hat\upsilon_t = [\hat\theta_t, \hat\lambda_t]^\top$) ensures that $\hat\upsilon_t \to \upsilon^*$ asymptotically. This justifies replacing the time-varying $\zeta_t = q(\mathbf{s}_t)/h(\mathbf{s}_t; \hat\upsilon_t)$ in the TD update (42) with its steady-state counterpart $\zeta_t^* = q(\mathbf{s}_t)/h(\mathbf{s}_t; \upsilon^*)$. The substitution decouples the distribution approximation error from the value estimation error, permitting the simplified TD recursion (See Chapter 6 of (Borkar, 2008)). We need the following assumption on the design distribution $q$:

**Assumption 7.** *The design distribution $q$ is strictly positive over all states: $q(s) > 0$ for all $s \in S$.*

Now, we rewrite $\mathbf{x}_t$ update as follows (we let $g_t^x = \rho_t\,\zeta_t^*\big(\mathbf{r}_{t+1} + \gamma\phi_{t+1}^\top x - \phi_t^\top x\big)\phi_t$ and $h^*(\cdot) = h(\cdot; \upsilon^*)$):

$$\mathbf{x}_{t+1} = \mathbf{x}_t + \beta_t\rho_t\zeta_t^*\big(\mathbf{r}_{t+1} + \gamma\phi_{t+1}^\top\mathbf{x}_t - \phi_t^\top\mathbf{x}_t\big)\phi_t, \ \text{where} \ \rho_t = \frac{\pi(\mathbf{a}_t|\mathbf{s}_t)}{\pi_b(\mathbf{a}_t|\mathbf{s}_t)} \ \text{and} \ \zeta_t^* = \frac{q(\mathbf{s}_t)}{h^*(\mathbf{s}_t)}$$

$$= \mathbf{x}_t + \beta_t\left(b_t^x + G(\mathbf{x}_t) + \mathbb{M}_{t+1}^x\right). \tag{56}$$

Here,

$$\begin{aligned}
G(x) = \mathbb{E}[g_t^x] &= \mathbb{E}\Big[\rho_t\,\zeta_t^*\big(\mathbf{r}_{t+1} + \gamma\phi_{t+1}^\top x - \phi_t^\top x\big)\phi_t\Big] \\
&= \mathbb{E}_\mathbf{s}\Big[\frac{q(\mathbf{s})}{h^\star(\mathbf{s})}\mathbb{E}_a\big[\rho_t\big(\mathbf{r}_{t+1} + \gamma\phi_{t+1}^\top x - \phi_t^\top x\big)\phi_t \mid \mathbf{s}\big]\Big] \\
&= \mathbb{E}_\mathbf{s}\Big[\frac{q(\mathbf{s})}{h^\star(\mathbf{s})}\big(R_\pi(\mathbf{s}) + \gamma\,(P_\pi\Phi x)(\mathbf{s}) - (\Phi x)(\mathbf{s})\big)\phi(\mathbf{s})\Big] \\
&= \Phi^\top\Xi_{\nu_b}\,\Xi_{h^\star}^{-1}\,\Xi_q\left(R_\pi + (\gamma\,P_\pi - I)\Phi x\right) \\
&= \underbrace{\Phi^\top\Xi_{\nu_b}\,\Xi_{h^\star}^{-1}\,\Xi_q\,(\gamma\,P_\pi - I)\Phi}_{\Lambda_c}\,x + \underbrace{\Phi^\top\Xi_{\nu_b}\,\Xi_{h^\star}^{-1}\,\Xi_q R_\pi}_{\xi}. \tag{57}
\end{aligned}$$

Also,

$$\mathbb{M}_{t+1}^x = g_t - \mathbb{E}\big[g_t \mid \mathcal{F}_t\big], \ \text{and} \ b_t^x = \mathbb{E}\big[g_t \mid \mathcal{F}_t\big] - h(\mathbf{x}_t), \ \text{where} \ g_t = \rho_t\,\zeta_t^*\big(\mathbf{r}_{t+1} + \gamma\phi_{t+1}^\top\mathbf{x}_t - \phi_t^\top\mathbf{x}_t\big)\phi_t. \tag{58}$$

Further, note that since we have finite state and action spaces $|\mathbf{r}_t| \le R_\infty$, $\|\phi(s)\| \le \Phi_\infty$, and $0 \le \rho_t \le \rho_\infty$, $0 < \zeta_t \le \zeta_\infty$.

We first bound the TD error as follows:

$$\begin{aligned}
|\delta_t| &= \big|\mathbf{r}_{t+1} + \gamma\phi_{t+1}^\top\mathbf{x}_t - \phi_t^\top\mathbf{x}_t\big| \\
&\le |\mathbf{r}_{t+1}| + \gamma|\phi_{t+1}^\top\mathbf{x}_t| + |\phi_t^\top\mathbf{x}_t| \\
&\le R_\infty + \gamma\Phi_\infty\|x_t\| + \Phi_\infty\|\mathbf{x}_t\| = R_\infty + (1+\gamma)\Phi_\infty\|\mathbf{x}_t\|
\end{aligned}$$

Then the update term $g_t$ satisfies

$$\|g_t\| = |\rho_t\zeta_t\delta_t| \cdot \|\phi_t\| \le \rho_\infty\zeta_\infty\left(R_\infty + (1+\gamma)\Phi_\infty\|x_t\|\right) \cdot \Phi_\infty \le C_1 + C_2\|\mathbf{x}_t\|,$$

where $C_1 = \rho_\infty\zeta_\infty R_\infty\Phi_\infty$, and $C_2 = \rho_\infty\zeta_\infty(1+\gamma)\Phi_\infty^2$. Now,

$$\begin{aligned}
\|\mathbb{M}_t^x\| &= \|g_t - \mathbb{E}[g_t \mid \mathcal{F}_t]\| \\
&\le \|g_t\| + \|\mathbb{E}[g_t \mid \mathcal{F}_t]\| \le 2\sup\|g_t\| \\
&\le 2(C_1 + C_2\|\mathbf{x}_t\|) \le \widetilde{C}_1(1 + \|\mathbf{x}_t\|).
\end{aligned}$$

Further, using $\|a - b\|^2 \le 2\|a\|^2 + 2\|b\|^2$, we get,

$$\begin{aligned}
\mathbb{E}\big[\|\mathbb{M}_{t+1}^x\|^2 \mid \mathcal{F}_t\big] &= \mathbb{E}\Big[\big\|g_t - \mathbb{E}[g_t \mid \mathcal{F}_t]\big\|^2 \mid \mathcal{F}_t\Big] \\
&\le 2\,\mathbb{E}\big[\|g_t\|^2 \mid \mathcal{F}_t\big] + 2\,\big\|\mathbb{E}[g_t \mid \mathcal{F}_t]\big\|^2 \\
&\le 4\,\mathbb{E}\big[\|g_t\|^2 \mid \mathcal{F}_t\big] \le 4\big(C_1 + C_2\|\mathbf{x}_t\|\big)^2 \\
&\le 8C_1^2 + 8C_2^2\|\mathbf{x}_t\|^2. \tag{59}
\end{aligned}$$

Now we write $b_t^x = \mathbb{E}[g_t \mid \mathcal{F}_t] - G(\mathbf{x}_t) = \mathbb{E}[g_t|\mathbf{s}_t, \mathbf{x}_t] - \mathbb{E}_{\nu_b}[g_t]$.

$$\text{Thus, } \|b_t^x\| = \|\mathbb{E}[g_t|\mathbf{s}_t, \mathbf{x}_t] - \mathbb{E}_{\nu_b}[g_t]\|$$

$$\leq \sum_{s \in S} |P_{\pi_b}(\mathbf{s}_t, s) - \nu_b(s)| \cdot \|\mathbb{E}[g_t \mid \mathbf{s}_t, \mathbf{x}_t]\|$$

$$\leq \sup_s \|\mathbb{E}[g_t \mid s, \mathbf{x}_t]\| \cdot \|P_{\pi_b}(\mathbf{s}_t, \cdot) - \nu_b\|_1$$

$$= 2 \sup_s \|\mathbb{E}[g_t \mid s, \mathbf{x}_t]\| \cdot \|P_{\pi_b}(\mathbf{s}_t, \cdot) - \nu_b\|_{TV}$$

$$\leq 2M\rho^t(C_1 + C_2\|\mathbf{x}_t\|) \leq \widetilde{C}_2 \rho^t(1 + \|\mathbf{x}_t\|). \tag{60}$$

To establish convergence of the sequence $\{\mathbf{x}_t\}$, we must first ensure the iterates remain stochastically bounded. While classical stochastic approximation theory (Borkar, 2008) often assumes almost sure boundedness, we prove the following weaker but sufficient condition for our setting.

**Lemma 6.** *The iterates $\mathbf{x}_t$ satisfies $\sup_t \mathbb{E}\left[\|\mathbf{x}_t\|^2\right] < \infty$.*

*Proof.* Note $G$ satisfes the drift inequality

$$\mathbf{x}^\top G(\mathbf{x}) \leq -c\|\mathbf{x}\|^2 + d, \qquad \forall \mathbf{x} \in \mathbb{R}^k, \tag{61}$$

where $c = \frac{1}{2}\lambda_{\min}(-\Lambda_c)$ and $d = \|\xi\|^2/(2\lambda_{\min}(-\Lambda_c)))$.

Using $\mathbb{E}[\mathbb{M}_{t+1}^x \mid \mathcal{F}_t] = 0$ and expanding the square,

$$\mathbb{E}[V_{t+1} \mid \mathcal{F}_t] = \|\mathbf{x}_t\|^2 + 2\alpha_t \mathbf{x}_t^\top \left(G(\mathbf{x}_t) + b_t^x\right) + \alpha_t^2\left(\|G(\mathbf{x}_t) + b_t^x\|^2 + \mathbb{E}[\|\mathbb{M}_{t+1}^x\|^2 \mid \mathcal{F}_t]\right).$$

Apply (60), (59), (61), and Young's inequality $2\mathbf{x}_t^\top b_t^x \leq c\|\mathbf{x}_t\|^2 + c^{-1}\|b_t^x\|^2$, and the bound $\|G(\mathbf{x}_t)\|^2 \leq 2\|\Lambda_c\|^2\|\mathbf{x}_t\|^2 + 2\|\xi\|^2$, to obtain

$$\mathbb{E}[V_{t+1}] \leq \left(1 - (2c - c)\alpha_t + L_2\,\alpha_t^2\right)\mathbb{E}[V_t] + 2d\,\alpha_t + L_0\,\alpha_t^2 + \left(c^{-1}\alpha_t + 2\alpha_t^2\right)8M^2\rho^{2t}\left(C_2^2\,\mathbb{E}[V_t] + C_1^2\right), \tag{62}$$

where $L_2 = 4\|\Lambda_c\|^2 + 8C_2^2, \quad L_0 = 4\|\xi\|^2 + 8C_1^2$.

Further, by rearranging, we get,

$$\mathbb{E}[V_{t+1}] \leq \left(1 - c\beta_t + \underbrace{L_2\beta_t^2 + 8M^2C_2^2\,(c^{-1}\beta_t + 2\beta_t^2)\rho^{2t}}_{=G_t}\right)\mathbb{E}[V_t] + 2d\,\beta_t + L_0\,\beta_t^2 + \underbrace{8M^2C_1^2\,(c^{-1}\beta_t + 2\beta_t^2)\rho^{2t}}_{=e_t}. \tag{63}$$

Since $\rho^{2t} \to 0$ geometrically and $\beta_t \to 0$, the perturbation terms $G_t, e_t$ vanish; for all large $t$ one can ensure $G_t \leq \frac{c}{2}\beta_t$, leading to the following stochastic approximation form

$$\mathbb{E}[V_{t+1}] \leq \left(1 - \tfrac{c}{2}\beta_t\right)\mathbb{E}[V_t] + 2d\,\beta_t + c'\beta_t^2. \tag{64}$$

Now using mathematical induction, we will show that $\sup_t \mathbb{E}[V_t] < \infty$. For $t = 0$, $\mathbb{E}[V_0] = \mathbb{E}[\|\mathbf{x}_0\|^2]$ is finite since $\mathbf{x}_0$ is initialized with finite variance (base case). Now assume $\mathbb{E}[V_t] \leq K$ for some constant $K$ and all $t \leq T$, where

$$K = \max\left(\mathbb{E}[\|\mathbf{x}_0\|^2], \frac{4d}{c} + \frac{2}{c}\sup_{t \geq 0} c'\beta_t\right).$$

Then,

$$\mathbb{E}[V_{T+1}] \leq \left(1 - \frac{c}{2}\beta_T\right)K + 2d\beta_T + c'\beta_T^2$$

$$\leq K + \beta_T\left(-\frac{c}{2}K + 2d\right) + c'\beta_T^2. \tag{65}$$

Since $K \geq \dfrac{4d}{c} + \dfrac{2}{c} \sup\limits_{t \geq 0} c' \beta_t$, we have

$$-\frac{c}{2}K + 2d \leq -\frac{c}{2}\left(\frac{4d}{c} + \frac{2}{c}\sup_{t \geq 0} c'\beta_t\right) + 2d$$
$$= -c' \sup_{t \geq 0} \beta_t \leq -c'\beta_T < 0.$$

Thus from (65), we get

$$\mathbb{E}[V_{T+1}] \leq K - c'\alpha_T^2 + c'\alpha_T^2 \leq K.$$

By induction, $\mathbb{E}[V_t] \leq K$ for all but a finite number of $t$. $\qquad\square$

To establish the convergence of $\mathbf{x}_t$, we must show that the bias and noise terms are manageable. Specifically, the next lemma establishes that the series formed by the weighted bias and martingale noise terms converge almost surely.

**Lemma 7.** *For the martingale noise $\mathbb{M}_t^x$ and the bias $b_t^x$, we have*

$$\mathbb{P}\left(\sum_t \beta_t \mathbb{M}_{t+1}^x < \infty, \quad \sum_t \beta_t b_t^x < \infty\right) = 1.$$

*Proof.* From (59) and using tower property, we have

$$\mathbb{E}[\|\mathbb{M}_{t+1}^x\|^2 \mid \mathcal{F}_t] \leq 8C_1^2 + 8C_2^2\|\mathbf{x}_t\|^2$$
$$\Rightarrow \mathbb{E}[\|\mathbb{M}_{t+1}^x\|^2] \leq 8C_1^2 + 8C_2^2\mathbb{E}[\|\mathbf{x}_t\|^2]. \tag{66}$$

Hence, $\mathbb{M}_{t+1}^x$ is square-integrable. Now, by the convergence theorem for square-integrable martingale (for vector-valued martingales), it is enough to show that

$$\sum_t \mathbb{E}[\|\beta_t \mathbb{M}_{k+1}^x\|^2 \mid \mathcal{F}_t] < \infty \quad a.s.$$

Thus it is enough to show that

$$\mathbb{E}\left[\sum_t \mathbb{E}[\|\beta_t \mathbb{M}_{t+1}^x\|^2 \mid \mathcal{F}_t]\right] < \infty.$$

Therefore, by monotone convergence theorem, we get

$$\mathbb{E}\left[\sum_t \mathbb{E}[\|\beta_t \mathbb{M}_{t+1}^x\|^2 \mid \mathcal{F}_t]\right] = \sum_t \mathbb{E}\left[\mathbb{E}[\|\beta_t \mathbb{M}_{t+1}^x\|^2 \mid \mathcal{F}_t]\right]$$
$$\leq \sum_t \beta_t^2 \left(8C_1^2 + 8C_2^2 \mathbb{E}[\|\mathbf{x}_t\|^2]\right)$$
$$\leq \sum_t \beta_t^2 \left(8C_1^2 + 8C_2^2 \sup_t \mathbb{E}[\|\mathbf{x}_t\|^2]\right) < \infty.$$

The last inequality follows from Lemma 6 and $\sum_t \beta_t^2 < \infty$. This implies that $\mathbb{P}\left(\sum_t \beta_t \mathbb{M}_{t+1}^x < \infty\right) = 1$. Now for $b_t^x$, it follows from (60),

$$\mathbb{E}\left[\sum_t \beta_t \|b_t^x\|\right] = \sum_t \beta_t \mathbb{E}[b_t^x]$$
$$\leq \sum_t 2M\beta_t \rho^t (C_1 + C_2 \mathbb{E}[\|\mathbf{x}_t\|])$$
$$\leq \sum_t 2M\beta_t \rho^t (C_1 + C_2 \sqrt{\mathbb{E}[\|\mathbf{x}_t\|^2]})$$
$$\leq \sum_t 2M\beta_t \rho^t (C_1 + C_2 \sup_t \sqrt{\mathbb{E}[\|\mathbf{x}_t\|^2]}) < \infty.$$

The last inequality follows again from Lemma 6, $\beta_t \to 0$ and $\rho \in (0,1)$. Hence, $\mathbb{P}\left(\sum_t \beta_t b_{t+1}^x < \infty\right) = 1$. $\square$

Having established the stochastic boundedness of the iterates $\mathbf{x}_t$ and the almost sure summability of the martingale noise $\sum_t \alpha_t \mathbb{M}_{t+1}^x$ and bias terms $\sum_t \alpha_t b_t^x$, we now prove almost sure convergence of the sequence $\{\mathbf{x}_t\}$.

**Theorem 5** (Convergence of the TD Iterates). *Let Assumptions 1-7 hold. Also, assume that the matrix $\Lambda_c = \Phi^\top \Xi_{\nu_b} \Xi_{h^*}^{-1} \Xi_q (\gamma P_\pi - I)\Phi$ is **Hurwitz** (all eigenvalues have strictly negative real parts) and **diagonalizable**. Then the sequence $\{\mathbf{x}_t\}$ converges almost surely to the unique solution $\mathbf{x}^* = \mathbf{x}_{\mathrm{TD}}^c$ satisfying:*

$$\Phi^\top \Xi_{\nu_b} \Xi_{h^*}^{-1} \Xi_q (I - \gamma P_\pi)\Phi \mathbf{x}^* = \Phi^\top \Xi_{\nu_b} \Xi_{h^*}^{-1} \Xi_q R_\pi$$

*Proof.* Rearranging recursion (56) of $\mathbf{x}_t$ as follows:

$$\mathbf{x}_{t+1} - \mathbf{x}^* = (\mathbf{x}_t - \mathbf{x}^*) + \beta_t \Lambda_c (\mathbf{x}_t - \mathbf{x}^*) + \alpha_t (\Lambda_c \mathbf{x}^* + \xi) + \beta_t (b_t^x + \mathbb{M}_{t+1}^x).$$

But note that $\Lambda_c \mathbf{x}^* + \xi = 0$ by definition of $\mathbf{x}^*$. So,

$$\mathbf{x}_{t+1} - \mathbf{x}^* = (\mathbf{x}_t - \mathbf{x}^*) + \beta_t \Lambda_c (\mathbf{x}_t - \mathbf{x}^*) + \beta_t (b_t^x + \mathbb{M}_{t+1}^x).$$

Let $e_t = \mathbf{x}_t - \mathbf{x}^*$. Then,

$$e_{t+1} = (I + \beta_t \Lambda_c)e_t + \beta_t \eta_t, \text{ where } \eta_t = b_t^x + \mathbb{M}_{t+1}^x.$$

We know that $\sum_t \beta_t \|\eta_t\| < \infty$ *a.s.* by Lemma 7 (since both $b_t^x$ and $\mathbb{M}_{t+1}^x$ are summable in absolute value *a.s.*). Now, because $\Lambda_c$ is negative definite, the matrix $I + \beta_t \Lambda_c$ has eigenvalues in $(0,1)$ for small $\beta_t$. After unraveling the above recursion, we obtain

$$e_{t+1} = \left(\prod_{k=0}^t (I + \beta_k \Lambda_c)\right) y_0 + \sum_{k=0}^t \beta_k \eta_k \left(\prod_{j=k+1}^t (I + \beta_j \Lambda_c)\right).$$

Let

$$Q(t,k) = \prod_{j=k}^{t-1}(I + \beta_j \Lambda_c) \quad \text{for } t > k, \quad Q(k,k) = I. \tag{67}$$

Then,

$$e_{t+1} = Q(t+1,0)e_0 + \sum_{k=0}^t \beta_k Q(t+1, k+1)\eta_k. \tag{68}$$

Now, since $\Lambda_c$ is negative definite, let $\lambda_{\min} > 0$ be such that the real parts of the eigenvalues of $\Lambda_c$ are less than or equal to $-\lambda_{\min}$. Then, there exists a constant $C > 0$ and $\beta > 0$ such that,

$$\|Q(t,k)\| \le C \exp\left(-\beta \sum_{j=k}^{t-1} \alpha_j\right). \tag{69}$$

Since $\Lambda_c$ is diagonalizable, let $\Lambda_c = PDP^{-1}$ where $D = \mathrm{diag}(\lambda_1, \ldots, \lambda_d)$ is diagonal. Then,

$$Q(t,k) = \prod_{j=k}^{t-1}(I + \beta_j \Lambda_c) = P\left(\prod_{j=k}^{t-1}(I + \beta_j D)\right)P^{-1}$$

$$= P\left(\prod_{j=k}^{t-1} D_j\right)P^{-1}$$

where $D_j = I + \beta_j D = \operatorname{diag}(1 + \alpha_j \lambda_1, \ldots, 1 + \beta_j \lambda_d)$. The norm satisfies the following:

$$\|Q(t,k)\| \leq \|P\| \cdot \|P^{-1}\| \cdot \left\| \prod_{j=k}^{t-1} D_j \right\| \tag{70}$$

The diagonal matrix norm is given by:

$$\left\| \prod_{j=k}^{t-1} D_j \right\| = \max_{1 \leq i \leq d} \left| \prod_{j=k}^{t-1} (1 + \beta_j \lambda_i) \right|$$

Now we establish a uniform bound for each eigenvalue product $\prod_{j=k}^{t-1}(1 + \beta_j \lambda_i)$. For any $\epsilon > 0$, there exists $\beta_0 > 0$ such that for $0 \leq \beta_j \leq \beta_0$:

$$|1 + \beta_j \lambda_i| \leq e^{\beta_j \operatorname{Re}(\lambda_i) + \epsilon \beta_j} \tag{71}$$

This follows from the logarithm expansion:

$$\log(1 + \beta_j \lambda_i) = \beta_j \lambda_i - \frac{(\beta_j \lambda_i)^2}{2} + \cdots$$
$$= \beta_j \operatorname{Re}(\lambda_i) + i \beta_j \operatorname{Im}(\lambda_i) + O(\beta_j^2)$$

so the real part is $\alpha_j \operatorname{Re}(\lambda_i) + O(\alpha_j^2)$. For sufficiently small $\alpha_j$, we have:

$$\operatorname{Re}\left(\log(1 + \beta_j \lambda_i)\right) \leq \beta_j \operatorname{Re}(\lambda_i) + \epsilon \beta_j$$

Thus $|1 + \beta_j \lambda_i| = e^{\operatorname{Re}(\log(1 + \beta_j \lambda_i))} \leq e^{\beta_j \operatorname{Re}(\lambda_i) + \epsilon \beta_j}$.

Set $\epsilon = \lambda_{\min}/2 > 0$ where $\lambda_{\min} = \min_i |\operatorname{Re}(\lambda_i)|$. Since $\operatorname{Re}(\lambda_i) \leq -\lambda_{\min}$:

$$|1 + \beta_j \lambda_i| \leq e^{\beta_j \operatorname{Re}(\lambda_i) + \beta_j \lambda_{\min}/2} \leq e^{-\beta_j \lambda_{\min} + \beta_j \lambda_{\min}/2} = e^{-\beta_j \lambda_{\min}/2}$$

when $\beta_j \leq \beta_0$. Since $\beta_j \to 0$, there exists $K_0 \in \mathbb{N}$ such that $\beta_j \leq \beta_0$ for all $j \geq K_0$.
**Case 1:** $k \geq K_0$
For all $j \geq k \geq K_0$, we have $\beta_j \leq \beta_0$, so:

$$\left| \prod_{j=k}^{t-1} (1 + \beta_j \lambda_i) \right| \leq \exp\left( -\frac{\lambda_{\min}}{2} \sum_{j=k}^{t-1} \beta_j \right)$$

**Case 2:** $k < K_0$
Split the product at $K_0$:

$$\prod_{j=k}^{t-1}(1 + \beta_j \lambda_i) = \underbrace{\left( \prod_{j=k}^{K_0-1} (1 + \beta_j \lambda_i) \right)}_{(*)} \cdot \underbrace{\left( \prod_{j=K_0}^{t-1} (1 + \beta_j \lambda_i) \right)}_{(**)}$$

Term $(*)$ is a finite product (since $K_0$ is fixed). Using $|1 + \beta_j \lambda_i| \leq 1 + |\lambda_i|\beta_j$:

$$|(*)| \leq \prod_{j=k}^{K_0-1} (1 + |\lambda_i|\beta_j) \leq \exp\left( |\lambda_i| \sum_{j=k}^{K_0-1} \beta_j \right) \leq C_i(k)$$

where $C_i(k) = \exp\left( |\lambda_i| \sum_{j=0}^{K_0-1} \beta_j \right)$ is bounded (as $\beta_j > 0$ and fixed $K_0$). Term $(**)$ is bounded by Case 1:

$$|(**)| \leq \exp\left( -\frac{\lambda_{\min}}{2} \sum_{j=K_0}^{t-1} \beta_j \right)$$

$$\leq \exp\left( -\frac{\lambda_{\min}}{2} \sum_{j=k}^{t-1} \beta_j \right) \cdot \exp\left( \frac{\lambda_{\min}}{2} \sum_{j=k}^{K_0-1} \beta_j \right)$$

Combining both terms:

$$\left|\prod_{j=k}^{t-1}(1+\beta_j\lambda_i)\right| \le C_i(k)\exp\left(\frac{\lambda_{\min}}{2}\sum_{j=k}^{K_0-1}\beta_j\right)\exp\left(-\frac{\lambda_{\min}}{2}\sum_{j=k}^{t-1}\beta_j\right)$$

$$= C_i''(k)\exp\left(-\frac{\lambda_{\min}}{2}\sum_{j=k}^{t-1}\beta_j\right)$$

where $C_i''(k) = C_i(k)\exp\left(\frac{\lambda_{\min}}{2}\sum_{j=k}^{K_0-1}\beta_j\right)$.

Since $k < K_0$ and there are only finitely many such $k$, we define the following

$$C' = \max\left\{\max_{\substack{1\le i\le d \\ 0\le k<K_0}} C_i''(k), 1\right\} < \infty$$

For $k \ge K_0$, we have $C_i''(k) = 1$. Thus, for all $i$, $k$, and $t > k$:

$$\left|\prod_{j=k}^{t-1}(1+\beta_j\lambda_i)\right| \le C'\exp\left(-\frac{\lambda_{\min}}{2}\sum_{j=k}^{t-1}\beta_j\right)$$

Therefore,

$$\left\|\prod_{j=k}^{t-1}D_j\right\| = \max_i\left|\prod_{j=k}^{t-1}(1+\beta_j\lambda_i)\right| \le C'\exp\left(-\frac{\lambda_{\min}}{2}\sum_{j=k}^{t-1}\beta_j\right)$$

Substituting into (70),

$$\|Q(t,k)\| \le \|P\|\cdot\|P^{-1}\|\cdot C'\exp\left(-\frac{\lambda_{\min}}{2}\sum_{j=k}^{t-1}\beta_j\right)$$

Set $C = \|P\|\cdot\|P^{-1}\|\cdot C'$ and $\bar{\beta} = \lambda_{\min}/2$ to obtain:

$$\|Q(t,k)\| \le C\exp\left(-\beta\sum_{j=k}^{t-1}\beta_j\right)$$

for all $t > k \ge 0$, with $C, \beta > 0$ independent of $t$ and $k$.
Therefore from (68),

$$\|e_{t+1}\| \le C\exp\left(-\beta\sum_{j=0}^{t}\beta_j\right)\|e_0\| + \sum_{k=0}^{t}\beta_k\|Q(t+1,k+1)\|\|\eta_k\|. \tag{72}$$

The first term goes to zero as $t \to \infty$ because $\sum_{j=0}^{t}\beta_j \to \infty$. For the second term, note that

$$\sum_{k=0}^{t}\beta_k\|Q(t+1,k+1)\|\|\eta_k\| \le C\sum_{k=0}^{t}\beta_k\exp\left(-\bar{\beta}\sum_{j=k+1}^{t}\beta_j\right)\|\eta_k\|. \tag{73}$$

By the summability of $\beta_k\|\eta_k\|$ and the exponential decay, this term goes to zero. Indeed, for any fixed $k$, the term goes to zero as $t \to \infty$. Moreover, the tail of the series $\sum_k \beta_k\|\eta_k\|$ is small. Therefore, by the Toeplitz lemma or direct estimation, the entire sum goes to zero.

Thus, $e_t \to 0$ a.s., i.e., $\mathbf{x}_t \to \mathbf{x}^*$ a.s. $\qquad\square$

A natural question is whether our correction mechanism can guarantee that the residual bias stays proportional to the unavoidable approximation error. The next theorem answers this affirmatively, showing that the corrected TD fixed point is never worse than a constant-factor multiple of the best value function representable by the chosen features.

First, we define the total error as:

$$
\begin{aligned}
e &= \Phi x_c^{TD} - V_\pi \\
&= \Phi(x_c^{TD} - w^*) + (\Phi w^* - V_\pi) = \Phi u + \delta,
\end{aligned}
\tag{74}
$$

where

- $w^* = \arg\min_w \|\Phi w - V_\pi\|_q$ is the best approximation under $q$-norm

- $u = x_c^{TD} - w^*$ is the difference between the TD solution and the best approximation

- $\delta = \Phi w^* - V_\pi$ is the approximation error

**Lemma 8** (Orthogonality Condition for TD Fixed Point). *The TD fixed point satisfies the orthogonality condition:*

$$
\Phi^\top \Xi_{\nu_b} \Xi_{h^*}^{-1} \Xi_q (I - \gamma P_\pi)e = 0
\tag{75}
$$

*Proof.* The TD fixed point satisfies:

$$
\Phi^\top \Xi_{\nu_b} \Xi_{h^*}^{-1} \Xi_q (I - \gamma P_\pi)\Phi x_c^{TD} = \Phi^\top \Xi_{\nu_b} \Xi_{h^*}^{-1} \Xi_q R_\pi
$$

Substituting the $R_\pi = (I - \gamma P_\pi)V_\pi$ (from Bellman equation) into the TD fixed point equation:

$$
\Phi^\top \Xi_{\nu_b} \Xi_{h^*}^{-1} \Xi_q (I - \gamma P_\pi)\Phi x_c^{TD} = \Phi^\top \Xi_{\nu_b} \Xi_{h^*}^{-1} \Xi_q (I - \gamma P_\pi)V_\pi
$$

Rearranging all terms to one side:

$$
\Phi^\top \Xi_{\nu_b} \Xi_{h^*}^{-1} \Xi_q (I - \gamma P_\pi)(\Phi x_c^{TD} - V_\pi) = 0 \quad \Rightarrow \quad \Phi^\top \Xi_{\nu_b} \Xi_{h^*}^{-1} \Xi_q (I - \gamma P_\pi)e = 0.
$$

$\square$

To rigorously validate the effectiveness of our distributional correction mechanism, we bound the approximation error relative to the fundamental limit imposed by the expressivity of the features and the target state weighting. The following result provides a worst-case guarantee that our method does not amplify unavoidable approximation errors and quantifies how design choices (e.g., the target distribution $q$, feature selection, and mixture model complexity) influence performance.

**Theorem 6** (Error Bound for Off-Policy TD with Steady-State Bias Correction). *Let $\Lambda_c$ be Hurwitz and diagonalizable and Assumptions 1-7 hold. Then, the error of the off-policy TD solution with steady-state bias correction satisfies:*

$$
\|\Phi x_c^{TD} - V_\pi\|_{\nu_b} \le C \cdot \min_w \|\Phi w - V_\pi\|_q
$$

*where:*

$$
C = \left( \frac{\|P\| \cdot \|P^{-1}\|}{|\alpha(\Lambda_c)|} \right) \cdot \sqrt{\max_s \nu_b(s)} \cdot K \cdot \sqrt{\max_s q(s)} \sigma_{\max}(\Phi)^2 \cdot (1 + \gamma\sqrt{\kappa_q}) + \sqrt{\max_s \frac{\nu_b(s)}{q(s)}}
$$

*with $\kappa_q = \max_{s'} \dfrac{\sum_s q(s)P_\pi(s'|s)}{q(s')}$, and $\alpha(\Lambda_c) = \max_i Re(\lambda_i(\Lambda_c)) < 0$ being the spectral abscissa of $\Lambda_c$.*

*Proof.* By applying triangle inequality on (74), we get

$$\|e\|_{\nu_b} \leq \|\Phi u\|_{\nu_b} + \|\delta\|_{\nu_b} \tag{76}$$

From Lemma 8, we have

$$\langle \Phi^\top \Xi_{\nu_b} \Xi_{h^*}^{-1} \Xi_q (I - \gamma P_\pi), e \rangle = 0 \tag{77}$$

Substituting $e = \Phi u + \delta$:

$$\Phi^\top \Xi_{\nu_b} \Xi_{h^*}^{-1} \Xi_q (\gamma P_\pi - I) \Phi u = -\Phi^\top \Xi_{\nu_b} \Xi_{h^*}^{-1} \Xi_q (\gamma P_\pi - I) \delta$$
$$\Rightarrow \Lambda_c u = -b_\delta,$$

where $b_\delta = \Phi^\top \Xi_{\nu_b} \Xi_{h^*}^{-1} \Xi_q (\gamma P_\pi - I) \delta$.

Since $\Lambda_c$ is Hurwitz and diagonalizable, $\Lambda_c$ is invertible and can be written as $\Lambda = PDP^{-1}$, where $D = \mathrm{diag}(\lambda_1, \dots, \lambda_k)$ with $\mathrm{Re}(\lambda_i) < 0$ for all $i$. Therefore,

$$u = -\Lambda_c^{-1} b_\delta = -PD^{-1}P^{-1} b_\delta \tag{78}$$

By taking norms on either side, we get

$$\|u\| \leq \|\Lambda_c^{-1}\| \cdot \|b_\delta\| \leq \|P\| \cdot \|P^{-1}\| \cdot \|D^{-1}\| \cdot \|b_\delta\| \tag{79}$$

Since $D$ is diagonal with entries $\lambda_i$:

$$\|D^{-1}\| = \max_i \left| \frac{1}{\lambda_i} \right| = \frac{1}{\min_i |\lambda_i|}$$

Let $\alpha(\Lambda_c) = \max_i \mathrm{Re}(\lambda_i) < 0$. For any eigenvalue $\lambda_i = a_i + b_i i$, we have $|\lambda_i| = \sqrt{a_i^2 + b_i^2} \geq |a_i| = |\mathrm{Re}(\lambda_i)|$. Therefore,

$$\|D^{-1}\| \leq \frac{1}{|\alpha(\Lambda_c)|} \tag{80}$$

So, from (79), we get

$$\|u\| \leq \left( \frac{\|P\| \cdot \|P^{-1}\|}{|\alpha(\Lambda_c)|} \right) \cdot \|b_\delta\| \tag{81}$$

Now we bound $\|b_\delta\| = \|\Phi^\top \Xi_{\nu_b} \Xi_{h^*}^{-1} \Xi_q (\gamma P_\pi - I) \delta\|$. Using the $q$-weighted inner product and Cauchy-Schwarz inequality:

$$|v^\top b_\delta| = |\langle (\nu_b/h^*)\Phi v, (\gamma P_\pi - I)\delta \rangle_q|$$
$$\leq \|(\nu_b/h^*)\Phi v\|_q \cdot \|(\gamma P_\pi - I)\delta\|_q \tag{82}$$

We first, bound $\|(\nu_b/h^*)\Phi v\|_q$:

$$\|(\nu_b/h^*)\Phi v\|_q^2 = \sum_s q(s) \left( \frac{\nu_b(s)}{h^*(s)} \right)^2 (\Phi v(s))^2$$

$$\leq \max_s \left( \frac{\nu_b(s)}{h^*(s)} \right)^2 \cdot \max_s q(s) \cdot \|\Phi v\|^2$$

$$\leq K^2 \cdot \max_s q(s) \cdot \sigma_{\max}(\Phi)^2 \cdot \|v\|^2$$

Therefore,

$$\|(\nu_b/h^*)\Phi v\|_q \leq K \cdot \sqrt{\max_s q(s)} \cdot \sigma_{\max}(\Phi) \cdot \|v\| \tag{83}$$

Next, we bound $\|(\gamma P_\pi - I)\delta\|_q$. Let $\mu(s') = \sum_s q(s) P_\pi(s'|s)$, which is the next-state distribution under policy $\pi$ when starting from distribution $q$. Similar to Lemma 1, one can obtain the following:

$$\|P_\pi f\|_q^2 = \sum_s q(s)(P_\pi f(s))^2 \leq \sum_s q(s) \sum_{s'} P_\pi(s'|s) f(s')^2 \leq \underbrace{\left( \max_{s'} \frac{\mu(s')}{q(s')} \right)}_{\kappa_q} \cdot \|f\|_q^2$$

Then, $\|P_\pi\|_q \leq \sqrt{\kappa_q}$. Therefore,

$$\|(\gamma P_\pi - I)\delta\|_q \leq \gamma\|P_\pi\delta\|_q + \|\delta\|_q \leq \gamma\sqrt{\kappa_q} \cdot \|\delta\|_q + \|\delta\|_q = (1 + \gamma\sqrt{\kappa_q}) \cdot \|\delta\|_q$$

Combining these results:

$$\|b_\delta\| \leq K \cdot \sqrt{\max_s q(s)} \cdot \sigma_{\max}(\Phi) \cdot (1 + \gamma\sqrt{\kappa_q}) \cdot \|\delta\|_q \tag{84}$$

Now, we bound the approximation error under $\nu_b$:

$$\|\delta\|_{\nu_b}^2 = \sum_s \nu_b(s)\delta(s)^2 \leq \left(\max_s \frac{\nu_b(s)}{q(s)}\right) \cdot \sum_s q(s)\delta(s)^2 = \left(\max_s \frac{\nu_b(s)}{q(s)}\right) \cdot \|\delta\|_q^2$$

Therefore,

$$\|\delta\|_{\nu_b} \leq \sqrt{\max_s \frac{\nu_b(s)}{q(s)}} \cdot \|\delta\|_q \tag{85}$$

Finally, we combine all components in (74):

$$\|e\|_{\nu_b} \leq \|\Phi u\|_{\nu_b} + \|\delta\|_{\nu_b}$$

$$\leq \left(\frac{\|P\| \cdot \|P^{-1}\|}{\alpha(\Lambda_c)}\right) \cdot \sqrt{\max_s \nu_b(s)} \cdot K \cdot \sqrt{\max_s q(s)} \cdot \sigma_{\max}(\Phi)^2 \cdot (1 + \gamma\sqrt{\kappa_q}) \cdot \|\delta\|_q + \sqrt{\max_s \frac{\nu_b(s)}{q(s)}} \cdot \|\delta\|_q$$

Since $\|\delta\|_q = \min_w \|\Phi w - V_\pi\|_q$, and by the definition (74) of $e$ we obtain the claim:

$$\|\Phi x_c^{TD} - V_\pi\|_{\nu_b} \leq C \cdot \min_w \|\Phi w - V_\pi\|_q, \tag{86}$$

where

$$C = \left(\frac{\|P\| \cdot \|P^{-1}\|}{\alpha(\Lambda_c)}\right) \cdot \sqrt{\max_s \nu_b(s)} \cdot K \cdot \sqrt{\max_s q(s)}\sigma_{\max}(\Phi)^2 \cdot (1 + \gamma\sqrt{\kappa_q}) + \sqrt{\max_s \frac{\nu_b(s)}{q(s)}}.$$

$\square$

The above theorem demonstrates that the error of our corrected solution is proportional to the minimal approximation error under the target distribution $q$, scaled by factors capturing policy misalignment, feature conditioning, and steady-state estimation accuracy. This establishes that our algorithm achieves near-optimal performance within the constraints of the representation, while explicitly quantifying the cost of distribution shift correction. The bound further elucidates the trade-offs between policy similarity, distribution estimation quality, and feature design. This bound further provides several relevant insights, which are in order:

1. *Fundamental Error Relationship*: The error in the TD solution is proportional to the best possible approximation error, establishing that the algorithm achieves the best possible performance within the function approximation class.

2. *Steady-State Estimation Quality*: The term $K = \max_s \frac{\nu_b(s)}{h(s)}$ quantifies the impact of steady-state distribution estimation error. When $K \approx 1$ (accurate estimation), the bound tightens, validating the steady-state bias correction approach.

3. *Policy Alignment*: The term $(1 + \gamma\sqrt{\kappa_q})$ with $\kappa_q = \max_{s'} \frac{\mu(s')}{q(s')}$ measures policy dissimilarity. Smaller $\kappa$ (more similar policies) leads to tighter bounds, explaining why off-policy learning becomes challenging with dissimilar policies.

4. *Feature Representation*: The term $\sigma_{\max}(\Phi)^2$ shows that well-conditioned feature representations (smaller $\sigma_{\max}$) lead to better error bounds.

5. *Distributional Factors*: The terms $\sqrt{\max_s \nu_b(s)}$, $\sqrt{\max_s q(s)}$, and $\sqrt{\max_s \frac{\nu_b(s)}{q(s)}}$ capture how state distribution properties affect performance.

While Theorem 3 provides an error bound under the behaviour policy's stationary distribution $\nu_b$, in many practical scenarios we are ultimately interested in the prediction accuracy under the design distribution $q$. The following corollary establishes that our steady-state bias correction method also provides strong guarantees in the design distribution norm.

**Corollary 2** (Error Bound in $q$-Norm). *Under the same assumptions as Theorem 3, the error of the corrected off-policy TD solution under the design distribution satisfies:*

$$\left\|\Phi\mathbf{x}_c^{\mathrm{TD}} - V_\pi\right\|_q \leq C_q \cdot \min_{\mathbf{w}} \left\|\Phi\mathbf{w} - V_\pi\right\|_q, \;\; where \; C_q = \sqrt{\max_{s \in S} \frac{q(s)}{\nu_b(s)}} \cdot C,$$

*and $C$ is the constant from Theorem 3.*

*Proof.* Since the state space $S$ is finite and $\nu_b(s) > 0$ for all $s \in S$ (by Assumption 1), $\max_{s \in S} \frac{q(s)}{\nu_b(s)}$ is finite. Hence,

$$\begin{aligned}
\|\Phi\mathbf{x}_c^{\mathrm{TD}} - V_\pi\|_q^2 &= \sum_{s \in S} q(s)(\Phi\mathbf{x}_c^{\mathrm{TD}} - V_\pi)(s)^2 \\
&= \sum_{s \in S} \nu_b(s) \left(\frac{q(s)}{\nu_b(s)}\right)(\Phi\mathbf{x}_c^{\mathrm{TD}} - V_\pi)(s)^2 \\
&\leq \sum_{s \in S} \nu_b(s) \cdot \max_{s \in S} \frac{q(s)}{\nu_b(s)} \cdot (\Phi\mathbf{x}_c^{\mathrm{TD}} - V_\pi)(s)^2 \\
&= \max_{s \in S} \frac{q(s)}{\nu_b(s)} \|\Phi\mathbf{x}_c^{\mathrm{TD}} - V_\pi\|_{\nu_b}^2 .
\end{aligned}$$

Taking square roots and applying Theorem 3, we get,

$$\|\Phi\mathbf{x}_c^{\mathrm{TD}} - V_\pi\|_q \leq \sqrt{\max_{s \in S} \frac{q(s)}{\nu_b(s)}} \|\Phi\mathbf{x}_c^{\mathrm{TD}} - V_\pi\|_{\nu_b} \leq \sqrt{\max_{s \in S} \frac{q(s)}{\nu_b(s)}} \cdot C \cdot \min_{\mathbf{w}} \|\Phi\mathbf{w} - V_\pi\|_q .$$

$\square$

**Remark 4.** *The constant $C_q$ reveals an important trade-off: while our correction mechanism aims to align the solution with the target distribution $q$, the final error bound depends on the maximum density ratio between $q$ and $\nu_b$. This highlights the fundamental importance of coverage: if the behaviour policy rarely visits states that are important under $q$ (i.e., $\nu_b(s) \ll q(s)$ for some $s$), then $R$ becomes large and the bound degrades. This aligns with intuition and provides theoretical justification for the empirical observation that good behaviour policies should have adequate coverage of the target distribution's support.*

**Remark 5.** *When $q = \nu_\pi$ (the ideal case where we know the target policy's stationary distribution), Corollary 2 provides a bound on the error relative to the true evaluation metric of interest. The constant $C_q$ then depends on the distribution mismatch coefficient $\max_s \frac{\nu_\pi(s)}{\nu_b(s)}$, which quantifies how well the behaviour policy covers the target policy's state visitation pattern.*

To complete the convergence analysis, we now provide a sufficient condition for the matrix $\Lambda_c$ to be Hurwitz, which ensures the asymptotic stability of the algorithm.

**Theorem 7** (Hurwitz Condition). *$\Lambda_c = \Phi^\top \Xi_{\nu_b} \Xi_{h^*}^{-1} \Xi_q (\gamma P_\pi - I)\Phi$ is Hurwitz (all eigenvalues have strictly negative real parts) if and only if $K_q \kappa_q \gamma^2 < 1$, where $\kappa_q = \max_{s'} \frac{\mu_q(s')}{q(s')}$ with $\mu_q(s') = \sum_s q(s)P_\pi(s'|s)$ and $K_q = \max_s \frac{\Xi_{\nu_b}(s)\Xi_{h^*}^{-1}(s)}{q(s)}$.*

*Proof.* Consider the quadratic form $\mathbf{w}^\top \Lambda_c \mathbf{w}$ for any $\mathbf{w} \neq 0$:

$$\mathbf{w}^\top \Lambda_c \mathbf{w} = \mathbf{w}^\top \Phi^\top \Xi_{\nu_b} \Xi_{h^*}^{-1} \Xi_q (\gamma P_\pi - I)\Phi\mathbf{w}$$

Let $\mathbf{u} = \Phi\mathbf{w}$. By Assumption 2, $\mathbf{u} \neq 0$ since $\text{rank}(\Phi) = k$. Then:

$$\mathbf{w}^\top \Lambda_c \mathbf{w} = \mathbf{u}^\top \Xi_{\nu_b} \Xi_{h^*}^{-1} \Xi_q (\gamma P_\pi - I) \mathbf{u} = \gamma \mathbf{u}^\top \Xi_{\nu_b} \Xi_{h^*}^{-1} \Xi_q P_\pi \mathbf{u} - \mathbf{u}^\top \Xi_{\nu_b} \Xi_{h^*}^{-1} \Xi_q \mathbf{u}$$

Let $Q_1 = \gamma \mathbf{u}^\top \Xi_{\nu_b} \Xi_{h^*}^{-1} \Xi_q P_\pi u$ and $Q_2 = \mathbf{u}^\top \Xi_{\nu_b} \Xi_{h^*}^{-1} \Xi_q \mathbf{u}$. First, observe that $Q_2 > 0$ since $\Xi_{\nu_b}, \Xi_{h^*}^{-1}, \Xi_q$ are all positive definite diagonal matrices (by Assumption 1 and the fact that $h^*$ is a valid distribution estimate).

For $Q_1$, apply the Cauchy-Schwarz inequality:

$$|Q_1| \leq \gamma \|\mathbf{u}\|_{\Xi_{\nu_b} \Xi_{h^*}^{-1} \Xi_q} \cdot \|P_\pi \mathbf{u}\|_{\Xi_{\nu_b} \Xi_{h^*}^{-1} \Xi_q}, \text{ where } \|\mathbf{x}\|_{\Xi_{\nu_b} \Xi_{h^*}^{-1} \Xi_q} = \sqrt{\mathbf{x}^\top \Xi_{\nu_b} \Xi_{h^*}^{-1} \Xi_q \mathbf{x}}.$$

Now, we need to bound $\|P_\pi \mathbf{u}\|_{\Xi_{\nu_b} \Xi_{h^*}^{-1} \Xi_q}^2$:

$$\begin{aligned}
\|P_\pi \mathbf{u}\|_{\Xi_{\nu_b} \Xi_{h^*}^{-1} \Xi_q}^2 &= \mathbf{u}^\top (P_\pi)^\top \Xi_{\nu_b} \Xi_{h^*}^{-1} \Xi_q P_\pi \mathbf{u} \\
&= \sum_s \sum_{s'} \sum_{s''} \mathbf{u}(s) P_\pi(s'|s) \Xi_{\nu_b}(s) \Xi_{h^*}^{-1}(s) \Xi_q(s) P_\pi(s''|s') \mathbf{u}(s'') \\
&= \sum_{s'} \Xi_q(s') u(s') \left( \sum_s \frac{\Xi_{\nu_b}(s) \Xi_{h^*}^{-1}(s)}{\Xi_q(s')} P_\pi(s'|s) u(s) \right)
\end{aligned}$$

Note that $K_q = \max_s \frac{\Xi_{\nu_b}(s) \Xi_{h^*}^{-1}(s)}{q(s)}$, is bounded since $\nu_b$ and $h^*$ are positive distributions on a finite state space. Then,

$$\begin{aligned}
\|P_\pi \mathbf{u}\|_{\Xi_{\nu_b} \Xi_{h^*}^{-1} \Xi_q}^2 &\leq K_q \sum_{s'} q(s') \mathbf{u}(s') \left( \sum_s P_\pi(s'|s) \mathbf{u}(s) \right) \\
&\leq K_q \sum_{s'} q(s') \mathbf{u}(s') \sqrt{\sum_s P_\pi(s'|s) \mathbf{u}(s)^2} \quad \text{(by Jensen's inequality)} \\
&\leq K_q \sqrt{\sum_{s'} q(s') \mathbf{u}(s')^2} \cdot \sqrt{\sum_{s'} q(s') \sum_s P_\pi(s'|s) \mathbf{u}(s)^2} \\[2ex]
&= K_q \|\mathbf{u}\|_q \cdot \sqrt{\sum_s \mathbf{u}(s)^2 \sum_{s'} q(s') P_\pi(s'|s)} \\
&= K_q \|\mathbf{u}\|_q \cdot \sqrt{\sum_s \mathbf{u}(s)^2 \mu_q(s)} \\
&\leq K_q \sqrt{\kappa_q} \|\mathbf{u}\|_q^2 \quad \text{where } \kappa_q = \max_{s'} \frac{\mu_q(s')}{q(s')}
\end{aligned}$$

Thus, $\|P_\pi \mathbf{u}\|_{\Xi_{\nu_b} \Xi_{h^*}^{-1} \Xi_q} \leq \sqrt{K_q \kappa_q} \|\mathbf{u}\|_q$. Now, since $\|\mathbf{u}\|_{\Xi_{\nu_b} \Xi_{h^*}^{-1} \Xi_q} \geq \sqrt{K_q^{-1}} \|\mathbf{u}\|_q$, we have,

$$|Q_1| \leq \gamma \sqrt{K_q \kappa_q} \|\mathbf{u}\|_{\Xi_{\nu_b} \Xi_{h^*}^{-1} \Xi_q}^2$$

Therefore:

$$\mathbf{w}^\top \Lambda_c \mathbf{w} \leq (\gamma \sqrt{K_q \kappa_q} - 1) \|\mathbf{u}\|_{\Xi_{\nu_b} \Xi_{h^*}^{-1} \Xi_q}^2$$

When $K_q \kappa_q \gamma^2 < 1$, we have $\gamma \sqrt{K_q \kappa_q} < 1$, and thus $\mathbf{w}^\top \Lambda_c \mathbf{w} < 0$ for all $w \neq 0$.

This proves that $\Lambda_c$ is negative definite, and therefore all its eigenvalues have strictly negative real parts (Hurwitz). $\qquad\square$

In Algorithm 1, the mixture weights $\bar{\lambda}_t$ must satisfy probability simplex constraints ($\lambda_i \geq 0$, $\sum_{i=1}^{\ell} \lambda_i = 1$) after each gradient update. To enforce this, we employ an efficient projection method that provides an optimal $O(\ell \log \ell)$ Euclidean projection onto the simplex $\Delta^\ell$ (Wang & Carreira-Perpinán, 2013). The method sorts the components of $\bar{\lambda}$, determines an optimal threshold, and redistributes mass, providing the closest valid point in the simplex $\Delta^\ell$ while preserving sparsity patterns when possible which attributes to its $O(\ell \log \ell)$ complexity. Specifically, it seeks the solution to the following optimization problem:

$$\min_{\lambda} \frac{1}{2}\|\lambda - v\|^2, \quad \text{subject to} \sum_{i=1}^{\ell} \lambda_i = 1, \quad \lambda_i \geq 0.$$

Using a Lagrange multiplier $\tau$ for the equality constraint $\sum_{i=1}^{\ell} \lambda_i = 1$, we define the Lagrangian:

$$\mathcal{L}(\lambda, \tau) = \frac{1}{2} \sum_{i=1}^{\ell} (\lambda_i - v_i)^2 - \tau \left( \sum_{i=1}^{\ell} \lambda_i - 1 \right).$$

Now solving for $\lambda_i$ by taking the derivative $w.r.t.$ $\lambda_i$:

$$\frac{\partial \mathcal{L}}{\partial \lambda_i} = \lambda_i - v_i - \tau = 0 \Rightarrow \lambda_i = v_i + \tau.$$

Now to enforce the simplex constraint, we sum over all $i$:

$$\sum_{i=1}^{\ell} \lambda_i = \sum_{i=1}^{\ell} (v_i + \tau) = 1 \Rightarrow \tau = \frac{1 - \sum_{i=1}^{\ell} v_i}{\ell}.$$

Thus, the projection without considering the non-negativity constraint is: $\lambda_i = v_i + \frac{1 - \sum_{i=1}^{\ell} v_i}{\ell}$. If any $\lambda_i < 0$, we modify the solution by clipping negative values to zero and redistributing the remaining weight. This is efficiently handled by sorting $v$ in descending order and determining a threshold $\tau$ such that the projected vector remains non-negative.

---

**Algorithm 2:** Euclidean projection onto $\Delta^\ell$

---

**Function** $\Pi_{\Delta^\ell} (\lambda \in \mathbb{R}^\ell)$

$\quad$ Sort $\lambda$ into $\eta$: $\eta_1 \geq \eta_2 \geq \cdots \geq \eta_\ell$

$\quad$ $\tau = \texttt{max}\{1 \leq j \leq \ell : \eta_j + \frac{1}{j} \left( 1 - \sum_{i=1}^{j} \eta_i \right) > 0\}$

$\quad$ $y = \frac{1}{\tau} \left( 1 - \sum_{i=1}^{\tau} \eta_i \right)$

$\quad$ return $\hat{\lambda} = \texttt{max}\{\lambda_i + y, 0\}$, $i \in \{1, 2, \ldots, \ell\}$

---

## 4 Experiments & Results

Here, we present a comprehensive empirical evaluation of the proposed Steady-State Bias Correction (SSBC-TD) algorithm across diverse benchmark domains. The experiments are designed to validate the method's effectiveness in mitigating steady-state distribution mismatch in off-policy TD learning with linear function approximation. We assess performance using Root Mean Square Error (RMSE) of value predictions: $\text{RMSE} = \|V_\pi - \Phi\mathbf{x}\|_2$ against true value functions. All results are averaged over 10 independent runs to ensure statistical robustness. Key aspects evaluated include:

- **Generalization across domains**: Discrete (Circle Chain (Prediction + Control), Gridworld Cliff Walking, Taxi) and continuous (Mountain Car, CartPole, Acrobot) state spaces

- **Hyperparameter sensitivity**: Impact of step-sizes ($\alpha_t$, $\beta_t$) on convergence

- **Trajectory robustness**: Performance under varying episodic path structures

- **Distributional fidelity**: Accuracy of Gaussian mixture approximations for stationary distributions

- **Discount factor and design distribution sensitivity**: Impact of discount factor $\gamma$ and design distribution $q$ on prediction error.

All environments are modified to ensure ergodicity (e.g., respawning agents in terminal states) for well-defined steady-state distributions.

## 4.1 Discrete Domain

### 4.1.1 Circle Chain

We consider a ring-structured Markov chain with $n = 100$ discrete states $S = \{0, 1, \ldots, n-1\}$ and circular distance $d(i, j) = \min\{|i - j|, n - |i - j|\}$. Each chain is defined by a transition kernel that moves Left or Right between neighbouring states while preserving irreducibility and aperiodicity. To ensure that the stationary distribution of each chain is non-uniform yet unique, we construct the transition probabilities to satisfy detailed balance with respect to a desired wrapped-Gaussian distribution $\nu$ on the ring:

$$\nu(i)\, P_{i,i+1} = \nu(i+1)\, P_{i+1,i}, \qquad i \in S.$$

This is implemented by choosing

$$P_{i,i+1} = \tfrac{1}{2}\min\left\{1, \tfrac{\nu(i+1)}{\nu(i)}\right\}, \quad P_{i,i-1} = \tfrac{1}{2}\min\left\{1, \tfrac{\nu(i-1)}{\nu(i)}\right\}, \quad P_{i,i} = 1 - P_{i,i+1} - P_{i,i-1},$$

yielding an irreducible, reversible, and uniquely stationary Markov chain with $\nu$ as its stationary law. This Metropolis–Hastings construction ensures that the induced steady-state remains non-trivial even after full mixing. The target stationary distribution is fixed as a wrapped Gaussian centred at 0:

$$\nu_\pi(i) \propto \exp\left[ -\tfrac{d(i,0)^2}{2\sigma^2} \right],$$

while the behaviour chain's stationary distribution is its spatially shifted counterpart

$$\nu_b^{(\rho)}(i) \propto \exp\left[ -\tfrac{d(i,c(\rho))^2}{2\sigma^2} \right], \qquad c(\rho) = (1-\rho)\, n/2, \quad \rho \in [0,1].$$

As $\rho$ decreases from 1 to 0, the mode of $\nu_b^{(\rho)}$ moves antipodally across the ring, generating controllable distributional mismatch between behaviour and target steady-states. The auxiliary design distribution $q$ is uniform, approximating the steady-state of a fully mixing kernel and representing an uninformative bias baseline. This choice reinstates the equal-weight treatment of states. The behaviour steady-state distribution $\nu_b^{(\rho)}$ is approximated by a Gaussian-mixture surrogate $h_{\widehat{\theta}^\star, \widehat{\lambda}^\star}$ with $\ell = 10$ wrapped components.

Figure 3a shows that SSBC-TD consistently attains lower RMSE than baseline off-policy TD yet remains above the on-policy limit, confirming effective bias mitigation without instability. Figure 3b highlights the structural discrepancy between the uniform design $q$ and the localized target steady-state $\nu_\pi$, a principal source of off-policy bias. Finally, Figure 3c demonstrates how the behavioural distribution $\nu_b^{(\rho)}$ moves from orthogonal to aligned as $\rho$ increases, with the fitted mixture surrogate accurately tracking this progression. Together, these results verify that SSBC-TD maintains asymptotic consistency and achieves a favourable bias–variance trade-off as the steady-state overlap between behaviour and target policies improves.

## 4.2 Control: Policy Optimization under Steady-State Bias Correction

We extend the steady-state bias correction framework from off-policy prediction to control on the Circle-Chain MDP. As before, the environment comprises $n = 100$ cyclically connected states and two actions $\mathcal{A} = \{\text{Left}, \text{Right}\}$. Rewards follow a Gaussian centered at $s = 0$:

$$r(s) = \exp[-d(s,0)^2/(2\sigma_r^2)], \qquad \sigma_r = 5, \tag{87}$$

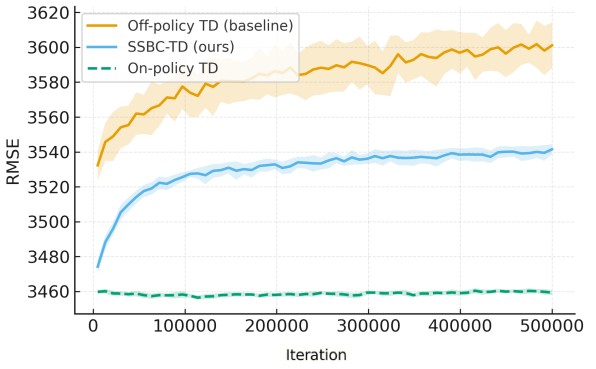 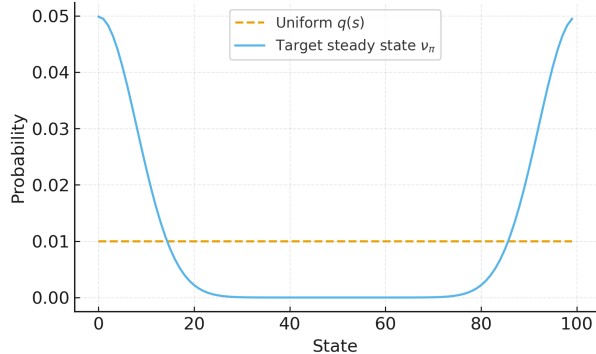

**(a)** RMSE evolution across horizons; SSBC-TD interpolates between on- and off-policy TD.

**(b)** Uniform design distribution $q(s)$ versus non-uniform target steady-state $\nu_\pi$.

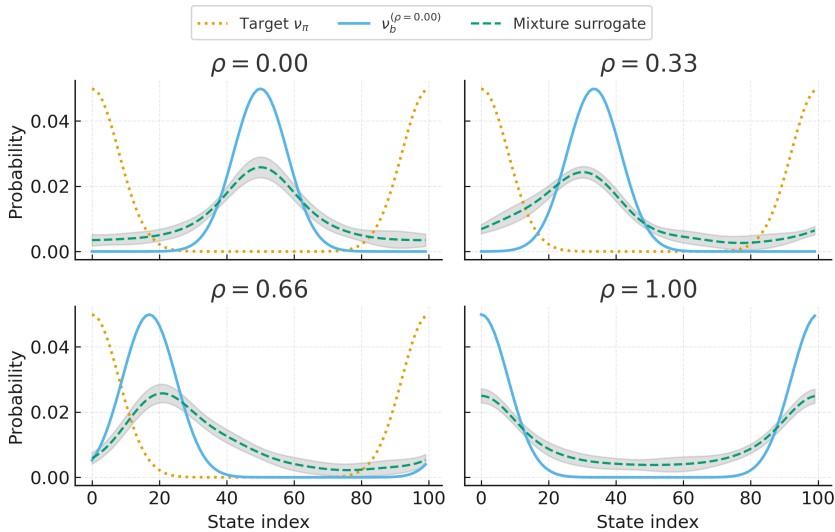

**(c)** Evolution of $\nu_b^{(\rho)}$ and its mixture surrogate across $\rho \in \{0, 0.33, 0.66, 1\}$; the surrogate accurately tracks the shifting behavioural distribution.

**Figure 3: Circle Chain experiments.** (a) SSBC-TD achieves intermediate RMSE between on- and off-policy TD, mitigating steady-state bias while maintaining stability. (b) The uniform weighting $q$ differs from $\nu_\pi$, motivating reweighted updates. (c) As $\rho$ increases, $\nu_b^{(\rho)}$ progressively aligns with $\nu_\pi$, and the mixture surrogate ($\ell = 10$) closely follows this transition, illustrating smooth bias reduction as overlap grows.

with discount factor $\gamma = 0.9$. The behaviour policy induces a stationary distribution $\nu_b$ concentrated near the antipodal region of the reward peak ($s \approx 50$), creating a strong distributional mismatch with the optimal policy's stationary measure $\nu_\pi$. The auxiliary design distribution is uniform, $q(s) = 1/n$, representing an uninformative baseline.

**Rationale for a uniform $q$:** In classical dynamic programming (value or policy iteration with a known model), Bellman backups are pointwise, and the contraction is measured in the $\| \cdot \|_\infty$ norm, giving equal weight to all states. In contrast, trajectory-based off-policy TD with function approximation minimizes a projected Bellman error under the behaviour steady-state geometry: $\Phi^\top \Xi_{\nu_b}(\Phi x - T_\pi(\Phi x)) = 0$, which overweights frequently visited states and underweights others, inducing a steady-state bias. Replacing $\Xi_{\nu_b}$ by $\Xi_q$ through multiplicative weights $\zeta(s) \propto q(s)/h(s)$ yields the fixed point of $\Pi_q T_\pi$. Choosing a uniform $q$ restores the state-agnostic weighting of classical value or policy iteration, where all states contribute equally to the Bellman residual. This aligns off-policy TD with the unbiased model-based ideal, removing systematic

skew from $\nu_b$ while maintaining numerical stability. Practically, a uniform $q$ serves as a robust surrogate for the (unknown) target steady-state distribution $\nu_\pi$, bridging trajectory-based and classical value-iteration perspectives.

**Algorithmic procedure:** Policy iteration alternates between value estimation (via naive off-policy TD or SSBC-TD) and greedy improvement using

$$Q_k(s,a) = r(s,a) + \gamma \sum_{s'} P(s,a,s')\, \phi(s')^\top x_k, \tag{88}$$

where $x_k$ denotes the value-parameter estimate from the preceding evaluation phase and $\phi(s)$ are the RBF features. The next policy is updated as

$$\pi_{k+1}(a \mid s) = \mathbf{1}\Big\{a = \arg\max_{a'} Q_k(s,a')\Big\}, \tag{89}$$

with deterministic tie-breaking. Each evaluation phase performs 2000 TD updates with the same feature representation (20 RBFs) and step-size schedule; results are averaged across 50 independent random seeds.

Figure 4a shows the discounted return across policy iterations. SSBC-TD attains faster and higher returns than naive off-policy TD, with narrower variability bands. This confirms that re-projecting the Bellman operator in $L^2(q)$ suppresses the instability caused by the skewed visitation frequencies of $\nu_b$. Figure 4b reports the RMSE between the estimated and true values $v^*$. The bias-corrected variant converges more rapidly and achieves a lower steady-state error, verifying that geometric alignment with $q$ mitigates the accumulation of long-horizon bias. Figure 4c visualizes the learned action probabilities $P(\text{Right} \mid s)$. The SSBC-TD policy exhibits a sharp, symmetric decision boundary that efficiently drives trajectories toward the reward centre, whereas the naive policy remains blurred and asymmetric, reflecting persistent influence from $\nu_b$.

Across all metrics, SSBC-TD obtains superior control behaviour—higher asymptotic returns, reduced RMSE, and a sharper optimal policy—while maintaining stability comparable to on-policy TD. By explicitly reweighting the evaluation step toward a uniform $q$, the algorithm neutralizes steady-state bias from the behaviour chain and ensures that the policy-improvement operator follows the true Bellman gradient under the intended state weighting. Empirically, these findings confirm that steady-state correction not only restores consistency in value prediction but also enables bias-free off-policy control with markedly improved sample efficiency.

### 4.2.1 Gridworld Cliff Walking

In the Gridworld Cliff Walking domain, the agent operates on a discrete $10 \times 10$ grid ($|S| = 100$) where the objective is to reach the goal while avoiding the cliff region. We set $\gamma = 0.9$ and employ a behaviour policy biased toward the left side of the grid, producing a left-skewed steady-state distribution $\nu_b$. The target policy $\pi$ prefers the safer right-side cells, inducing a right-skewed steady-state $\nu_\pi$. The correction distribution $q$ is uniform across states, providing a flat reference for bias correction. The surrogate $h_{\widehat{\theta}^*,\widehat{\lambda}^*}$ approximates $\nu_b$ using a mixture of $\ell = 5$ Gaussian components over discrete state indices.

Figure 5a demonstrates that SSBC-TD reduces the steady-state bias substantially compared with the baseline off-policy TD method, approaching the accuracy of on-policy learning while maintaining stability across iterations. The results confirm that steady-state correction yields lower asymptotic error and improved smoothness of convergence. Figures 5b–5c illustrate the underlying distributional mismatch and its correction. The uniform $q(s)$ does not capture the asymmetry present in $\nu_\pi$, which leads to biased value estimation in standard off-policy TD. By contrast, the surrogate $h_{\widehat{\theta}^*,\widehat{\lambda}^*}$ sufficiently approximates $\nu_b$, ensuring acceptable reweighting of the temporal-difference updates and stabilizing the projected Bellman operator. The narrow uncertainty band observed for the mixture indicates that the surrogate fit is consistent across multiple runs. Overall, SSBC-TD achieves a balanced trade-off between bias reduction and variance control, validating its robustness in environments with asymmetric occupancy distributions.

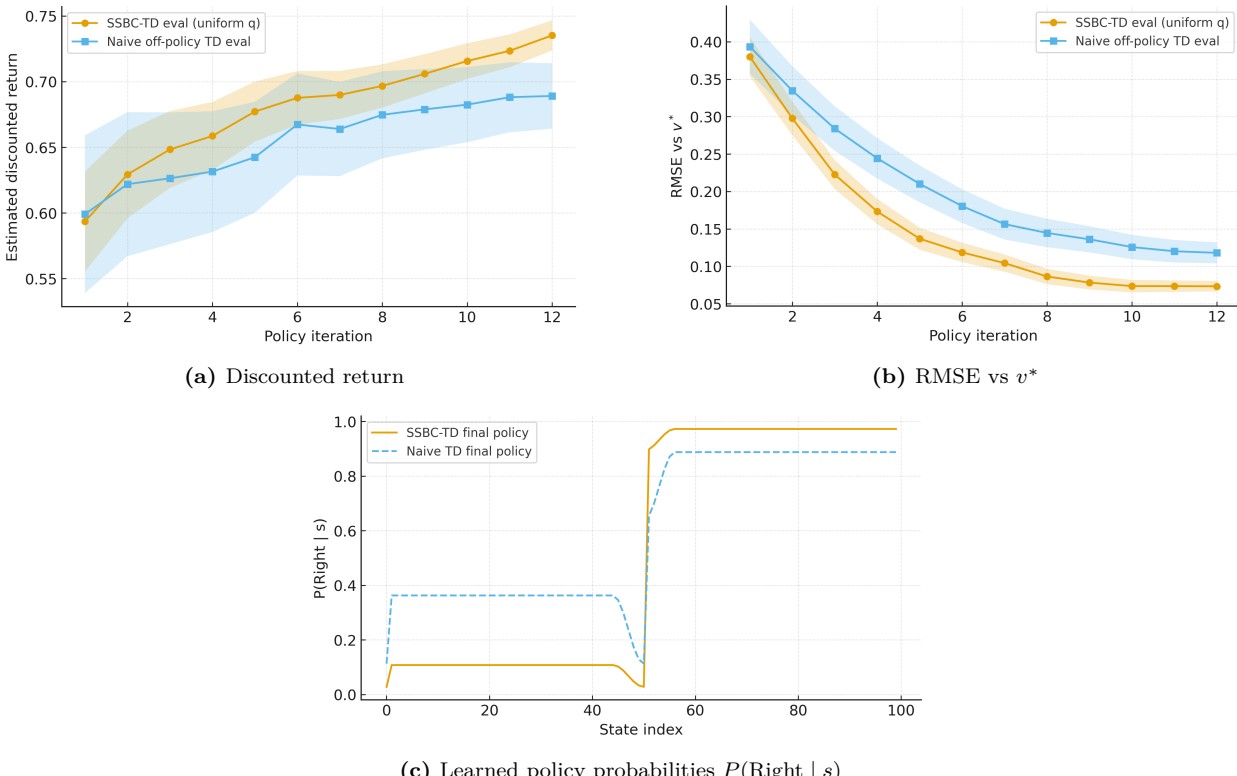

**(a)** Discounted return

**(b)** RMSE vs $v^*$

**(c)** Learned policy probabilities $P(\text{Right} \mid s)$

**Figure 4: Circle Chain control results.** (a) Discounted return per policy iteration. (b) RMSE between estimated and true values. (c) Final policy distribution $P(\text{Right} \mid s)$. SSBC-TD achieves higher asymptotic return, reduced steady-state error, and a sharper optimal boundary compared to naive off-policy TD, demonstrating effective bias correction and improved sample efficiency.

### 4.2.2 Taxi Domain

The Taxi environment consists of a finite tabular state space $S$ with $|S| = 500$ states encoding the taxi location, passenger position, and destination, and a discrete action set representing navigation and pickup/drop-off actions. We consider an off-policy prediction task where the behaviour policy $\pi_b$ selects each action uniformly at random in every state. Under this policy, the induced stationary distribution $\nu_b$ is empirically close to uniform, and hence the steady-state bias that arises from evaluating the value function under $\nu_b$ instead of the evaluation distribution is inherently small.

We evaluate our algorithm with a discount factor of $\gamma = 0.2$. The behaviour steady-state distribution $\nu_b$ is approximated by a Gaussian mixture, where each component is a truncated Gaussian distribution over the closed interval $[0, 600]$. We set $\ell = 10$ mixture components. The design distribution $q$ is chosen to be uniform over all states, $q(s) = 1/|S|$, ensuring that all states receive equal weight in the RMSE metric. The correction factor $\zeta_t$ then reweights TD updates so that their effective sampling distribution aligns with this uniform target.

From Figure 6, both the base off-policy algorithm and our approach achieve nearly identical results. Because the uniform behaviour policy induces an almost uniform stationary distribution across the 500 discrete states, the mismatch between $\nu_b$ and the evaluation distribution is minimal in the Taxi domain. Consequently, the uncorrected off-policy TD baseline already exhibits low bias, and all methods converge to similar asymptotic RMSE values. Nevertheless, Figure 6a shows that SSBC-TD consistently tracks the on-policy TD curve while maintaining slightly higher variance due to its correction factor $\zeta_t$. The improvement over the plain off-policy baseline indicates that even small distributional mismatches are effectively mitigated by the correction. The on-policy and SSBC-TD curves nearly coincide, confirming that the state-distribution correction produces on-policy–like behaviour in this domain. Figure 6b further illustrates that the learned surrogate $h_{\widehat{\theta}^*,\widehat{\lambda}^*}$

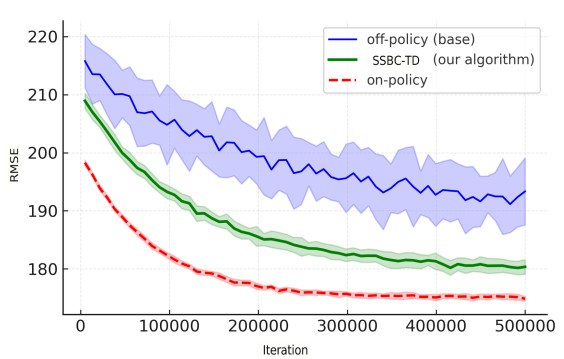
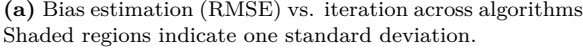

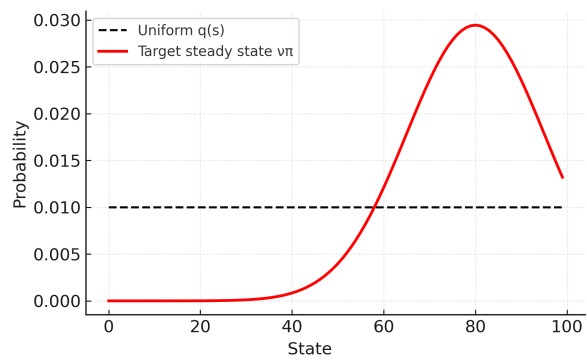

**(a)** Bias estimation (RMSE) vs. iteration across algorithms. Shaded regions indicate one standard deviation.

**(b)** Uniform $q(s)$ vs. target steady-state distribution $\nu_\pi$. The target policy induces a right-skewed stationary occupancy measure, while $q$ remains flat.

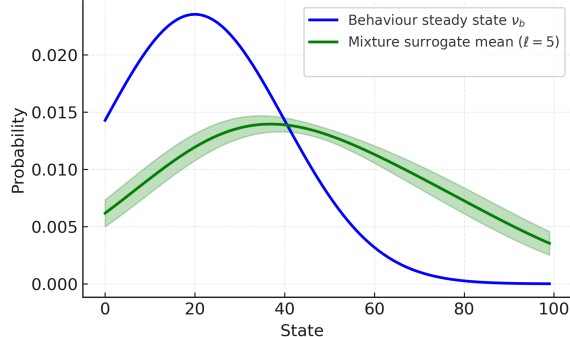

**(c)** Behaviour steady-state $\nu_b$ vs. Gaussian-mixture surrogate $h(\cdot; \widehat{\theta}^*, \widehat{\lambda}^*)$ ($\ell = 5$). The mixture captures the left-skewed nature of $\nu_b$ with uncertainty.

**Figure 5: Gridworld Cliff Walking.** (a) SSBC-TD consistently bridges the gap between on- and off-policy learning, achieving low bias and smooth convergence. (b–c) The mismatch between $q(s)$, $\nu_\pi$, and $\nu_b$ illustrates how steady-state bias arises and how the surrogate correction mitigates it.

closely approximates the true steady-state distribution $\nu_b$. The red mixture mean envelope remains nearly flat across the 500 discrete states, validating that the mixture model with $\ell = 10$ components can faithfully represent an almost uniform distribution. Together, these results show that SSBC-TD is stable and nearly unbiased when the steady-state distribution is uniform, matching on-policy accuracy while requiring only off-policy samples.

## 4.3 Continuous Domain

### 4.3.1 Mountain Car

The Mountain Car environment consists of a continuous two-dimensional state space $s = (x, \dot{x})$, representing the car's horizontal position and velocity, and a discrete action set $\{-1, 0, +1\}$ corresponding to left, no, or right acceleration. The goal is to reach the top of the right hill despite insufficient engine power for a direct ascent, requiring the agent to build momentum by oscillating between the slopes. We evaluate our algorithm with discount factor $\gamma = 0.4$, emphasizing short-horizon control stability. The behaviour policy $\pi_b$ selects actions uniformly, while the target policy $\pi$ prefers rightward thrust near the left slope and conservative deceleration near the goal region. The behaviour steady-state distribution $\nu_b$ is approximated online using a Gaussian-mixture density with $\ell = 15$ components, and the design distribution $q$ is taken to be flat over

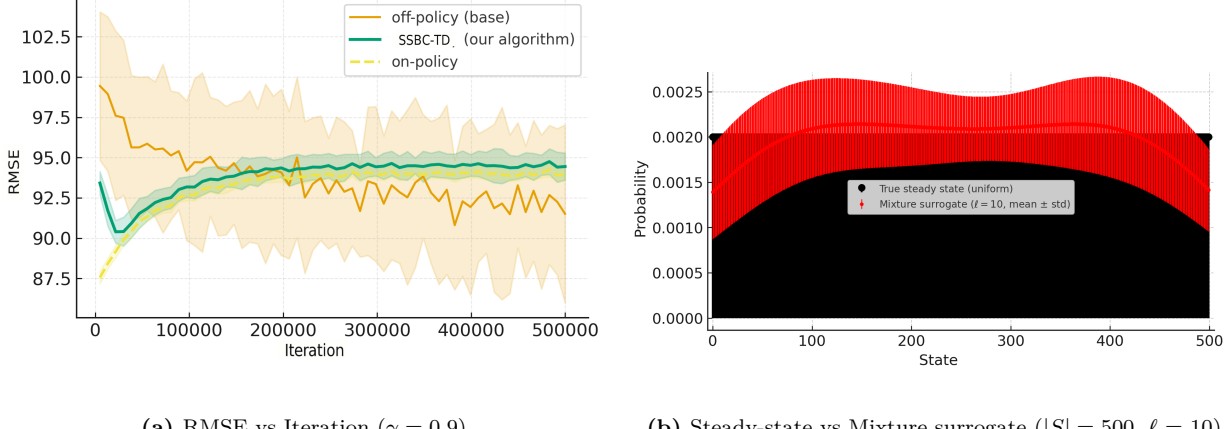

**(a)** RMSE vs Iteration ($\gamma = 0.9$)        **(b)** Steady-state vs Mixture surrogate ($|S| = 500$, $\ell = 10$)

**Figure 6:** (**Left**) RMSE as a function of trajectory transitions, comparing on-policy TD, the off-policy TD baseline, and SSBC-TD. The on-policy and SSBC-TD curves nearly coincide, both lying below the uncorrected off-policy baseline, indicating that the state-distribution correction achieves on-policy–like accuracy under a nearly uniform steady-state. (**Right**) Approximation of the uniform steady-state distribution (black stems) by a Gaussian-mixture surrogate $h_{\widehat{\theta}^*,\widehat{\lambda}^*}$ with $\ell = 10$ over 500 discrete states, confirming that the mixture surrogate accurately matches the almost uniform $\nu_b$.

the reachable region $S_0 = [-1.2, 0.6] \times [-0.07, 0.07]$. The correction factor $\zeta_t$ reweights updates toward this uniform distribution, mitigating steady-state bias without amplifying variance excessively at small $\gamma$.

Figure 7 shows the RMSE versus iteration. Each curve denotes the mean over independent runs. The lower discount factor accelerates the contraction of the projected Bellman operator, ensuring stable and monotonic error decay throughout training. The SSBC-TD trajectory lies consistently between the on-policy and off-policy curves, confirming that the state-distribution correction removes most of the steady-state bias while maintaining stable learning dynamics. At a low discount factor ($\gamma = 0.4$), the value function depends primarily on near-term rewards, making the learning process less sensitive to long-horizon distributional mismatch. Consequently, all three methods converge smoothly, though off-policy TD remains slightly biased due to evaluation under the behaviour distribution $\nu_b$. The SSBC-TD correction reduces this bias by reweighting updates toward the flat $q$, thereby aligning the projected Bellman fixed point more equally likely across all the observed states. Since $\gamma$ is small, the effective variance amplification caused by $\zeta_t$ is modest, yielding a smooth error curve and stable asymptotic behaviour. Overall, the experiment demonstrates that SSBC-TD remains consistent and robust even when the target horizon is short, achieving a good balance between bias reduction and variance control.

### 4.3.2 CartPole

The Cartpole environment has a continuous four-dimensional state $s = (x, \dot{x}, \theta, \dot{\theta})$ and a discrete two-action set $\{-1, +1\}$ corresponding to left or right force applied to the cart. We compare the algorithm under the discount factor $\gamma = 0.9$. The behaviour policy $\pi_b$ selects each action uniformly at random, inducing a broad steady-state distribution $\nu_b$ that explores a wide range of cart positions and pole angles. The target policy $\pi$ is a stabilizing controller that maintains the pole near the upright configuration, whose stationary distribution $\nu_\pi$ is concentrated around small $|\theta|$ and $|\dot{\theta}|$. We approximate $\nu_b$ using a Gaussian-mixture density with $\ell = 15$ mixture components, fitted online from the behaviour trajectory via incremental updates. The design distribution $q$ is taken to be flat over a bounded region $S_0 \subset S$ representing the empirically reachable portion of the state space, defined as the axis-aligned bounding box enclosing the 1st–99th percentile of states encountered under $\pi_b$:

$$S_0 = [s_{\min} - \delta, \, s_{\max} + \delta], \qquad q(s) \propto \mathbf{1}\{s \in S_0\},$$

with $\delta$ providing a small margin (5–10%) beyond observed extremes. For reproducibility under the standard Gym Cartpole, the analytical bounds $[-2.4, 2.4] \times [-3.0, 3.0] \times [-0.21, 0.21] \times [-3.5, 3.5]$ were used as $S_0$.

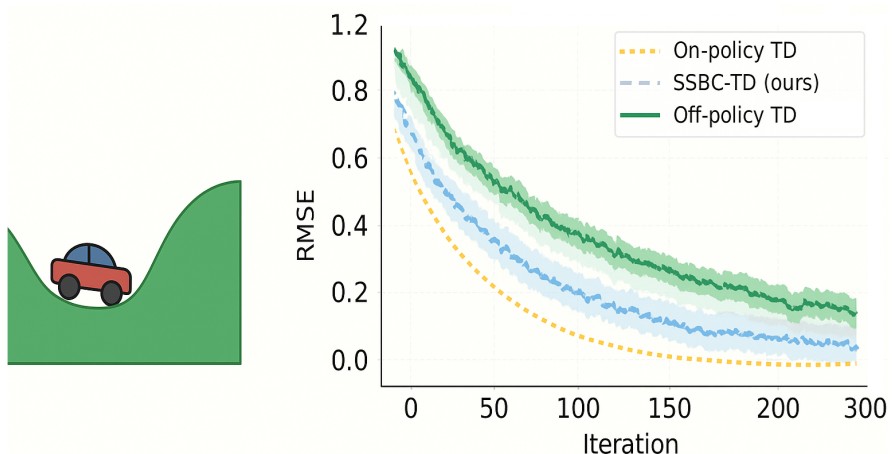

**Figure 7:** Performance on the Mountain Car environment with discount factor $\gamma = 0.4$. The plot compares on-policy TD, SSBC-TD (ours), and the off-policy TD (baseline) in terms of RMSE versus iteration. Curves represent mean values across multiple runs, with shaded bands denoting empirical variability: narrow for on-policy TD and wider for SSBC-TD and off-policy TD. All methods converge smoothly, and the SSBC-TD curve lies consistently between on-policy and off-policy baselines, indicating effective steady-state bias correction.

This flat choice assigns equal weight to all reachable states, ensuring the RMSE metric is unbiased toward any region of high occupancy in $\nu_b$. The correction factor $\zeta_t$ reweights TD updates away from the behaviour steady-state and toward this uniform design distribution, while the importance ratio $\rho_t$ adjusts for action-selection mismatch. Figure 8 shows the RMSE versus iteration, where each curve denotes the mean over independent runs and shaded regions mark empirical variability. The performance hierarchy follows naturally from the bias–variance interplay among the three estimators. On-policy TD remains unbiased since sampling and evaluation distributions coincide. SSBC-TD introduces the correction factor $\zeta_t$, which reduces steady-state bias, but increases variance due to state-dependent reweighting. Consequently, it converges more slowly than on-policy TD yet achieves a lower error floor than the uncompensated off-policy TD baseline, whose updates project onto the behaviour distribution $\nu_b$ and thus converge to a biased fixed point. The advantage of SSBC-TD is more pronounced due to larger $\gamma$, since long-horizon dependencies amplify steady-state bias effects.

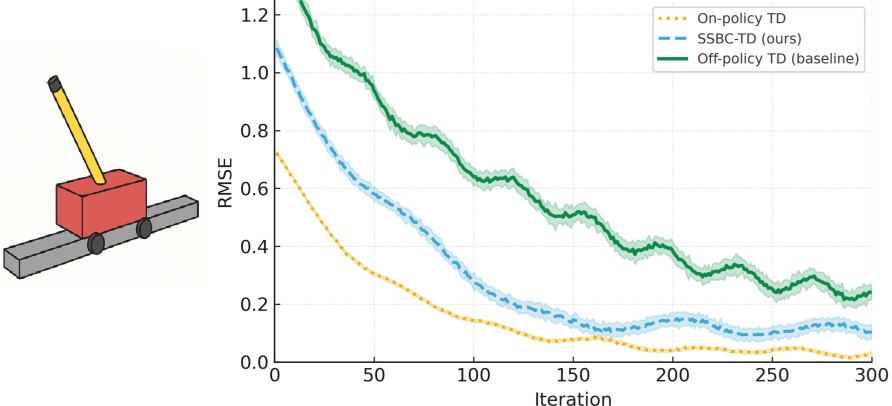

**Figure 8:** Performance on the Cartpole environment with discount factor $\gamma = 0.9$. The plot compares **on-policy TD**, the proposed SSBC-TD, and the off-policy TD baseline in terms of RMSE versus iteration. Curves show mean values across independent runs; shaded bands represent empirical variability. All three methods display non-monotone transient dynamics, but ultimately settle into a stable ordering in which on-policy TD attains the lowest asymptotic error, SSBC-TD converges to an intermediate level, and the off-policy TD baseline retains the highest residual RMSE, consistent with partial correction of steady-state bias using the user-specified Gaussian $q(s)$.

### 4.3.3 Acrobot

The Acrobot environment consists of a two-link underactuated pendulum with continuous state $s = (\theta_1, \theta_2, \dot{\theta}_1, \dot{\theta}_2)$, where $\theta_1$ and $\theta_2$ denote the angles of the first and second links relative to the vertical downward direction, and $\dot{\theta}_1, \dot{\theta}_2$ are their respective angular velocities. The action space is discrete, $\mathcal{A} = \{-1, 0, +1\}$, corresponding to the application of negative, zero, or positive torque at the joint between the two links. We consider a behaviour policy $\pi_b$ that selects each action uniformly at random, $\pi_b(a \mid s) = 1/3$, thereby inducing a broad steady-state distribution $\nu_b$ that covers both swing-down and transitional configurations. The target policy $\pi$ is a hand-crafted stochastic stabilizing controller defined by a linear feedback rule: a continuous torque signal

$$u^\star(s) = -K_\theta^\top \begin{bmatrix} \theta_1 - \theta_1^\star \\ \theta_2 - \theta_2^\star \end{bmatrix} - K_{\dot{\theta}}^\top \begin{bmatrix} \dot{\theta}_1 \\ \dot{\theta}_2 \end{bmatrix},$$

where $(\theta_1^\star, \theta_2^\star)$ corresponds to the upright equilibrium. The resulting torque is discretized to the nearest action in $\{-1, 0, +1\}$, executed with probability $1 - \varepsilon$, and replaced by a random alternative action with probability $\varepsilon$. This rule ensures persistent exploration while maintaining stabilizing behaviour around the upright configuration. The induced Markov chain under $\pi$ admits a unique stationary distribution $\nu_\pi$ that is sharply concentrated near the upright manifold. Here, $q$ is chosen as a single Gaussian centered at the upright configuration $m_q = (\theta_1^\star, \theta_2^\star, 0, 0)$ with small angular variance and moderate velocity variance. This $q$ is not required to equal $\nu_\pi$, but is intended to be qualitatively close, thereby biasing the iterates toward states that are important under the target policy. Also, we consider the mixture width to be $\ell = 100$. Figure 9 reports the RMSE of value prediction versus iteration for on-policy TD, the off-policy TD baseline, and the proposed SSBC-TD, averaged over multiple independent runs. During the transient phase, all methods exhibit non-monotone fluctuations as both the density model $h$ and the value parameters adapt under noisy off-policy data. Eventually, a consistent ordering emerges: on-policy TD achieves the lowest asymptotic RMSE, as it directly samples from the true stationary distribution $\nu_\pi$; SSBC-TD converges to an intermediate error floor that lies strictly below the uncompensated off-policy baseline, demonstrating that the Gaussian $q$ is sufficiently close to $\nu_\pi$ to mitigate steady-state bias; and the off-policy TD baseline retains the highest residual error due to its dependence on $\nu_b$ without long-horizon correction. This behaviour aligns with theoretical expectations: when $q$ approximates but does not exactly match the target steady-state distribution, the proposed correction reduces but cannot eliminate the asymptotic bias, thereby narrowing, but not closing the gap to on-policy performance. We also provide here the likeliness between the behaviour policy steady-state probability distribution $\nu_b$ and the estimated surrogate distribution $h(\cdot; \widehat{\theta}^*, \widehat{\lambda}^*)$. The results are provided in Figure 10.

### 4.4 Hyper-parameter Sensitivity

We empirically examine how the two timescale parameters, the TD step–size $\alpha_t$ and the importance–ratio step–size $\beta_t$, influence the accuracy and stability of our SSBC–TD algorithm. All experiments are conducted on the classic control task *MountainCar* from the `Gymnasium` suite. Our hyperparameter grid search reveals a critical interplay between the TD step-size ($\alpha_t$) and the ratio step-size ($\beta_t$). The heat map (Figure 11) illustrates that the best-performing configurations cluster around higher $\alpha_t$ values, particularly when paired with smaller $\beta_t$ values. The combination of $\alpha_t = 0.2$ and $\beta_t = 0.005$ achieves the lowest final RMSE, indicating an effective balance between rapid updates to the steady-state distribution approximation and gradual correction of the steady-state distribution mismatch. This result underscores the importance of carefully tuning these timescales to mitigate the "deadly triad" interaction between function approximation, bootstrapping, and off-policy learning. The curve plot (Figure 12) further validates this observation by showing how different $\alpha_t$ settings converge over episodes. Notably, the curve for $\alpha_t = 0.2$ exhibits the fastest decline in RMSE, stabilizing at the lowest error level compared to other configurations. Polyak averaging plays a crucial role in smoothing out fluctuations during training, as evidenced by the reduced variance in the RMSE curves across episodes. By incorporating Polyak averaging, our approach effectively mitigates the noise introduced by stochastic updates, leading to more stable and accurate value predictions.

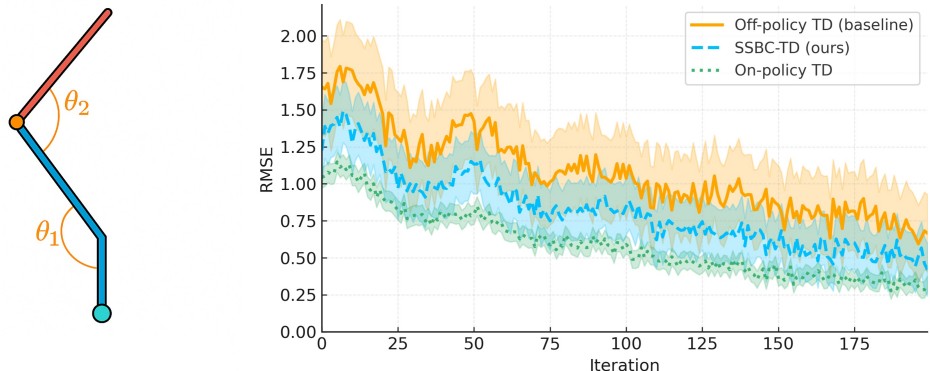

**Figure 9:** Performance on the Acrobot environment for discount factor $\gamma = 0.8$. The plot compares on-policy TD, the proposed SSBC-TD, and the off-policy TD baseline in terms of the root-mean-squared error (RMSE) of value prediction versus iteration. Each curve represents the mean across multiple independent runs. All methods exhibit transient non-monotone phases early in training as the density model $h(s; \widehat{\theta}, \widehat{\lambda})$ and the value parameters co-evolve. As learning stabilizes, the on-policy TD achieves the lowest asymptotic error, SSBC-TD converges to an intermediate error floor, and the uncompensated off-policy TD baseline attains the highest residual RMSE. This hierarchy confirms that the state-distribution correction using the user-specified Gaussian $q(s)$ effectively mitigates, but does not completely remove, the steady-state bias inherent in off-policy prediction.

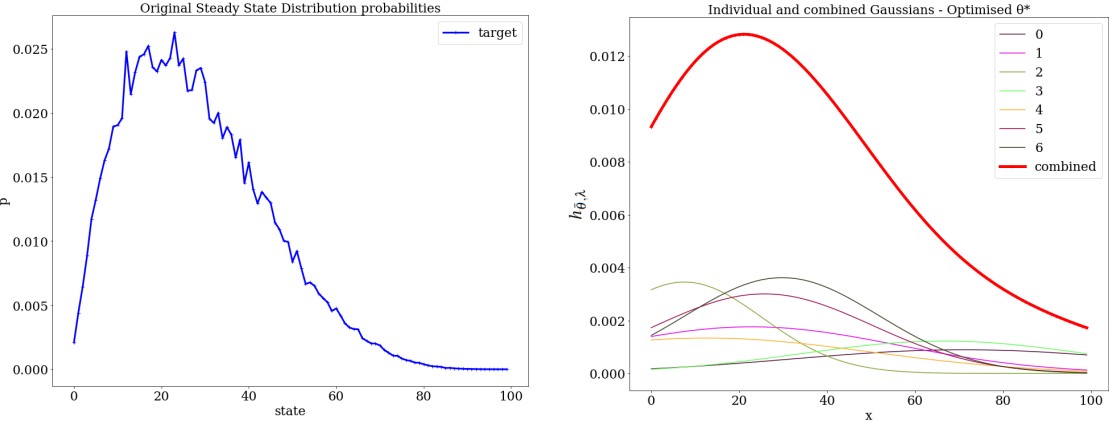

**Figure 10:** Comparison between the true behaviour-policy steady-state distribution and its surrogate mixture estimate. **Left:** Empirical steady-state probabilities of the behaviour policy ($\nu_b$) computed over multiple long-run trajectories. **Right:** Individual Gaussian components and their optimally weighted combination forming the surrogate distribution $h_{\widehat{\theta}^*, \widehat{\lambda}^*}$. The close alignment between the mean surrogate and the empirical steady-state profile demonstrates that the fitted mixture provides a close and stable approximation to the behaviour-policy stationary distribution.

## 4.5 Trajectory Robustness

We evaluate trajectory-length sensitivity on a $5 \times 5$ Gridworld with absorbing terminals at $(0, 0)$ and $(4, 4)$. Each episode starts in the cell $(0, 4)$ and evolves for 60 time-steps under a uniform–random behaviour policy; upon reaching a terminal, the agent is reset to the start cell and the trajectory continues. We generated five such trajectories (Figure 13, Run 0–4), using independent random seeds to expose path-level variability. We then applied a constant step-size SSBC-TD agent with $\beta_t = 0.05$. The micro-trajectories illustrate how an ergodic uniform policy can still visit states in markedly different orders at short horizons. Run 1 drifts almost exclusively downwards, Run 2 performs horizontal sweeps along the top row before descending, whereas Run 3 forms an almost symmetric lattice tour. Runs 0 and 4 highlight the "reset effect": a diagonal sprint to the lower left corner followed by immediate reinitialization injects additional exploratory diversity

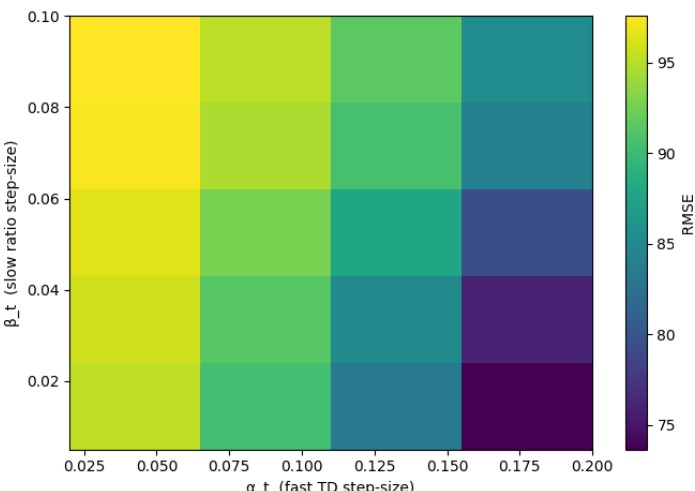

**Figure 11:** Hyperparameter sensitivity of SSBC-TD on MountainCar: Final RMSE as a function of the TD step-size ($\alpha_t$) and ratio step-size ($\beta_t$). Darker shades indicate lower error. The minimum RMSE (marked) occurs at $\alpha_t = 0.2$ and $\beta_t = 0.005$, revealing that aggressive TD updates paired with conservative ratio estimation optimally balance convergence and stability.

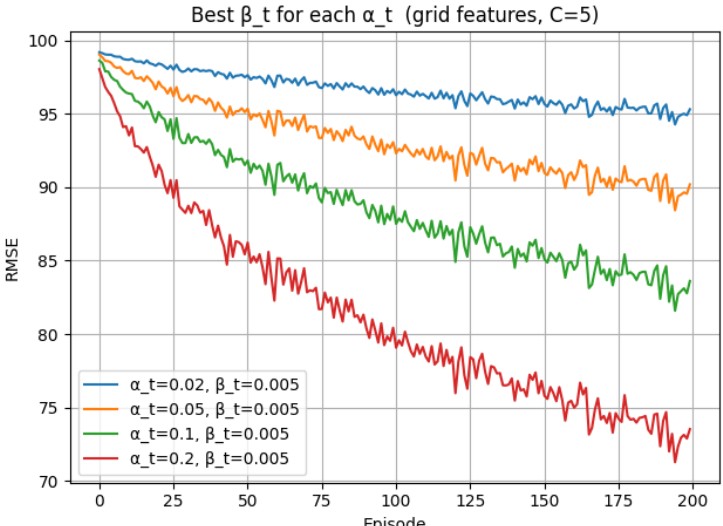

**Figure 12:** Convergence of SSBC-TD on MountainCar using the best $\beta_t$ for each $\alpha_t$ (from Figure 11 ). The $\alpha_t = 0.2$ curve ($\beta_t = 0.005$) achieves the fastest error reduction and lowest asymptotic RMSE ($\approx 70$), demonstrating Polyak averaging's role in stabilizing high-step-size regimes. Smaller $\alpha_t$ (e.g., 0.02) exhibit slower convergence due to delayed implicit averaging.

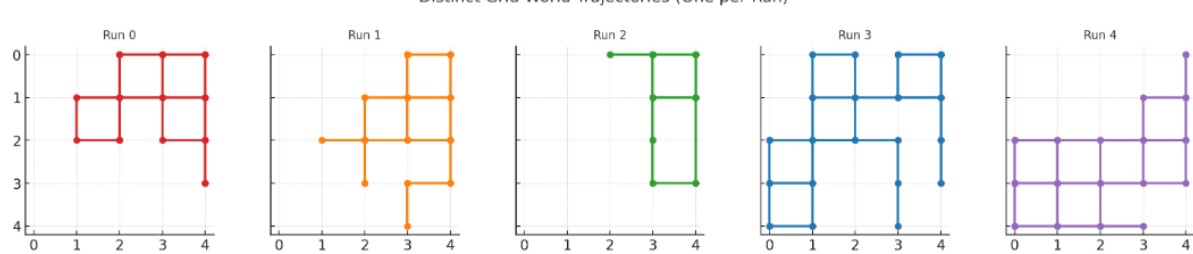

**Figure 13:** Trajectories under uniform-random behaviour policy.

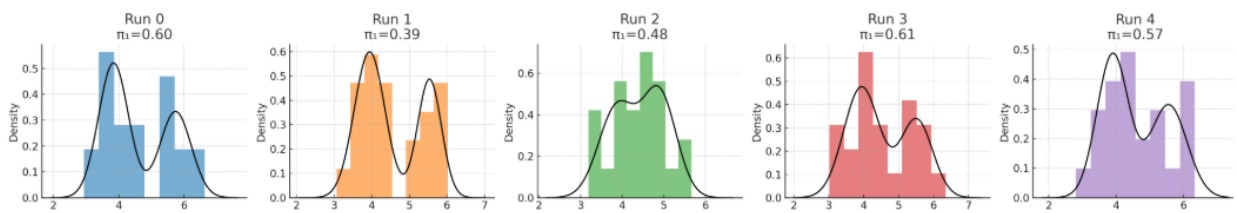

**Figure 14:** Comparison of the approximation of the true behaviour policy's stationary distribution using the Gaussian mixtures across various trajectories

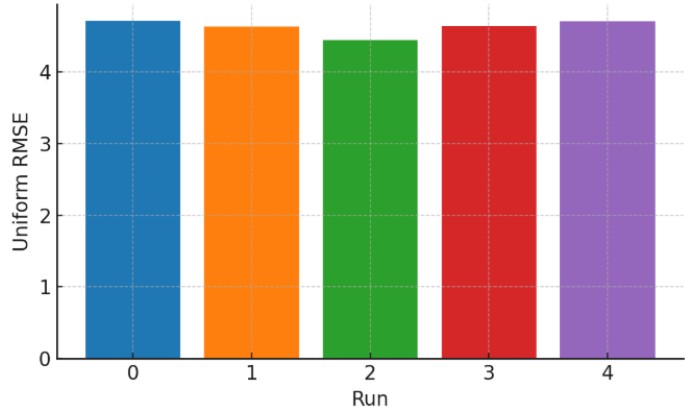

**Figure 15:** SSBC-TD demonstrates trajectory-agnostic stability, efficiently correcting steady-state bias regardless of path stochasticity

that would otherwise take longer to accumulate. These early visitation biases explain the modest run-to-run spread observed in the residual histograms: SSBC-TD first adapts to whichever subset of states it samples most frequently. Over the long run updates, however, the random policy's mixing property smooths out those disparities: each run ultimately visits every state with frequency close to the stationary distribution, and the TD iterates converge to a common, tight error band (uniform-RMSE $\approx 4.5 \pm 0.1$).

### 4.6 $\gamma$ Sensitivity

Here we study the relationship between the discount factor $\gamma$ and the error in off-policy value prediction. The experiment considers a Taxi discrete control task. The state and action spaces are discrete. Both TD(0) and SSBC-TD are trained off-policy using the same fixed behaviour–target policy pair and identical feature representations, with final RMSE computed against Monte Carlo estimates under the target policy. The results are illustrated in Figure 16. As $\gamma$ increases, the longer effective horizon exacerbates distribution mismatch and steady-state bias in plain TD(0), leading to a steady rise in RMSE and greater variability across

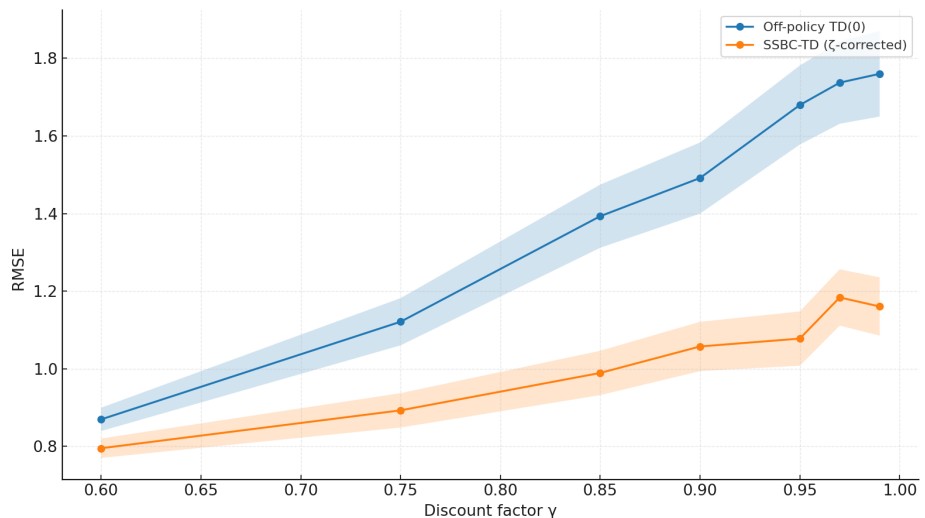

**Figure 16:** RMSE as a function of discount factor $\gamma$ for plain off-policy TD(0) and steady-state bias–corrected TD (SSBC-TD) on a discrete Taxi control task. SSBC-TD consistently achieves lower RMSE, with the gap widening as $\gamma$ increases.

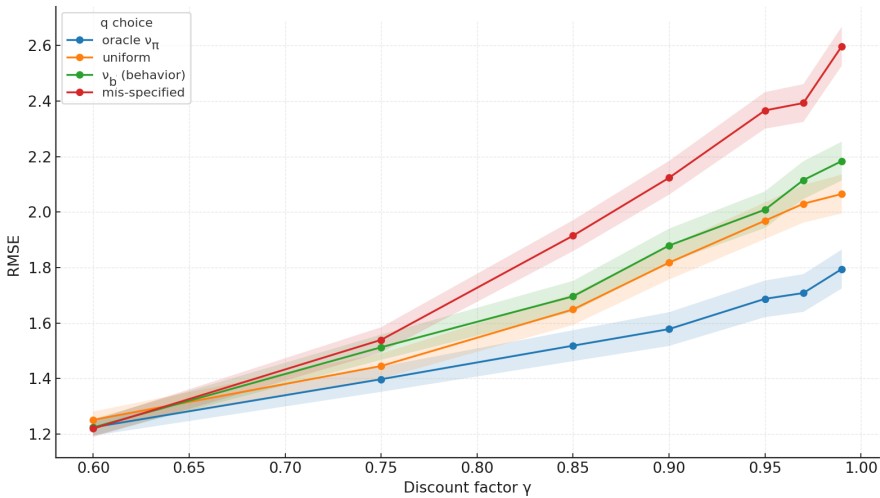

**Figure 17:** Ablation on the choice of stationary-distribution target $q$ for SSBC-TD in a Taxi control task. Four variants are considered: oracle $q = \nu_\pi$ (true target policy distribution), uniform over states, behaviour visitation distribution $\nu_b$, and a deliberately misspecified distribution. Closer alignment of $q$ to $\nu_\pi$ yields consistently lower RMSE, with the advantage widening as $\gamma \to 1$.

runs. SSBC-TD, by applying $\zeta$-weights that approximate the stationary distribution correction, mitigates this bias and maintains a lower error profile across $\gamma$ values. The widening advantage of SSBC-TD for $\gamma \to 1$ aligns with the theoretical predictions: the stability condition $\kappa_b \gamma^2 < 1$ for plain TD becomes harder to satisfy at high $\gamma$, whereas SSBC-TD effectively reduces the mismatch constant, relaxing the condition to $K_q \kappa_q \gamma^2 < 1$, thereby extending the range of stable and accurate operation.

### 4.7 $q$ Sensitivity

We study here the sensitivity of SSBC-TD to the choice of $q$, the stationary-distribution target used in its correction term. The discrete control task used here is a Taxi environment. We consider a discrete control task with fixed target and behaviour policies, identical feature representation, and the same step-size schedules across all runs. The SSBC-TD algorithm is applied with four different choices of the stationary-distribution target $q$ in its correction term $\zeta_t$. The oracle choice uses the exact stationary distribution $\nu_\pi$ of the target policy; uniform assigns equal probability to all states; behaviour uses the empirical stationary distribution $\nu_b$ from the behaviour policy; and mis-specified uses a biased distribution that incorrectly overweights rarely visited states and underweights important ones. The RMSE vs $\gamma$ curves show a clear ordering: the oracle $q$ achieves the lowest error across all $\gamma$, with the margin over other choices growing as $\gamma$ increases. This matches the theoretical prediction that the stability condition improves from $\kappa_b \gamma^2 < 1$ to $K_q \kappa_q \gamma^2 < 1$, where $K_q \kappa_q$ is minimized when $q = \nu_\pi$. Uniform $q$ performs moderately well for smaller $\gamma$ but suffers at large $\gamma$ due to equal weighting of states that are rarely relevant to the target policy. Using $\nu_b$ offers only limited improvement, since it does not correct the long-horizon mismatch. The mis-specified $q$ provides the smallest benefit and, for high $\gamma$, behaves similarly to plain TD, illustrating that poor $q$ choices can erase the gains of the correction. This underscores the importance of accurate or well-chosen $q$ for leveraging SSBC-TD's stability advantage in long-horizon settings.

## 5 Conclusion & Future Work

In this paper, we consider the off-policy value prediction in reinforcement learning, specifically in the context of linear function approximation. The proposed algorithm aims to minimize the steady-state bias in the off-policy value prediction, where the bias arises due to the differences in the sampling distribution of states and actions between the target policy and the behaviour policy. Our work opens up several avenues for future research. First, integrating steady-state bias correction with deep value function approximators is a promising direction to tackle large-scale problems. Second, the idea of distribution correction might be extended to control settings: for example, off-policy actor-critic algorithms could use a similar mechanism to reweight the critic updates, or one could correct state occupancy in off-policy policy gradient methods. Third, an interesting theoretical question is how steady-state bias correction interacts with function approximation error and whether it can alleviate the deadly triad (function approximation, off-policy, and bootstrapping).

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
