# OpenReview forum: "Mitigating Steady-State Bias in Off-Policy TD Learning via Distributional Correction"
_TMLR — Accepted by TMLR_

### Review · Reviewer_Hwf3 · 2025-08-31

**Summary Of Contributions:**

This paper studies the distribution mismatch problem in off-policy value prediction using linear value function approximation. The first contribution of the paper is an off-policy TD error bound relating approximation error to distribution shift and discount factor (theorem 1 & 2). This result motivates the proposed algorithm, where a policy distribution correction ratio and a marginal state distribution correction ratio are added to the standard TD learning update rule. An error bound is provided for this algorithm. The numerical experiments are conducted on a set of small discrete environments and discretized continuous environments such as mountain car and cartpole. The proposed algorithm is shown to produce lower value error than both regular off-policy and on-policy. Some ablation experiments are conducted to validate the theoretical analysis.

**Audience:**

Yes

**Audience Explanation:**

I believe the off-policy error bounds, especially theorem 1, is of value to the RL community. However, I am not confident whether or not similar results have been published before.

**Claims And Evidence:**

No

**Claims Explanation:**

As I detail in the "requested changes" section, the paper is lacking some experimental details, making the results hard to judge. I'll be happy to revise this answer if my questions are clarified.

**Requested Changes:**

Questions/comments:
* In (4), could you specify what $\Xi$ represents?
* The authors discussed choosing the target distribution $q(s)$ as a reasonable proxy for the target policy. But this could be a pretty strong assumption given the behavior of the target policy is most realistically unknown. Otherwise there's no need to evaluate it.
* Is there a way to compare the error bound in theorem 1 and theorem 6?
* In the off-policy evaluation setting, you already have data from the behavior policy. Why using a two-time scale learning method instead of a two-stage learning method, where you first learn the marginal distribution of the behavior policy, freeze that distribution, and then do TD learning separately? In that case you don't need to tune the relative scale of the two learning rates.
* I believe the authors should add more details about the experiments. Please see the next 3 questions.
* The authors mentioned discretizing the continuous environments. Did you still use features to represent the value functions or the value functions are tabular? If the latter then the theory and experiment are not really matching?
* For a number of environments, the authors randomly chose the target and behavior policy. I believe this potentially makes the problem trivial. For example, in mountain car, a random policy is unlikely to receive any reward, making value prediction the problem of predicting a constant.
* In most plots, the authors showed value error by horizon/trajectory length. Is horizon/trajectory length the number of environment steps that the behavior policy was rolled out to collect data? If the behavior policy is stochastic, the data could be quite random/vary a lot between each rollout? If that's the case, why the green and red learning curves are so smooth? Shouldn't you repeat the experiment a few times for each configuration?
* (Minor) At the beginning of section 4, you said "The discount factor is fixed at γ = 0.1 universally". This seems way to small for standard applications, where the discount factor is usually set between 0.9 and 0.99?
* (Minor) In section 4.6, the authors could add a reference to figure 18.

---

> ### Author Response · Authors · 2025-09-01
> **Response**
>
> We thank the reviewer for the constructive review
>
> **1.**  In (4), $\Xi_{\nu_b}$ denotes a diagonal matrix whose entries correspond to the stationary distribution  $\nu_b$ of the behavior policy: $\Xi_{\nu_b} = \operatorname{diag}\big(\nu_b(s_1),\nu_b(s_2),\ldots,\nu_b(s_n)\big).$
> This matrix weights the Bellman error according to state visitation frequencies under the behavior policy.
>
> **2. Choice of Target Distribution**
>
> Our method does not assume access to the target policy’s stationary distribution, nor do we require
> $q$ to approximate it. In fact, the design distribution $q$ is not intended to represent the true stationary distribution of the target policy. Instead, $q$ serves as a user-specified emphasis distribution that reflects the states on which accurate value prediction is desired. For example. $q$ may be chosen as the uniform distribution, which recovers the classical “equal-weighting across states” objective used in value and policy iteration.
> $q$ may encode safety-critical or task-relevant states in domains where the practitioner wishes to emphasize certain regions regardless of the target policy’s stationary behavior.
> When the target policy is unknown or hypothetical (common in off-policy prediction), $q$
>  provides a stable and well-defined surrogate objective even without access to $\nu_\pi$.	​
>
> **3.  Error Bounds (Thm 1 vs Thm 6)**
>
> (Theorem numbers are w.r.t. Revision 1) Thm 1 characterizes the error of standard off-policy TD, scaling with the mismatch coefficient  $\kappa_b$. Thm 6 shows that after our correction the error scales with $\kappa_q$, which can be much smaller if $q$ is chosen wisely. The constant $C$ in Thm 6 contains terms such as spectrum of $\Lambda_c$  and $\max_{s}\frac{\nu_b(s)}{q(s)}$
> quantifying an appropriate choice of $q$.
>
> **4. Two-Timescale vs Two-Stage**
>
> We employ a two-timescale stochastic approximation algorithm rather than a two-stage method. This requires the step-sizes to be compatible (32). Our algorithm is fully online using a single stream of experience; all parameters are updated per transition $(s_t,a_t,r_{t+1},s_{t+1})$. This is computationally efficient since $h(\cdot;\hat\theta^{t},\hat\lambda^{t})$ continuously tracks the steady-state distribution in parallel with TD learning. A two-stage method would require first estimating $\nu_b$ in batch, freezing it, and then running TD---less flexible, non-adaptive, non-online. Our online single-pass method is more general and aligns with the incremental nature of TD.  Further, the two-timescale SA allows convergence proofs under Markovian noise (Borkar  08). The slower timescale (Polyak--Ruppert avg) for the distribution parameters $(\hat\theta^{t},\hat\lambda^{t})$ yields quasi-stationarity from the perspective of faster TD updates, enabling effective decoupling.
>
> **5. Discretization and LFA**
>
> In Revision 1 of the paper, for continuous domains (MountainCar, CartPole), we discretized the state space and state vector itself is chosen as the feature. Hence we don't use any tables as the states are directly provided from the env. In Revision 2 of the paper, we have tested the continuous domain without discretization by directly operating in the continuum.
>
> **6. Policy Selection**
>
> We have removed random policy selection and now use carefully designed target–behaviour policy pairs that provide a meaningful and non-trivial evaluation of the proposed algorithm.
>
> **7. Smoothness of Curves**
>
> We now report mean performance across multiple runs along with error bands, providing a clearer picture of variability and stability. Furthermore, each algorithm is evaluated on a sufficiently long behaviour-policy trajectory, ensuring that all state–action pairs occur with adequate frequency, since the behaviour policy is ergodic.
>
> **Discount $\gamma=0.1$**
>
> We have now modified the experiments to include wide range of discount factors.  Also, in Sec 4.6, we perform abalation studies with respect to $\gamma$-sensitivity where we vary $\gamma$ and observe that our method outperforms baseline TD especially as $\gamma\to 1$.

---

> > ### Comment · Reviewer_Hwf3 · 2025-11-19
> >
> > Thank the authors for the clarification and paper updates. I have no other questions.

---

### Review · Reviewer_X7Kr · 2025-09-24

**Summary Of Contributions:**

The paper considers the problem of off-policy value function estimation in reinforcement learning with linear function approximation. The key challenge mentioned by the paper is steady-state bias where the learned value function parameter converges to a solution projected with respect to the behavior policy's stationary state distribution rather than the target policy's. The paper proposes an algorithm named Steady-State Bias Correction (SSBC) which introduces a second importance sampling-style correction to the standard temporal difference (TD) update (Equation (33)). This correction factor is the ratio of a predefined user-chosen state distribution, q(s), to an online estimate of the behavior policy's stationary distribution. The distribution h(s) is modeled as a Gaussian Mixture Model (GMM) and its parameters are learned concurrently with the value function via stochastic gradient ascent on a faster timescale. Experiments report RMSE of value prediction across small discrete/continuous control tasks, with ablations on stepsizes, trajectories, discount factor, and the choice of q.

**Audience:**

Yes

**Audience Explanation:**

The paper considers an important problem.

**Claims And Evidence:**

Yes

**Claims Explanation:**

- Overall, the theory seems reasonable and confirms the intuition that if one has a good model of the distribution mismatch (h) and a good target to correct towards (q), one can fix the problem. The other challenge which still remain is the practical difficulty of obtaining these good models and targets.

- Consistent On-Line Off-Policy Evaluation Hallak & Mannor, 2017 seems to have looked at similar problem ("distribution-mismatch bias in off-policy TD with function approximation") earlier but is not cited in the paper. Please mention the similarities or differences with their proposed approach.

**Requested Changes:**

- The writing of the paper can be improved significantly. The paragraphs and sentences are too long making it harder to read.

---

> ### Author Response · Authors · 2025-11-16
> **Response**
>
> We thank the reviewer for the constructive feedback. We have revised the manuscript to address all points raised.
>
>
> [Comment] “The writing of the paper can be improved significantly. The paragraphs and sentences are too long making it harder to read.”
>
> [Response] We appreciate this observation. The manuscript has now been thoroughly revised to improve exposition. We have split long sentences,  and restructured dense paragraphs. Also, transitions between sections have been simplified. We have revamped the Introduction, Related Work, and Algorithm sections. We believe the paper is now much more comprehensible.
>
> [Comment] “Consistent On-Line Off-Policy Evaluation (Hallak & Mannor, 2017) looked at a similar problem earlier but is not cited. Please mention similarities or differences.”
>
> [Response] Thank you for pointing out this important prior work. We have now added a detailed comparison to Hallak & Mannor (2017) in the Related Work sections (last para of Introduction). Specifically, we highlight that their method learns the stationary density ratio by solving a constrained fixed-point equation, whereas our approach avoids the instability of this fixed-point system by directly modeling the behaviour policy's stationary distribution  through a parametric Gaussian mixture surrogate. We further clarify that our method supports arbitrary design distributions $q$, enabling emphasis weighting beyond the target policy—an aspect that complements and generalizes classical density-ratio correction methods.
>
> [Comment]
> “The theory seems reasonable… but there remains the practical difficulty of obtaining good models and targets.”
>
> [Response] We have expanded the discussion to clarify the practical role of the design distribution $q$ and the behavior distribution model $h$. The revised version emphasizes that $q$ need not approximate the target stationary distribution; instead, it can be selected as a uniform, safety-weighted, or task-prioritized distribution. We also include new experiments illustrating robustness to imperfect $h$ and multiple choices of $q$ and we provide additional remarks on how the spectral negativity condition ensuring convergence is preserved even under model mismatch

---

### Review · Reviewer_q7Tc · 2025-11-03

**Summary Of Contributions:**

This paper proposes an algorithm for off-policy value prediction with linear function approximation, which mitigates the steady-state bias by correcting the steady-state distribution discrepancies. The method first estimates the steady-state distribution of the behavior policy with a Gaussian mixture model, then uses the ratio between the probability of a given target distribution and this estimated probability to correct steady-state bias in the TD update. The paper provides theoretical guarantees on the value estimation error and demonstrates empirical performance on several environments.

**Strengths**
1. The off-policy evaluation problem addressed in this paper is an important area in reinforcement learning.
2. The paper provides theoretical proof for key intermediate conclusions such as distribution approximation error, and presents an error bound for the predicted value function.

**Weaknesses**
1. The proposed algorithm relies on a predefined target distribution $q$, yet in most off-policy evaluation settings, the target policy's true stationary distribution is unknown.
2. The main experiments set $\gamma=0.1$, which cannot demonstrate the policy evaluation ability in long-horizon scenarios. Additionally, in most experiments, the standard deviation across independent runs is not reported.
3. The main error bound in Theorem 6 is for $\lVert \Phi x_c^{TD} - V^\pi \rVert_{\nu_b}$ rather than $\lVert \Phi x_c^{TD} - V^\pi \rVert_{q}$. Adding a bound that relates $\lVert \Phi x_c^{TD} - V^\pi \rVert_{q}$ to the RHS via $\max_s \frac{q(s)}{\nu_b(s)}$ would make the conclusion clearer.

**Audience:**

Yes

**Audience Explanation:**

The paper tackles the off-policy value prediction problem in reinforcement learning, an important problem that fits TMLR’s scope.

**Broader Impact Concerns:**

None.

**Claims And Evidence:**

Yes

**Claims Explanation:**

Overall, the proposed algorithm and theorems are well supported by theoretical proofs and empirical experiments.

**Requested Changes:**

1. (Critical) There are a few places where the paper is not rigorous. For example, in the discussion after Theorem 3, reducing the kernel bandwidth $\sigma$ will loosen the approximation rather than tighten it, as indicated by Theorem 3 and Figure 3. In Equation 24, it should be $\leq$ rather than $=$ according to Lemma 1.
2. (Strengthen) The main experiments should also test a larger $\gamma$ value. More baselines should be included, and standard deviations across independent runs should be reported.
3. (Strengthen) More detailed discussion of how this algorithm can be used when the target policy's steady-state distribution is unknown should be provided. Additionally, an error bound on $\lVert \Phi x_c^{TD} - V^\pi \rVert_{q}$ should be included in Section 3.

---

> ### Author Response · Authors · 2025-11-16
> **Response**
>
> We thank the reviewer for the constructive comments. We have revised the manuscript substantially to address all concerns
>
> **Requested Changes**
> 1. We have now made all the analysis rigorous and extensive. The errors are all removed and proofs redone for clarity. Please see the updated manuscript.
> 2. We have now expanded the experimental section to include wide range of discount factor values. We have also added more complex settings in the experimental section for thorough evaluation of the algorithm. The experiments have  been re-run and now reports mean performance over multiple independent runs along with the std deviation bands in the plots.  We also provide  all the naunced details about the execution and also the interpretation of the results. Please see the updated manuscript
> 3. SSBC does not require the true steady-state distribution of the target policy. Here $q$ is a design choice with direct parallels to emphasis functions in ETD, safety-critical weighting, and risk-sensitive RL.  For example,  $q$ may be chosen as the uniform distribution, which recovers the classical “equal-weighting across states” objective used in value and policy iteration. We have also now provided the additional error bound $\Vert \Phi x_c^{TD} - V_\pi\Vert_q$

---

### Review · Reviewer_FWmD · 2025-11-16

**Summary Of Contributions:**

I apologize for my late review. Please find my comments below.

The paper studies off-policy value prediction in RL. The problem, stated plainly, is that one generally wishes to compute the value function of a certain desired target policy, but is only given access to trajectories by a behavior policy which may differ in the steady-state. Importance sampling only corrects the single-step dynamics, the steady state may differ in a more sustained manner which calls for a more systematic approach. Figure 1 in the manuscript conveys this idea well.

They estimate the steady-state distribution of the behavior policy with a Gaussian mixture model and natural exponentiated family, and uses an importance-sampling-like approach to mitigate the bias. Specifically, their algorithm, Steady-State Bias Correction (SSBC), does an importance sampling-like correction to the standard temporal difference (TD) update. The paper shows theoretical guarantees of this algorithm along with experiments to numerically verify the results.

**Audience:**

Yes

**Audience Explanation:**

The paper solves an interesting problem in the field. While it is essentially a variant of importance sampling, the analyses provided is strong and backs up the claims in the paper.

**Claims And Evidence:**

Yes

**Claims Explanation:**

All the claims are backed with rigorous theoretical results and proofs. Upon taking a close look at the proofs, I did not find any errors.

**Requested Changes:**

- I would encourage the authors to use \citet, \citep for inline citations,
- In the second last paragraph of page 3, the authors say that the computational complexity of computing $V_\pi$ is $O(n^3)$. Technically speaking, this should be $O(n^\omega)$, where $\omega \leq 2.372$ is the exponent of matrix multiplication,
- It would be great if the authors could specify how they discretized the continuous environments in their simulations (they allude to this, but not explicitly, in the manuscript)

---

> ### Author Response · Authors · 2025-11-19
> **Response**
>
> We thank the reviewer for the postive and constructive feedback,
>
> ** Requested Changes **
>
> 1) We will certainly incorporate this suggestion in the next revision. We had submitted the current revision just before receiving this review.
>
> 2) We will update this as well in the upcoming revision.
>
> 3) In Revision 1 of the paper, for continuous domains (MountainCar and CartPole), we discretized the state space by partitioning the domain into grids, and the resulting state vector was used directly as the feature representation. This choice was aligned with our theoretical development, which assumed a discrete state space.
>
> In Revision 2, we additionally evaluated the algorithm in the continuous domain without discretization, operating directly in the continuum. This is feasible because the algorithm is agnostic to the cardinality of the state space, and our results confirm that it remains effective in this setting. Please refer the experimental section.

---

> > ### Author Response · Authors · 2025-12-01
> > **Response - 2**
> >
> > We have revised the manuscript to incorporate all the requested changes

---

### Author Response · Authors · 2025-11-24
**New version  uploaded**

**Changes**

1) A few typos fixed
2) Added few sentences to improve the exposition
3) Corrected Lemma 4 to improve the bound

---

### Decision · Action_Editor_kkCS · 2025-12-17

**Recommendation:** Accept as is

**Additional Comments:**

The reviewers reached a unanimous consensus in favor of accepting this paper.

The paper proposes an algorithm which applies an importance sampling-style correction to standard temporal difference updates for off-policy value prediction. The reviewers collectively praised the paper for its solid theoretical contributions and comprehensive experimental verification.

However, the reviewers also highlighted several limitations that should be noted:

**Restricted Distributions**: Reviewer noted that the theoretical guarantees are currently limited to Gaussian mixtures and exponentiated families, rather than more general distributions.

**Target Distribution Requirement**: Reviewers raised concerns regarding the practicality of the method, specifically the requirement of a known target distribution, which is not always available in standard off-policy evaluation (OPE) settings.

**Audience:**

Yes

**Audience Explanation:**

This paper will be interesting for RL community.

**Claims And Evidence:**

Yes

**Claims Explanation:**

The paper is theoretical solid with strong empirical study support.

---

> ### Author Response · Authors · 2026-01-16
> **Response**
>
> We thank the editors and anonymous reviewers for their constructive feedback and for the positive evaluation of this work, which helped strengthen both its technical content and exposition.